# FROM TRACE TO LINE: LLM AGENT FOR REAL-WORLD OSS VULNERABILITY LOCALIZATION

## ABSTRACT

Large language models show promise for vulnerability discovery, yet prevailing methods inspect code in isolation, struggle with long contexts, and focus on coarse function or file level detections which offers limited actionable guidance to engineers who need precise line-level localization and targeted patches in real-world software development. We present T2L-Agent (Trace-to-Line Agent), a project-level, end-to-end framework that plans its own analysis and progressively narrows scope from modules to exact vulnerable lines. T2L-Agent couples multi-round feedback with an Agentic Trace Analyzer (ATA) that fuses runtime evidence such as crash points, stack traces, and coverage deltas with AST-based code chunking, enabling iterative refinement beyond single pass predictions and translating symptoms into actionable, line-level diagnoses. To benchmark line-level vulnerability discovery, we introduce T2L-ARVO, a diverse, expert-verified 50-case benchmark spanning five crash families and real-world projects. `T2L-ARVO` is specifically designed to support both coarse-grained detection and fine-grained localization, enabling rigorous evaluation of systems that aim to move beyond file-level predictions. On `T2L-ARVO`, T2L-Agent achieves up to 58.0% detection and 54.8% line-level localization, substantially outperforming baselines. Together, the framework and benchmark push LLM-based vulnerability detection from coarse identification toward deployable, robust, precision diagnostics that reduce noise and accelerate patching in open-source software workflows.

## 1 INTRODUCTION

Software vulnerabilities now occur at unprecedented scale and cost. In 2023, more than 29,000 Common Vulnerabilities and Exposures (CVEs) were recorded CVE Details (2024). In the first half of 2025, 1,732 data breaches were reported–an 11% increase Indusface (2024). The economic toll is mounting: software supply-chain attacks are projected to cost the global economy 80.6 billion annually by 2026, up from 45.8 billion in 2023 Dark Reading (2024); some estimates put total damages at $9.5 trillion in 2024 Liu et al. (2024). Critically, 14% of breaches in 2024 began with vulnerability exploitation—nearly triple the prior year. Despite advances in automated vulnerability detection, effectiveness in localization remains underexplored Zhang et al. (2024b), leaving developers with coarse, file-level predictions while 32% of critical vulnerabilities remain unpatched for over 180 days. These vulnerabilities are exploding in volume and growing ever more severe, driving escalating security and economic risks worldwide.

**Motivation.** The rise of Large Language Models (LLMs) has accelerated AI-driven automation across software development and cybersecurity Zhang et al. (2025a). From code assistants like GitHub, Inc. (2021) to automated security analysis and automation Nunez et al. (2024). LLMs show strong aptitude for understanding complex codebases and flagging potential issues Divakaran & Peddinti (2024). This momentum naturally extends to vulnerability detection, spanning model-level approaches such as fine-tuning LLMs for classification—and agentic frameworks that harness LLM reasoning for automated analysis tmylla and contributors (2024).

Yet today's LLM-based detection methods face practical barriers. Most operate at the function level, asking whether a fragment is vulnerable rather than pinpointing where the flaw lies Zhang et al. (2024b), Sovrano et al. (2025). These tasks are also defined purely on static code, ignoring binaries, runtime behavior, and project-level diffs, further limiting their usefulness for real remedi-

ation. Evaluations also rely heavily on lightweight, synthetic datasets that miss the complexity of production systems Guo et al. (2024).

This research–practice gap is most visible in vulnerability localization. While studies show promising results on isolated functions or snippets, engineers must navigate large repositories with cross-module dependencies and need line-level localization to craft minimal, targeted patches precision current methods rarely deliver. Moreover, artificial benchmarks obscure the true difficulty of project-level detection, where context spans multiple files and demands system-wide reasoning.

Given the surge in vulnerability reports outlined earlier, the field urgently needs approaches that close this reality gap: leveraging LLM automation while tackling project-scale challenges practitioners face daily. Only by embracing these real-world constraints can we convert the promise of LLM-assisted security into tools that meaningfully lighten the load on development teams.

To address these gaps, we introduce T2L (Trace-to-Line), which reframes vulnerability detection as a two-tier problem (a) coarse-grained detection - flagging suspicious code chunks and (b) fine-grained localization - pinpointing exact vulnerable lines. This separation enables systematic evaluation of LLM capabilities from repository-scale reasoning to human-expert precision.

**Contribution.** This work makes three contributions: **(1)** *New Task formulation.* We define a new task for agentic AI, a runtime trace guided, project-level vulnerability detection, and formulate it as two structured subtasks: chunk-level detection and line-level localization. AST-based chunking adapts large codebases to LLM context while preserving semantics, enabling refinement from coarse predictions to exact lines, bridging research setups and real-world needs for LLMs. **(2)** *T2L-ARVO benchmark.* We present the first benchmark for agentic fine-grained localization, featuring 50 expert verified cases across five vulnerability types with balanced category distribution. `T2L-ARVO` enables realistic, project-scale evaluation of LLM-based systems. **(3)** *T2L-Agent framework.* We propose a multi-agent system with the following innovation: (i) a Trace Analyzer (Agentic Trace Analyzer) integrating the ensemble of tools such as static analysis, sanitizers, and runtime monitoring for observability as human debuggers; (ii) a Proposal module (Divergence Tracing) that iteratively forms and tests vulnerability hypotheses; and (iii) a two-stage pipeline (Detection Refinement) that first detects coarse chunks and then verifies exact lines with runtime feedback.

Overall, `T2L-Agent` provides a comprehensive baseline for this new vulnerability localization task. Unlike prior snippet-level detectors, it is designed to operate on fully reproducible real-world projects, using crashes, sanitizer reports, and project diffs as runtime evidence. On `T2L-ARVO`, it achieves up to **58%** chunk-level detection and **54.8%** exact line localization, showing that this runtime-evidence–guided setting is challenging yet tractable for LLM agents.

## 2 BACKGROUND

**Vulnerability Localization.** Classical localization combines static and dynamic analyses with retrieval and graph methods. Static slicing selects statements that may affect a slicing criterion to form behavior-preserving slices for debugging and comprehension Weiser (1981). Dynamic slicing refines this by deriving input-specific slices from execution-time dependence graphs, yielding higher precision near crashes or variables Agrawal & Horgan (1990). Information-retrieval approaches treat bug reports as queries, ranking files/methods by textual and historical signals to surface likely locations Zhou et al. (2012); Saha et al. (2013); Wang & Lo (2014). Code Property Graphs unify classic program-analysis concepts in a joint representation, enabling scalable pattern searches that uncovered 18 previously unknown vulnerabilities Yamaguchi et al. (2014).

**AI for Cybersecurity.** AI is widely applied to cybersecurity tasks with tool execution and feedback in runnable environments Yang et al. (2023). One prominent use is offensive security Shao et al. (2024b), where agents reason about attack chains and interact with benchmark environments Shao et al. (2024c); Zhang et al. (2024a) to simulate human CTF players Udeshi et al. (2025); Shao et al. (2025); Abramovich et al. (2025). In parallel, pen-testing systems orchestrate modules or multi-agent collaboration to automate reconnaissance, vulnerability analysis, and exploitation against targets Deng et al. (2024); Shen et al. (2025). Further work builds agent-based patching frameworks that localize vulnerabilities and synthesize fixes Yu et al. (2025); Xue et al. (2025), and explores LLM-augmented red and blue teaming, where agents emulate adaptive attackers and assist defenders in threat hunting and incident investigation Abuadbba et al. (2025); Liu et al. (2025).

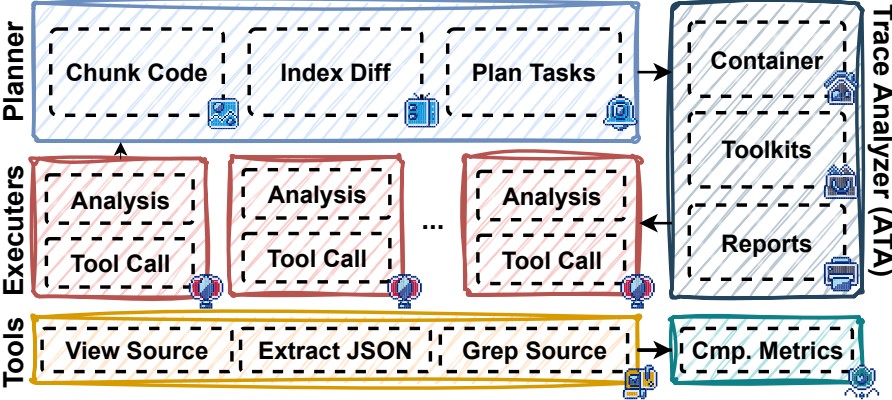

Figure 1: `T2L-Agent` Framework overview.

**LLM Agentic Systems.** Beyond cybersecurity, large language models (LLMs) are increasingly applied to automate workflows across diverse domains Shao et al. (2024a), including AI for scientific research Bran et al. (2023); Basit et al. (2025); Jin et al. (2024), software engineering Yang et al. (2024c), healthcare Isern & Moreno (2016), and hardware design Wang et al. (2024). Many systems integrate long-term memory to retain intermediate results, environment states, and user preferences across sessions Zhang et al. (2025b). LLM as judge are often used to evaluate actions, select among tool outputs, and provide feedback for self-improvement Li et al. (2025). Some agents employ self-reflection and critique loops to iteratively refine their reasoning or code before execution Renze & Guven (2024). Multi-agent and role-specialized architectures further enable collaboration among agents with distinct skills to solve complex, interleaved subtasks in parallel Li et al. (2024a).

## 3 RELATED WORK

LLMs show strong potential on vulnerability localization tasks. Early learning-based detectors improve localization by training models to classify whether a unit is vulnerable. LLMAOYang et al. (2024a) fine-tunes LLMs on small, manually curated buggy programs, while BAPStein et al. (2025) learns state-of-the-art vulnerability localization without any direct localization labels, outperforming traditional baseline over eight benchmarks. However, subsequent analyses revealed data issues in widely used datasets such as Big-Vul and DevignZhou et al. (2019), making people doubt about performance numbers and highlighting the need for realistic evaluation settings Croft et al. (2023).

Many prior works frame vulnerability localization at the file or function level, but this coarse granularity often fails to offers limited guidance for developers. GenLoc Asad et al. (2025) identifies potentially vulnerable files from bug reports and iteratively analyzes them using code exploration tools. AgentFL Qin et al. (2024) applies a multi-agent framework for function-level localization, modeling the task as a three-step pipeline with specialized agents and tools. CoSIL Jiang et al. (2025) narrows the function-level search space using module call graphs and iteratively traverses them for relevant context. Similarly, AutoFL Kang et al. (2024) prompts LLMs to localize method-level vulnerabilities via function-call navigation, showing that multi-step reasoning mitigates context limits.

| Study | Line Lv. | Mult. Ag. | Runtime | Iterative |
|---|---|---|---|---|
| LLMAO 2024a | ✓ | ✗ | ✗ | ✗ |
| BAP 2025 | ✓ | ✗ | ✗ | ✗ |
| GenLoc 2025 | ✗ | ✗ | ✓ | ✓ |
| AgentFL 2024 | ✗ | ✓ | ✓ | ✓ |
| CoSIL 2025 | ✗ | ✗ | ✗ | ✓ |
| AutoFL 2024 | ✗ | ✓ | ✓ | ✓ |
| LineVul 2022 | ✓ | ✗ | ✗ | ✗ |
| LOVA 2024b | ✓ | ✗ | ✗ | ✗ |
| MatsVD 2024 | ✗ | ✗ | ✗ | ✗ |
| xLoc 2024b | ✓ | ✗ | ✗ | ✗ |
| LLM4FL 2024 | ✗ | ✓ | ✓ | ✓ |
| MemFL 2025 | ✗ | ✗ | ✓ | ✓ |
| **T2L (ours)** | ✓ | ✓ | ✓ | ✓ |

Figure 2: Related Works

Recent studies have shifted toward line or statement level localization. LineVul Fu & Tantithamthavorn (2022) uses a Transformer-based classifier for line-level prediction, while LOVA Li et al. (2024b) introduces a self-attention framework to score and highlight vulnerable lines. MatsVD Weng et al. (2024) enhances statement level localization with dependency-aware attention, and xLoc Yang et al. (2024b) learns multilingual, task-specific knowledge for bug detection and localization. Building on these efforts, LLM4FL Rafi et al. (2024) proposes a multi-agent framework leveraging graph-based retrieval and navigation to reason about failure causes. MemFL Yeo et al. (2025) introduces external memory to incorporate project-specific knowledge, improving localization in complex, repository-scale systems. CThey push localization to line level and use more multi-agent coordination. However, most still rely on limited runtime evidence, single-pass predictions, or benchmarks that lack realistic project settings.

T2L formalizes a new task of runtime-trace-guided vulnerability localization at the project level. Prior learning-based approaches such as BAP, LLMAO, and LineVul rely on static code analysis, training models to classify whether a given code snippet is vulnerable based solely on syntactic and semantic patterns. These methods typically operate on isolated single files and produce coarse-grained predictions at the function or file level without leveraging dynamic execution information. In contrast, our task targets multi-file codebases and incorporates runtime evidence including reproducible crash logs, stack traces, and sanitizer reports to pinpoint exact vulnerable lines. This formulation more closely reflects real-world vulnerability triage workflows where developers rely on both static code inspection and dynamic program behavior to diagnose security flaws.

## 4    METHODS

### 4.1    T2L AGENT

`T2L-Agent` uses a hierarchical planner–executor design Udeshi et al. (2025) that splits localization into evidence gathering, hypothesis generation, and iterative refinement.Unlike single-pass analyzers, it uses a human-like process: gather runtime signals, link them to code, and narrow the search space. Details of the tool list that T2L originally supports are provided in A.5.

**Evidence Tracing** T2L Planner coordinates repository analysis and runtime evidence capture in a single, structured pipeline.   First, code-structure analysis partitions the codebase into function-aligned, semantically coherent units that preserve syntactic relationships while fitting LLM context windows; known patch locations are also indexed to establish evaluation baselines. Next, runtime evidence becomes the cornerstone: Sanitizer records memory-violation patterns, allocation traces, and stack frames, while interactive debugging (GDB/LLDB) provides symbolic context via backtraces and variable-state snapshots at crash points.  This dual-layer design yields comprehensive, observable crash logs rather than speculative static signals. We collect multiple types of runtime and static evidence, including crash logs, sanitizer reports, stack traces, and static analyzer results, all evidence is merged into a single unified textual block and fed to

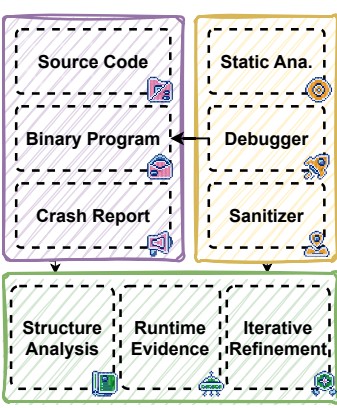

Figure 3: ATA Components

the LLM at once. This design ensures that the LLM forms its reasoning based on the global evidence context, rather than aggregating per-evidence predictions to avoids issues of cross-signal inconsistency. The integrated workflow is a key T2L innovation; we detail it in Sec. 4.1.

**Hypothesis Generation** A "hypothesis" an LLM-generated candidate vulnerability produced from the unified evidence block of an existing vulnerable project in T2L system. The LLM proposes a list of `file:line` candidate pairs ranked by confidence. Each hypothesis is independently checked against the ground-truth patch diff, and candidates do not affect one another during evaluation.

**Two-stage Refinement** To ensure accurate detection, T2L uses a two-stage refinement process.  It begins with `coarse-grained filtering`: the system derives broad candidate regions from crash logs, using LLM code comprehension to link symptoms to likely causes and produce a ranked list of candidates with confidence and rationale. It then identifies code fragments the LLM flagged as relevant in this first pass–such as missing bounds checks, uninitialized variables, or improper memory management.  Next comes `fine-grained refinement`: these extracted fragments are added to the evidence for a second LLM pass, which updates the candidate set.  Any newly generated locations are merged into the existing candidates.  This refinement loop continues until the LLM produces no additional candidates or the cost budget is reached.

**Feedback Control** Each cycle outputs a brief task summary with success indicators and confidence. The Planner adapts the next step—continue refining or stop, preventing premature termination and avoiding over-analysis. The loop improves precision while controlling compute cost.

**Agentic Trace Analyzer (ATA)** The Agentic Trace Analyzer (ATA) is the core module of `T2L-Agent`.  Its purpose is to connect runtime behavior with relevant source code regions, addressing a gap in prior work that relies mainly on static signals. ATA constructs a unified "evidence block" by running targets in reproducible Docker, partitioning repositories into semantically meaningful chunks using tree-sitter, and instrumenting executions with analysis toolkit such as Sanitizers,

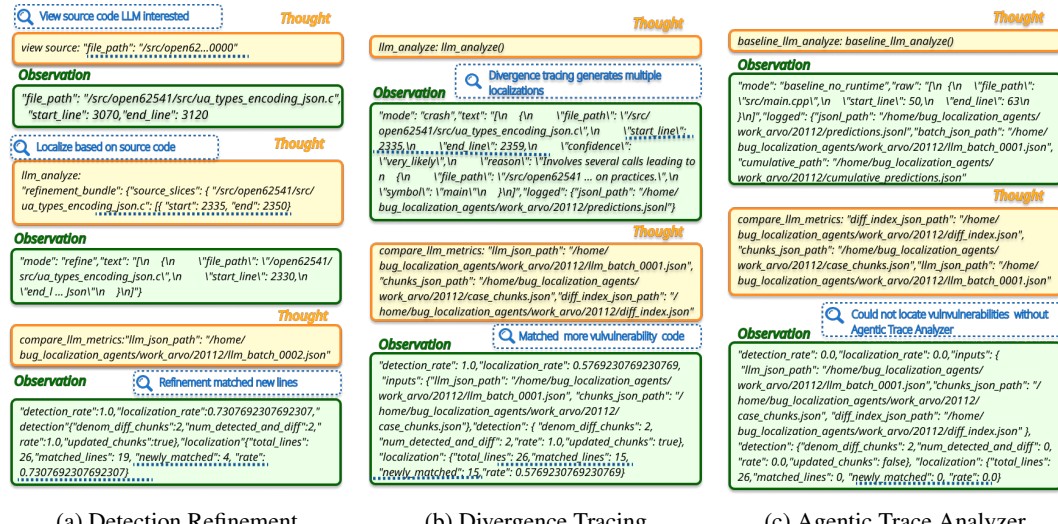

(a) Detection Refinement     (b) Divergence Tracing     (c) Agentic Trace Analyzer

Figure 4: Partial `T2L-Agent` logs to show the how the three proposed technique on `T2L-Agent` work and help the task: Detection Refinement, Divergence Tracing and Agentic Trace Analyzer.

Debuggers, and Static analyzers to collect stack traces, memory errors, and other runtime cues. The output is a unified "evidence block" which is then passed to `T2L-Agent`'s reasoning modules.

We use hierarchical refinement, starting from crash logs and narrowing to specific code. The agent cross-validates runtime signals with static features and ranks candidates by LLM confidence. Like human debugging, `T2L-Agent` seeds candidates via static–dynamic correlation and iteratively refines them with source–feedback loops. ATA brings fewer single-shot failures, improved compute efficiency, and behavior-anchored decisions that enable precise, line-level localization even in large, tightly coupled codebases. With ATA disabled, the LLM localizes on its own, as shown in Figure 4(c). Without ATA, the LLM fails to localize vulnerabilities compared with (a) and (b).

## 4.2 T2L FEATURES

**Divergence Tracing.** Recognizing that complex vulnerabilities often involves multiple files and functions, `T2L-Agent` also uses divergence tracing to explore multiple hypotheses in parallel from the same crash log. This feature is inspired by how modern LLM interfaces offer multiple response variations for users, using the LLM's variability to ensure comprehensive coverage of potential vulnerability locations. Rather than committing to a single chain of thought, it expands several in parallel and returns a ranked list of candidate sites across the search space. This surfaces correct localizations that were not top ranked initially and is especially helpful for bugs that span multiple modules. From Figure 4 (b), we can observe that the divergence tracing generates more localization candidates and matched more vulnerable lines in this round.

**Detection Refinement.** The detection refinement begins based on crash logs and initial localization candidates. Each LLM-generated candidate gets a confidence label from `very unlikely` to `very likely`. We use these labels to sort and rank the candidates during the `coarse-grained filtering`. T2L agent selects code slices based on crash logs, stack trace information, and vulnerability patterns instead of exhaustively examining the full code snippet. In `fine-grained refinement`, we extract these slices and feed them back to the LLM along with the unified evidence, providing richer context surrounding the initially flagged locations.

The refinement process operates iteratively to help the agent correct early mistakes and discover vulnerabilities that are not immediately obvious from traces, particularly for complex vulnerabilities involving memory corruption where the crash point may be far away from the actual vulnerability. As Figure 4 (a) shows, based on the source code LLM interested, the refinement process successfully locates new lines that were not found during first step with only the runtime evidence.

## 4.3 T2L-ARVO Benchmark

The original `ARVO` dataset contains over 4,993 reproducible vulnerabilities across 250+ C/C++ projects, but its human-oriented design and build-centric structure do not directly support evaluating *agentic* vulnerability-localization systems. To bridge this gap, we developed `T2L-ARVO`, a 50-case benchmark derived from `ARVO` for assessing LLM agents in evidence-guided vulnerability localization, rather than human bug reproduction. `T2L-ARVO` maintains a moderate, calibrated difficulty while keeping the original crash-type diversity. Its construction employs a dual validation pipeline combining manual expert review with LLM-assisted checks to ensure each case is reproducible and appropriately challenging for automated agents. This yields a realistic, balanced benchmark for rigorous end-to-end evaluation of LLM-based vulnerability localization. In `T2L-ARVO`, each of the cases has multiple ground-truth bug lines. The average accuracy overall is denoted as *Accuracy = correct localized vulnerable lines/total vulnerable lines*.

**T2L-ARVO Composition** We analyzed 4,993 ARVO instances and grouped them by underlying failure mechanism: *Buffer Overflows* $49.9\%$ ($n = 2,490$), *Uninitialized Access & Unknown States* $35.4\%$ ($n = 1,768$), *Memory Lifecycle Errors* $11.5\%$ ($n = 573$), *Type Safety & Parameter Validation* $2.9\%$ ($n = 147$), and *System & Runtime Errors* $0.3\%$ ($n = 15$). Each family subsumes concrete subtypes (e.g., `heap-buffer-overflow`, `use-of-uninitialized-value`, `heap-use-after-free`, `bad-cast`). `T2L-ARVO` deliberately mirrors this distribution to avoid bias toward any single failure mode.

The final benchmark comprises 50 vulnerabilities samples, evenly sampled across five crash families (10 each) for broad yet controlled difficulty. Each family includes representative subtypes (e.g., `heap-buffer-overflow`, `use-of-uninitialized-value`, `heap-use-after-free`, `bad-cast`), covering single-file defects and cross-module interactions to prevent overflow bias and exercise diverse failure modes observed in real repositories.

**Verification Process** Our verification follows two main considerations. First, ARVO crash families contain multiple subtypes. To reduce redundancy and improve representativeness, we select only a few typical subtypes from each family (see A.1), preserving diversity without overloading the benchmark with

Figure 5: Crash types in `T2L-ARVO` Bench.

| Crash Family | Brief Description |
|---|---|
| Buffer Overflow | Violations of memory bounds (heap/stack) |
| Uninitialized Access | Reads from undefined or indeterminate state |
| Memory Lifecycle | Use-after-free / double-free / lifetime bugs |
| Type Safety | Bad casts, invalid args, contract violations |
| System Runtime | Environment and runtime interaction faults |

similar vulnerabilities. Second, experts evaluate each sample's complexity to keep `T2L-ARVO` at a moderate, calibrated difficulty. We score candidates using diff-based structural metrics (e.g., files changed, architectural spread, directory depth) and semantic factors (e.g., cross-module coupling, interface changes), ensuring that `T2L-ARVO` remains balanced in both coverage and difficulty.

## 5 Experiment Setup

**Metrics.** We report two complementary scores for project-level OSS vulnerability studies. *Detection* asks whether the agent flags a vulnerability within the correct module/chunk and materially shrinks the search space. *Localization* requires exact line matches to ground-truth patches. Together, they separate "finding the neighborhood" from "pinpointing the line", mirroring real debugging.

**Data Preparation.** We evaluate on the full `T2L-ARVO` set: 50 verified, structured challenges derived from `ARVO`, spanning diverse domains (e.g., imaging, networking) and balanced complexity so results reflect production patterns rather than a single project or bug family. Because ARVO lacks detection-ready chunking, `T2L-ARVO` adds AST-based segmentation: projects are partitioned into semantically meaningful units for scoring coarse *detection*, while exact line matches assess fine *localization*—a single framework for both levels.

**Model Selection.** We assess a set of state-of-the-art language models—both open-source and commercial to probe generality and robustness of `T2L-Agent` across architectures and scales, including open models such as Qwen3 Next, Qwen3 235B, DeepSeek 3.1, LLaMA 4 and commercial models like Claude4 Sonnet, GPT-5, GPT-4.1, GPT-4o-mini, Gemini 2.5 Pro, Gemini 2.5 Flash with a maximum budget $1.0. We use API keys from commercial model's official providers and Together.ai's inference service for open source models.

Table 1: Base `T2L-Agent` Localization and Detection Rate Performance Across Different Models.

| | % Avg. | | Buffer | | Initialize | | Memory | | Parameter | | Runtime | |
|---|---|---|---|---|---|---|---|---|---|---|---|---|
| | Det | Loc | Det | Loc | Det | Loc | Det | Loc | Det | Loc | Det | Loc |
| GPT-5 | 44.3 | **41.7** | 57.5 | **53.8** | 35.6 | 35.5 | **60.8** | **55.9** | 36.5 | 39.4 | 11.2 | 10.0 |
| GPT-4.1 | **48.0** | 38.5 | **60.8** | 36.5 | 50.6 | **46.4** | **60.8** | 46.1 | 26.5 | 29.9 | 21.2 | **20.5** |
| GPT-4o-mini | 44.3 | 22.6 | **60.8** | 20.2 | 48.1 | 12.5 | 55.8 | 24.7 | 28.7 | 26.7 | 11.2 | 10.0 |
| Claude 4 Sonnet | 45.9 | 30.5 | 57.5 | 50.6 | **60.8** | 36.1 | 1.3 | 31.6 | **37.8** | **46.1** | **24.8** | 0.5 |
| Gemini2.5 Pro | 17.4 | 10.5 | 25.0 | 5.6 | 25.0 | 20.0 | 11.3 | 10.8 | 5.4 | 11.9 | 10.0 | 10.5 |
| Qwen3 235B | 25.9 | 9.2 | 25.8 | 6.5 | 23.1 | 1.7 | 40.8 | 16.7 | 28.7 | 17.3 | 7.9 | 0.0 |
| Qwen3 Next 80B | 37.4 | 5.9 | 54.2 | 3.7 | 33.1 | 0.4 | 55.8 | 14.5 | 29.1 | 6.8 | 1.2 | 0.0 |

**Implementation.** We build `T2L-Agent` from scratch without LangChain, DSPy, and LlamaIndex to keep the core lightweight, retain fine-grained control over reasoning and tools, and maximize extensibility. The *Agentic Trace Analyzer* compiles targets with ASAN and collects crashes, stack traces, and allocation metadata to yield actionable traces for narrowing. Following ARVO's layout, we maintain registry-backed, per-project environments and provide both vulnerable and patched revisions in containers. Our harness runs dockerized `T2L-Agent`'s that interface with `T2L-ARVO` via the Docker SDK for Python, orchestrating build–run–reproduce cycles from within the agent loop, ensuring consistent conditions, deterministic reproduction, and auditable measurement.

# 6 EVALUATION

## 6.1 BASELINE BENCHMARKING

We evaluate `T2L-Agent` on `T2L-ARVO` end-to-end and report Detection Rate and Localization Rate. Table 1 covers five models—GPT-5, GPT-4.1, GPT-4o-mini, Qwen 3 Next 80B, and Qwen3 235B, running under identical per-case budgets, environments, and AST-based chunking. Overall, detection is higher than localization by design. Under this setting, GPT-5 leads localization at 41.7%, followed by GPT-4.1 at 38.5%; GPT-4o-mini lands at 22.6%, and open-source models trail Qwen 3 235B 9.2% and Qwen 3 Next 80B 5.9%. For detection, GPT-4.1 is highest at 48.0%, with GPT-5 and GPT-4o-mini both at 44.3%; Qwen3 reaches 37.4% and Qwen3 235B 25.9%. While Gemini2.5 Pro shows limited effectiveness with 17.4% detection and 10.5% localization rates.

Family-wise patterns are consistent across metrics. *Buffer* and *Memory* are easier due to concrete runtime cues: for localization, GPT-5 reaches 53.8% and 55.9%, and most models cluster near the mid-50s for detection. *Initialize* sits mid-range and benefits from multi-step reasoning (e.g., GPT-5 35.5% vs. GPT-4.1 46.4% in localization). *Parameter* is often solvable from interface/call-site context (39.4% for GPT-5; 29.9% for GPT-4.1). *Runtime* remains uniformly hardest: detection hovers around 11.1–21.2% even for top configurations, and the best localization we observe is 20.5% on GPT-4.1, reflecting sparse, unstable traces.

Taken together, equal budgets surface clear profiles. GPT-5 (low-think) converts evidence into the strongest line-level localization, while GPT-4.1 extracts slightly more coarse-grained signal at the chunk level. GPT-4o-mini's mix—competitive detection (44.3%) but weak localization (22.6%), suggests higher recall with looser ranking that does not always translate to precise line hits. Open-source models lag on both metrics under the same constraints, indicating gaps in code understanding and tool use rather than simple parameter tuning. Overall, improvements track the availability of concrete runtime evidence, and structured, tool-grounded reasoning appears more impactful than generation settings for end-to-end vulnerability localization.

## 6.2 DISCUSSION 1: FEATURE-WISE EVALUATION

**Agentic Trace Analyzer.** This table 2 demonstrates the critical effectiveness of our proposed Agentic Trace Analyzer (ATA) through ablation experiments. Without ATA, GPT-5 and Claude 4 Sonnet show 0.0% detection and localization across all families. This performance breakdown validates that our ATA component successfully bridges the gap between crash symptoms and vulnerability locations, addresses the fundamental challenge of vulnerability localization in complex codebases.

**Detection Refinement.** Compared with Tab 1, Tab. 2 shows broad, across-the-board gains after enabling refinement. Strong proprietary models improve steadily, while open-source models jump the most—Qwen3 235B's localization rises by roughly sevenfold. Improvements vary by crash family: *Initialize* bugs benefit most (they demand multi-step reasoning), whereas *Buffer* and *Memory* see smaller lifts because concrete runtime evidence already anchors the search. *Runtime* cases remain hard—when traces are sparse, refinement offers limited benefit. Net effect: higher recall and more

Table 2: Localization and Detection Rate Performance of `T2L-Agent` with Refinement and Divergence Tracing Across Different Models.

| | % Avg. | | | | Buffer | | Initialize | | Memory | | Parameter | | Runtime | |
|---|---|---|---|---|---|---|---|---|---|---|---|---|---|---|
| | Det | Loc | ΔDet | ΔLoc | Det | Loc | Det | Loc | Det | Loc | Det | Loc | Det | Loc |
| **w/ Detection Refinement** | | | | | | | | | | | | | | |
| GPT-5 | 52.4 | 44.5 | +8.1↑ | +2.8↑ | 57.5 | 55.0 | 55.6 | 43.1 | 60.8 | 41.3 | 43.5 | 48.2 | 21.2 | 20.5 |
| GPT-4.1 | 48.3 | 40.8 | +0.3↑ | +2.3↑ | 60.8 | 51.9 | 53.1 | 44.9 | 57.5 | 46.1 | 39.8 | 42.9 | 6.7 | 0.0 |
| GPT-4o-mini | 34.6 | 29.1 | -9.7↓ | +6.5↑ | 45.8 | 43.6 | 30.6 | 20.2 | 45.8 | 33.6 | 22.4 | 21.2 | 0.0 | 0.0 |
| Claude 4 Sonnet | 44.8 | 41.4 | -1.1↓ | +10.9↑ | 57.5 | 54.3 | 40.6 | 43.2 | 61.7 | 52.5 | 26.1 | 29.4 | 14.6 | 10.5 |
| Gemini2.5 Pro | 14.1 | 11.4 | -3.3↓ | -0.9↓ | 10.0 | 8.6 | 20.0 | 16.2 | 40.0 | 32.1 | 0.4 | 0.3 | 0.0 | 0.0 |
| Qwen3 Next 80B | 42.9 | 39.5 | +5.5↑ | +33.6↑ | 60.8 | 55.1 | 50.6 | 44.4 | 60.8 | 48.6 | 25.7 | 29.2 | 0.0 | 0.0 |
| Qwen3 235B | 34.1 | 26.7 | +8.2↑ | +17.5↑ | 34.2 | 32.0 | 30.6 | 34.5 | 57.5 | 38.8 | 23.3 | 13.3 | 5.0 | 8.2 |
| Gemini 2.5 Flash | 22.5 | 18.4 | – | – | 34.2 | 0.6 | 40.8 | 25.7 | 7.9 | 27.6 | 0.4 | 33.4 | 24.6 | 0.6 |
| Llama4 | 28.3 | 28.1 | – | – | 30.8 | 25.6 | 35.8 | 26.1 | 0.0 | 27.9 | 24.1 | 32.0 | 30.2 | 0.0 |
| Deepseek V3.1 | 53.9 | 53.4 | – | – | 60.8 | 55.6 | **62.5** | 47.5 | 16.3 | 58.1 | 55.0 | **60.2** | 47.6 | 19.4 |
| **w/ Divergence Tracing** | | | | | | | | | | | | | | |
| GPT-5 | **58.0** | 52.0 | +13.7↑ | +10.3↑ | 60.8 | 56.8 | 60.6 | 53.4 | **62.5** | 47.6 | 53.2 | 46.7 | 26.2 | **28.7** |
| GPT-4.1 | 52.0 | 49.9 | +4.0↑ | +11.4↑ | 60.8 | 53.5 | 60.6 | **57.3** | 57.5 | 53.0 | 43.2 | 37.1 | 21.2 | 20.1 |
| GPT-4o-mini | 47.2 | 43.3 | +2.9↑ | +20.7↑ | **64.2** | 55.6 | 55.6 | 48.8 | 52.5 | 47.0 | 33.9 | 32.4 | 1.2 | 0.5 |
| Claude 4 Sonnet | 48.7 | 49.8 | +2.8↑ | +19.3↑ | 60.8 | 55.6 | **62.5** | 39.8 | 11.3 | 57.5 | 53.7 | 57.6 | 46.3 | 10.6 |
| Qwen 3 Next 80B | 51.2 | **54.8** | +13.8↑ | +48.9↑ | **64.2** | **58.1** | **62.5** | 43.2 | 11.3 | **63.2** | **58.6** | 57.7 | **48.8** | 21.2 |
| Qwen 3 235B | 42.7 | 42.1 | +16.8↑ | +32.9↑ | 50.8 | 50.6 | 47.5 | 40.2 | 1.0 | 45.4 | 46.9 | 53.5 | 33.7 | 11.2 |
| **w/o Agentic Trace Analyzer** | | | | | | | | | | | | | | |
| GPT-5 | 0.0 | 0.0 | -44.3↓ | -41.7↓ | 0.0 | 0.0 | 0.0 | 0.0 | 0.0 | 0.0 | 0.0 | 0.0 | 0.0 | 0.0 |
| Claude 4 Sonnet | 0.0 | 0.0 | -45.9↓ | -30.5↓ | 0.0 | 0.0 | 0.0 | 0.0 | 0.0 | 0.0 | 0.0 | 0.0 | 0.0 | 0.0 |

precise line-level hits with minimal tuning. Several additional models show promising performance. Deepseek V3.1 achieves the highest overall results with 53.9% detection and 53.4% localization rate. LLaMa 4 demonstrates balanced capabilities on both metrics, and Gemini 2.5 Flash shows variable performance across crash families.

**Divergence Tracing.** Tab. 2 shows divergence tracing yields the strongest gains. All models improve on both metrics: GPT-5 is up 13.7% in detection and 10.3 in localization; GPT-4.1 gains 4.0% and 11.4% respectively. Qwen3 Next see the largest jumps with adding 13.8% in detection and 48.9% in localization, while Qwen3 235B adds 16.8% and 32.9%. These consistent lifts across architectures highlight divergence tracing as a core algorithmic upgrade for vulnerability localization.

## 6.3 DISCUSSION 2: PARAMETER TUNING

**Thinking Budget.** As shown in Tab. 3, more thinking didn't help. On GPT-5, the Medium budget outperforms High with 50.9% detection vs 41.3%, and 41.6% localization vs 36.1%. The Low setting trails Medium by a few points yet often matches or even exceeds High on key metrics, while sharply reducing compute and latency. This pattern suggests diminishing returns—and decision drag—at very high budgets: the model over-explores, delays commitment, and accumulates tool-use errors. In practice, Medium strikes the best accuracy–cost balance for vulnerability localization; Low is a strong option when throughput and responsiveness matter most.

Table 3: Localization and Detection Rate Performance of `T2L-Agent` for Temperature Tuning and Chain of Thought Across Different Models.

| | | % Avg. | | Buffer | | Initialize | | Memory | | Parameter | | Runtime | |
|---|---|---|---|---|---|---|---|---|---|---|---|---|---|
| | Config | Det | Loc | Det | Loc | Det | Loc | Det | Loc | Det | Loc | Det | Loc |
| **Temperature** | | | | | | | | | | | | | |
| GPT-4.1 | 0.2 | **51.0** | 43.6 | 57.5 | 44.4 | 55.6 | **51.1** | 55.8 | 45.7 | 43.2 | 35.4 | 21.2 | **20.2** |
| GPT-4.1 | 0.6 | 50.8 | 43.5 | **60.8** | 52.0 | 53.1 | 47.8 | 55.8 | 44.8 | 39.4 | **42.7** | 17.9 | 10.1 |
| Claude 4 Sonnet | 0.2 | 46.5 | 43.0 | 54.2 | **56.1** | 55.6 | 43.0 | **61.7** | 49.0 | 36.5 | 39.7 | 11.2 | 10.9 |
| Claude 4 Sonnet | 0.6 | 47.3 | **44.9** | 54.2 | **56.1** | 50.6 | 48.0 | 60.8 | **53.6** | 36.5 | 39.6 | 11.2 | 10.9 |
| **Reasoning Effect** | | | | | | | | | | | | | |
| GPT-5 | High | 41.3 | 36.1 | 54.2 | 47.8 | 40.6 | 32.4 | 55.8 | 39.3 | 32.8 | 36.7 | 10.0 | 10.0 |
| GPT-5 | Medium | 50.9 | 41.6 | **60.8** | 55.6 | **60.8** | 42.8 | 11.3 | 46.1 | **45.8** | 40.4 | **47.4** | 10.5 |

**Temperature.** Temperature changes barely matter from Tab. 3. On GPT-4.1, detection is 51.0% at 0.2 and 50.8% at 0.6, with localization 43.6% vs. 43.5%. Claude 4 Sonnet shows the same pattern: 46.5% vs. 47.3% detection and 43.0% vs. 44.9% localization. Performance is stable in 0.2-0.6 range. The exception is *Initialize* bugs, which are more temperature sensitive than *Buffer* and *Memory* cases that lean on concrete runtime evidence. Overall, precise localization benefits more from structured, tool-grounded reasoning than from extra sampling, making parameter choices simple.

## 6.4 DISCUSSION 3: CASE STUDY

Figure 6 illustrates a full `T2L-Agent` workflow on a real case from `T2L-ARVO`, showcasing the iterative planner–executor architecture in action. The process starts with the Planner orchestrating code chunking (`chunk_case`, 5441 chunks) and diff indexing (`diff_index`), then running sanitized execution (`run_san`) to collect crash logs before delegating reasoning to the Executor.

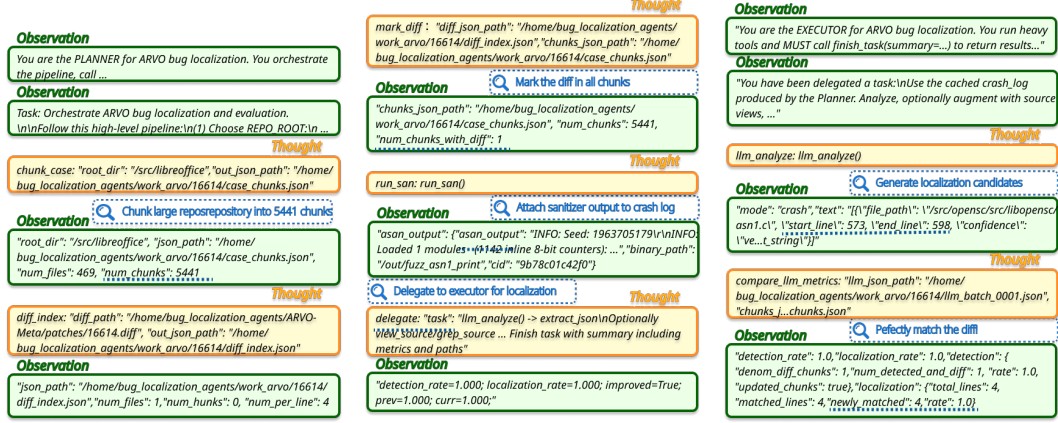

Figure 6: `T2L-Agent` pipeline visualization.

The Executor analyzes traces via `llm_analyze`, extracts ranked candidates (`extract_json`), and iteratively evaluates them against ground truth using `compare_llm_metrics`, combining static patterns and dynamic signals. This multi-round refinement achieves perfect localization (detection: 1.0, localization: 1.0), demonstrating `T2L-Agent`'s ability to convert crash symptoms into precise diagnostics. Each panel visualizes thought (function call) and observation (result) with hand-drawn borders and no hallucinated text. Together, they highlight data flow, role separation, and metric-driven validation across planner and executor components.

## 7 LIMITATION AND FUTURE WORK

Our work has three key limitations. First, the `T2L-ARVO` benchmark includes only 50 manually verified cases. While it offers broad vulnerability coverage and balanced categories, the limited sample size constrains evaluation due to the human verification efforts. Also, ARVO's dataset structure is designed for human developers, which needs more fine-grained metadata that could benefit LLM-based localization. Second, although `T2L-Agent` improves localization accuracy from 0% to 54.8% through three innovations, cost efficiency remains a concern. The agent operates effectively under a $1.0 budget via task-aware planning and early stopping, but large-scale deployment across thousands of vulnerabilities would demand significant optimization. Third, higher model thinking budgets fail to boost localization performance, indicating that increased compute alone is insufficient. This points to a need for smarter ways to exploit model reasoning. Future work should explore more efficient architectures. Such as model cascading to coordinate cheaper and stronger models, and specialized multi-agent systems where roles are tailored to tools like our Agentic Trace Analyzer. These strategies may retain quality while scaling to production workloads.

## 8 CONCLUSION

T2L addresses a key gap between LLM-based vulnerability localization and real-world practice. We contribute three advances that move from coarse identification to precise diagnostics. First, we propose a new formulation for LLM-based vulnerability detection: chunk-wise detection and line-level localization, enabling structured and fine-grained evaluation. Second, `T2L-ARVO` introduces a benchmark for agentic line-level localization, with 50 expert-verified cases across diverse vulnerabilities. Third, `T2L-Agent` improves performance via our Agentic Trace Analyzer, which fuses runtime and static signals, as well as Divergence Tracing and Detection Refinement in a feedback-driven workflow. `T2L-Agent` achieves 44–58% detection and 38–54.8% localization, marking a step toward deployable systems for real-world code security.

## ETHICS STATEMENT

This work builds upon the ARVO dataset for vulnerability analysis. Our T2L-ARVO benchmark is constructed using its full-version data, and we have properly cited the original ARVO project to acknowledge its contribution and comply with copyright and attribution standards. No additional data collection, user studies, or ethically sensitive procedures involved in the construction or evaluation of T2L-Agent. All experiments were conducted on Linux servers with only open source dependencies, and our system does not involve any privacy-sensitive data, bias-sensitive decision-making, or potentially harmful applications. We also note that large language models were used solely for light editing and polishing of manuscript, with no involvement in system design, code generation, or experimental results. Their use was limited to improving readability and presentation clarity.

## REPRODUCIBILITY STATEMENT

To ensure reproducibility, we will publicly release the full T2L-Agent framework upon publish of this paper, including all module codebase, evaluation scripts, and benchmark data used in this paper. Our implementation does not rely on any proprietary components. Detailed descriptions of our methodology are provided in the main text (Sections 4.1, 4.3) and supported by step-by-step examples in the appendix. All experimental configurations, including model versions, prompting strategies, and budget constraints, will be documented and made available upon publication to enable full replication of our results.

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

# A APPENDIX

## A.1 ARVO CRASH TYPE

To understand where our vulnerability localization agent should focus, we analyze the crash type distribution in ARVO dataset. ARVO's crash type distribution is dominated by classic memory corruption: Buffer Overflow accounts for 49.9% of crashes, led by heap-buffer-overflow (36.1% overall) and followed by stack-buffer-overflow (6.2%), index-out-of-bounds (3.3%), and global-buffer-overflow (3.2%), with underflows/containers in the long tail (around 1% each). Uninitialized Access & Unknown States is the second largest family at 35.4%, primarily use-of-uninitialized-value (20.3%), then UNKNOWN READ/WRITE (around 11.8% combined). Memory Lifecycle Errors contribute 11.5%, dominated by heap-use-after-free (7.8% overall) plus double-free, use-after-poison, and invalid frees. Type Safety & Parameter Validation is smaller (2.9%)—notably bad-cast (1.3%) and negative-size-param (0.8%). System & Runtime Errors are rare (0.3%). Overall, around 85% of ARVO crashes fall into Buffer Overflow or Uninitialized/Unknown categories.

Table 4: Crash Families and Subtypes Analysis

| Family | Subtype | Count | % within family | % of total |
|---|---|---|---|---|
| **Buffer Overflow Vulnerabilities** | | | | |
| | *Total* | *2490* | *—* | *49.9%* |
| | Heap-buffer-overflow | 1802 | 72.4% | 36.1% |
| | Stack-buffer-overflow | 308 | 12.4% | 6.2% |
| | Index-out-of-bounds | 165 | 6.6% | 3.3% |
| | Global-buffer-overflow | 160 | 6.4% | 3.2% |
| | Container-overflow | 33 | 1.3% | 0.7% |
| | Stack-buffer-underflow | 13 | 0.5% | 0.3% |
| | Dynamic-stack-buffer-overflow | 9 | 0.4% | 0.2% |
| **Uninitialized Access & Unknown States** | | | | |
| | *Total* | *1768* | *—* | *35.4%* |
| | Use-of-uninitialized-value | 1015 | 57.4% | 20.3% |
| | UNKNOWN READ | 462 | 26.1% | 9.3% |
| | Segv on unknown address | 134 | 7.6% | 2.7% |
| | UNKNOWN WRITE | 123 | 7.0% | 2.5% |
| | Null-dereference READ | 25 | 1.4% | 0.5% |
| | UNKNOWN | 8 | 0.5% | 0.2% |
| | Unknown-crash | 1 | 0.1% | 0.0% |
| **Memory Lifecycle Errors** | | | | |
| | *Total* | *573* | *—* | *11.5%* |
| | Heap-use-after-free | 389 | 67.9% | 7.8% |
| | Heap-double-free | 63 | 11.0% | 1.3% |
| | Use-after-poison | 48 | 8.4% | 1.0% |
| | Invalid-free | 29 | 5.1% | 0.6% |
| | Stack-use-after-return | 26 | 4.5% | 0.5% |
| | Stack-use-after-scope | 13 | 2.3% | 0.3% |
| | Bad-free | 5 | 0.9% | 0.1% |
| **Type Safety & Parameter Validation** | | | | |
| | *Total* | *147* | *—* | *2.9%* |
| | Bad-cast | 65 | 44.2% | 1.3% |
| | Negative-size-param | 42 | 28.6% | 0.8% |
| | Memcpy-param-overlap | 20 | 13.6% | 0.4% |
| | Object-size | 9 | 6.1% | 0.2% |
| | Incorrect-function-pointer-type | 6 | 4.1% | 0.1% |
| | Non-positive-vla-bound-value | 3 | 2.0% | 0.1% |
| | Strcpy-param-overlap | 1 | 0.7% | 0.0% |

Table 4: Crash Families and Subtypes Analysis (continued)

| Family | Subtype | Count | % within family | % of total |
|---|---|---|---|---|
| | Strncpy-param-overlap | 1 | 0.7% | 0.0% |
| **System & Runtime Errors** | | | | |
| | *Total* | *15* | *—* | *0.3%* |
| | Check failed | 6 | 40.0% | 0.1% |
| | Unknown signal | 6 | 40.0% | 0.1% |
| | Bad parameters to –sanitizer-annotate-contiguous-container | 2 | 13.3% | 0.0% |
| | Nested bug in the same thread, aborting. | 1 | 6.7% | 0.0% |

## A.2 ARVO DATASET PROFILING

We profiled the full ARVO corpus with all 4,993 vulnerabilities across 288 projects to guide `T2L-ARVO`'s design and document its coverage. The analysis maps distributional patterns, project traits, and crash-type frequencies, and clarifies how our 50-case subset aligns with the broader ARVO ecosystem. These profiles confirm that `T2L-ARVO` is representative across crash families, project complexity, and severity, providing a transparent baseline for extensions and alternative benchmarks. We include compact visualizations of these profiles to convey key ARVO factors—such as crash families, project complexity, and severity at a glance.

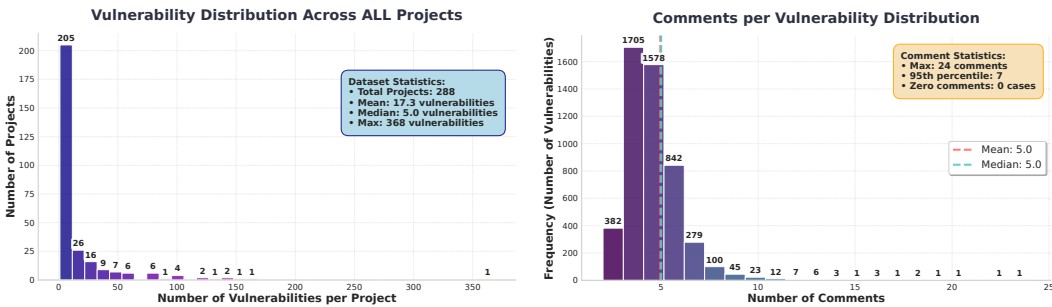

Figure 7: Distribution of vulnerability counts across the 288 ARVO projects. Most projects have under 25 vulnerabilities, with a long-tail of highly vulnerable ones.

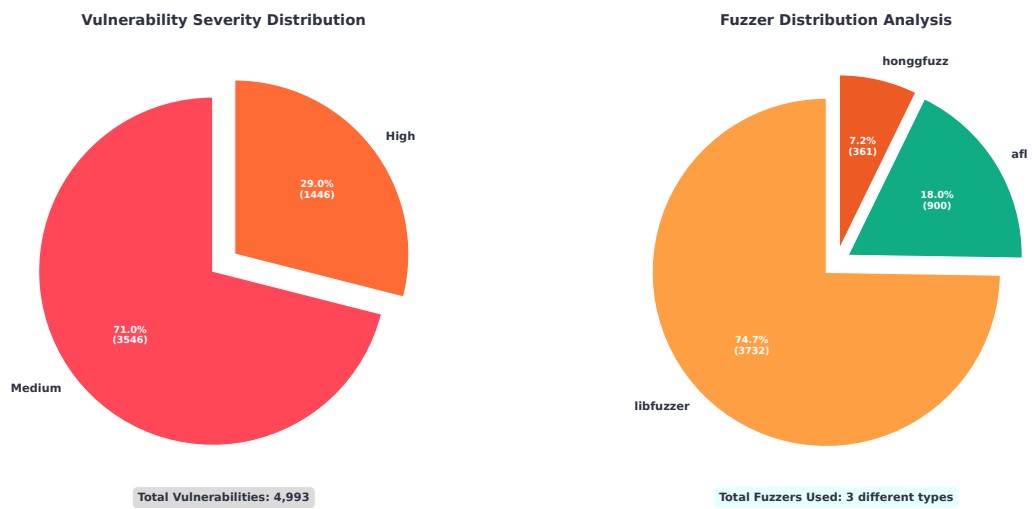

Figure 8: Breakdown of vulnerability severities in ARVO. Over 70% are medium severity, while high severity cases account for the remaining 29%.

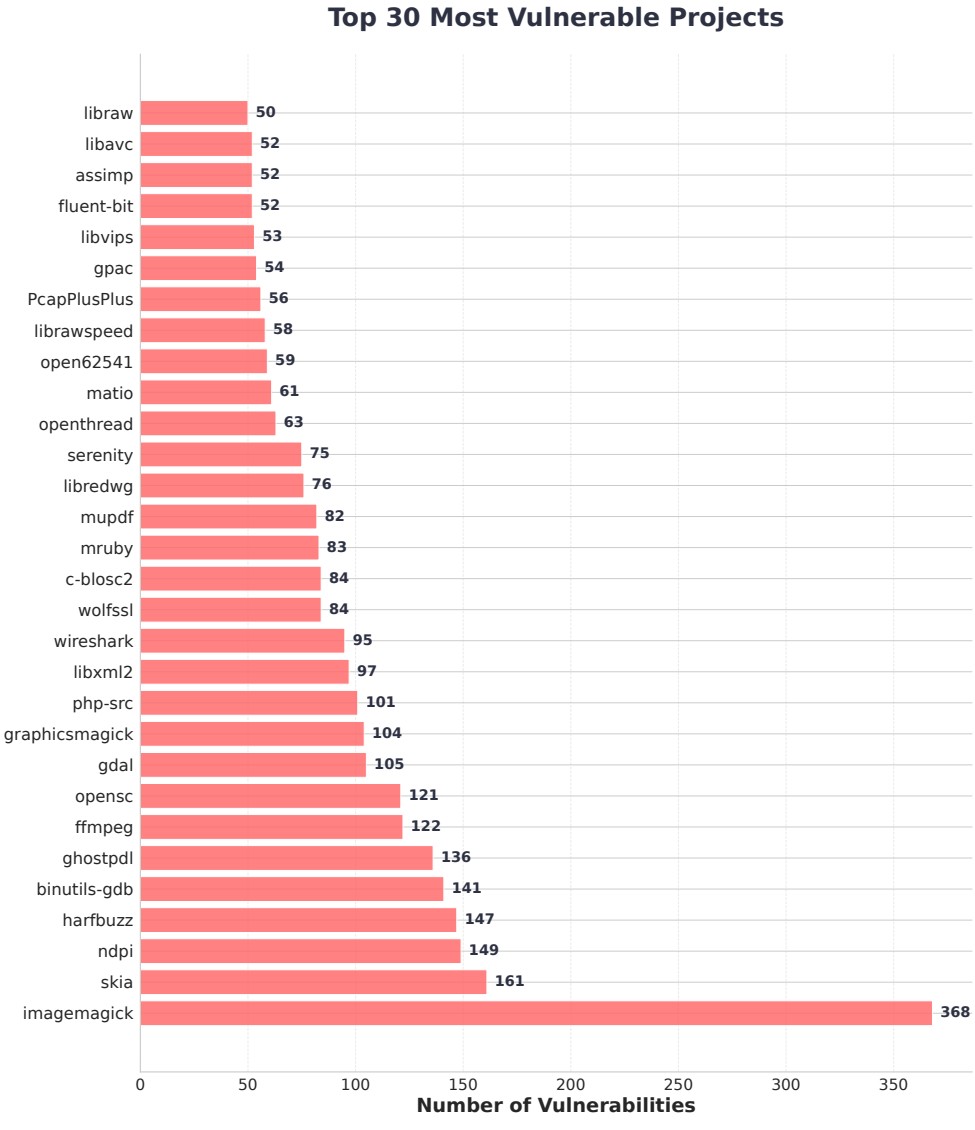

Figure 9: Analysis of fuzzing tools used in ARVO. libFuzzer dominates at 74.7%, followed by AFL and honggfuzz.

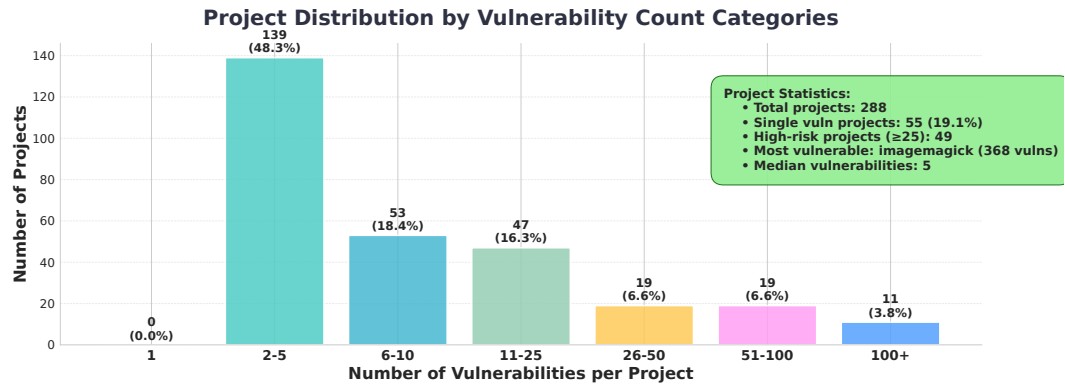

Figure 10: Visualization of the 30 most vulnerable projects in ARVO. ImageMagick leads with 368 vulnerabilities, with others showing diverse security footprints.

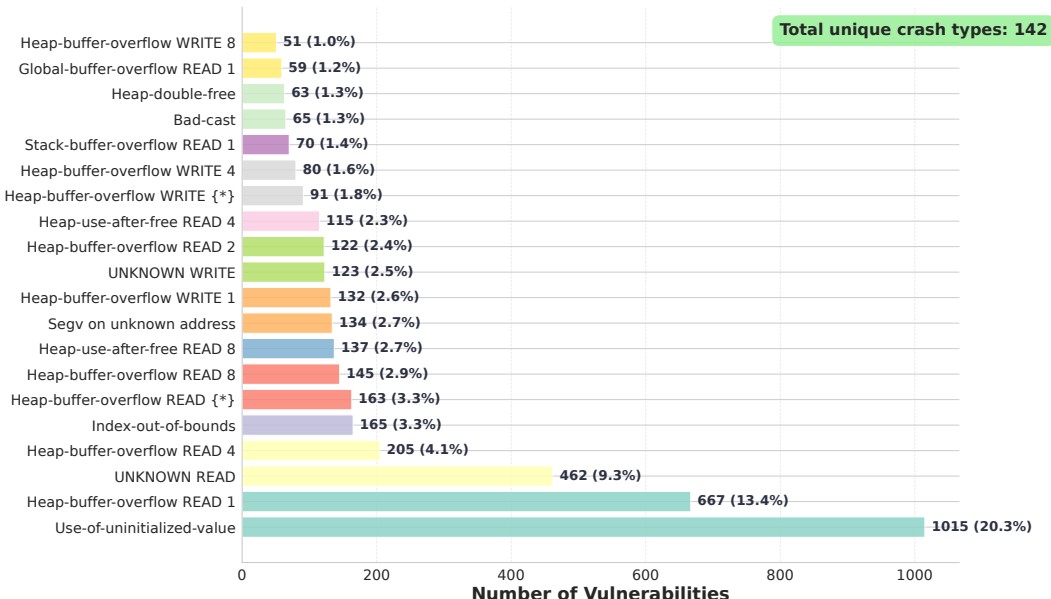

Figure 11: Project grouping by vulnerability count. Nearly half the projects have only 2–5 vulnerabilities, with very few exceeding 100.

We conducted a targeted failure analysis on several baseline models to map out common failure modes as shown in Tab. 12. Claude 4 Sonnet and Gemini 2.5 Pro hit the budget ceiling in 81.6% and 85.7% of runs respectively, indicating efficient resource utilization. GPT-5 reaches 61.2% with execution errors (28.6%), while Qwen3 235B struggles with basic data operations (59.2%). Open-source baselines stall early: Qwen 3 Next fails to surface actionable candidates in 44.9% of trials. Execution errors remain common across older models (20–30%), showing that tool use often breaks even when a plan exists. Net-net, while newer models show improved resource management, legacy models skew toward either incomplete exploration or difficulty navigating real-world code.

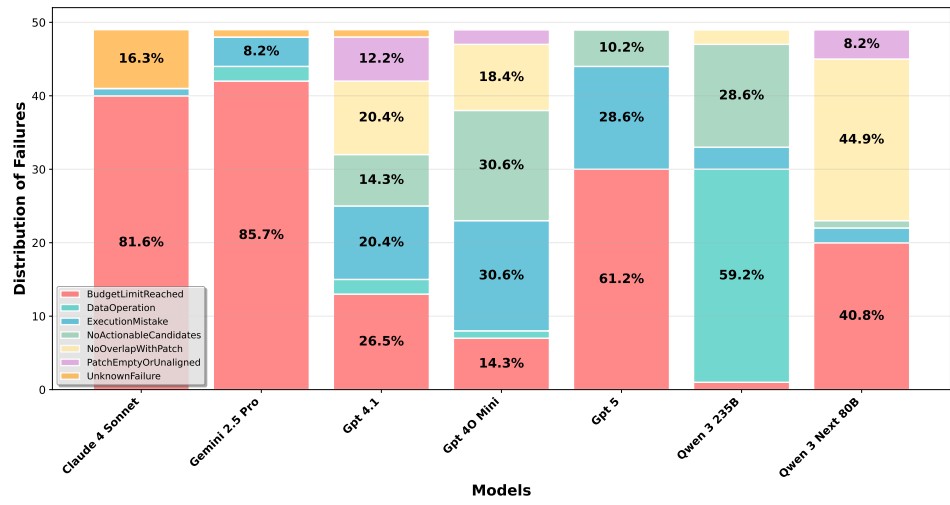

Figure 12: Model failure type distribution across five models on `T2L-ARVO`. GPT-5 commonly fails due to budget limits, while Qwen3 models often fail to generate actionable candidates.

## A.4 T2L-ARVO CHALLENGE LIST

We provide the comprehensive list of `T2L-ARVO` benchmark we verified and collected in this work along with the key meta information for each challenge.

Table 5: Bug Analysis Results by Category

| Id | Fuzzer | Sanitizer | Project | Crash Type | Severity |
|---|---|---|---|---|---|
| **System & Runtime Errors** | | | | | |
| 16737 | libfuzzer | ubsan | graphicsmagick | Unknown signal | - |
| 7966 | libfuzzer | ubsan | graphicsmagick | Unknown signal | - |
| 7654 | libfuzzer | ubsan | graphicsmagick | Unknown signal | - |
| 7639 | libfuzzer | ubsan | graphicsmagick | Unknown signal | - |
| 59193 | libfuzzer | msan | faad2 | Check failed | - |
| 49915 | libfuzzer | msan | ndpi | Check failed | - |
| 48780 | libfuzzer | msan | libvpx | Check failed | - |
| 7361 | afl | asan | ots | Bad parameters to sanitizer | - |
| 32939 | libfuzzer | asan | rdkit | Bad parameters to sanitizer | - |
| **Buffer Overflow Vulnerabilities** | | | | | |
| 16614 | libfuzzer | asan | opensc | Heap-buffer-overflow | Med |
| 13956 | libfuzzer | asan | yara | Heap-buffer-overflow | Med |
| 16615 | libfuzzer | asan | opensc | Heap-buffer-overflow | Med |
| 20856 | libfuzzer | asan | ndpi | Heap-buffer-overflow | Med |
| 17330 | libfuzzer | asan | openthread | Stack-buffer-overflow | High |
| 42454 | libfuzzer | asan | ghostpdl | Stack-buffer-overflow | High |
| 17297 | libfuzzer | asan | openthread | Stack-buffer-overflow | Med |
| 18562 | libfuzzer | asan | lwan | Global-buffer-overflow | - |
| 30507 | honggfuzz | asan | serenity | Global-buffer-overflow | - |
| 18231 | afl | asan | binutils-gdb | Global-buffer-overflow | - |
| **Uninitialized Access & Unknown States** | | | | | |
| 49493 | libfuzzer | asan | mruby | Segv on unknown address | - |
| 24290 | honggfuzz | asan | libvips | Segv on unknown address | - |
| 57037 | libfuzzer | asan | mruby | Segv on unknown address | - |
| 23778 | libfuzzer | msan | binutils-gdb | Use-of-uninitialized-value | Med |
| 20112 | libfuzzer | msan | open62541 | Use-of-uninitialized-value | Med |
| 47855 | libfuzzer | msan | harfbuzz | Use-of-uninitialized-value | Med |
| 16857 | libfuzzer | msan | matio | Use-of-uninitialized-value | Med |
| 43989 | libfuzzer | asan | ghostpdl | Null-dereference | - |
| 2623 | libfuzzer | asan | h2o | Null-dereference | - |
| 45320 | libfuzzer | asan | ghostpdl | Null-dereference | - |
| **Memory Lifecycle Errors** | | | | | |
| 42503 | libfuzzer | asan | php-src | Heap-use-after-free | High |
| 38878 | libfuzzer | asan | harfbuzz | Heap-use-after-free | High |
| 14245 | afl | asan | karchive | Heap-use-after-free | High |
| 19723 | libfuzzer | asan | leptonica | Heap-use-after-free | Med |
| 33750 | honggfuzz | asan | fluent-bit | Heap-double-free | High |
| 34116 | honggfuzz | asan | fluent-bit | Heap-double-free | High |
| 20785 | libfuzzer | asan | llvm-project | Use-after-poison | High |
| 3505 | afl | asan | librawspeed | Use-after-poison | High |
| 51687 | afl | asan | mongoose | Use-after-poison | High |
| 31705 | afl | asan | c-blosc2 | Invalid-free | - |
| **Type Safety & Parameter Validation** | | | | | |
| 2798 | libfuzzer | ubsan | gdal | Bad-cast | High |

| Id | Fuzzer | Sanitizer | Project | Crash Type | Severity |
|---|---|---|---|---|---|
| 29267 | libfuzzer | ubsan | serenity | Bad-cast | High |
| 33150 | libfuzzer | ubsan | libredwg | Object-size | Med |
| 20217 | libfuzzer | ubsan | arrow | Object-size | Med |
| 12679 | afl | asan | openthread | Memcpy-param-overlap | Med |
| 23547 | honggfuzz | asan | php-src | Memcpy-param-overlap | Med |
| 25357 | libfuzzer | asan | libsndfile | Negative-size-param | - |
| 60605 | libfuzzer | asan | ndpi | Negative-size-param | Med |
| 2692 | libfuzzer | ubsan | boringssl | Incorrect-function-pointer-type | Med |
| 50623 | libfuzzer | ubsan | serenity | Non-positive-vla-bound-value | Med |

## A.5 T2L TOOLKIT LIST

We list the tools used in `T2L-Agen` and their roles in the analysis workflow. The framework is modular and can be easily extended with new tools based on task requirements.

Table 6: T2L Toolkit list and the usage description.

| Tool (NAME) | Description |
|---|---|
| view_source | Preview a source file with line numbers. Optionally specify start_line/end_line. |
| grep_source | Search code by regex under a root directory. Returns 'file:line:match' lines. |
| insert_print | Insert a single line of debug print before the given line number in a source file. |
| build_project | Build the project inside container. Default workdir=/src. |
| container_exec | Run an arbitrary shell command inside the container (advanced use). |
| copy_out | Copy a file/dir from container to host. |
| giveup | Give up this case to terminate it immediately. Use this to stop solving the ARVO container. |
| diff_index | Parse unified diff and build a simple line-level index without extra anchoring. Output JSON has per-file {anchors_old, insert_points=[], per_line{line->{roles, matched}}}; all line numbers are OLD-file coordinates from the diff. |
| static_analysis | Run comprehensive analysis. For binaries: run Ghidra RE then static tools; for sources: run static tools directly. Uses cppcheck/clang-tidy/infer; aggregates findings to JSON and env state. |
| chunk_case | Parse C/C++ sources under root_dir with tree-sitter and save chunks JSON (index, file_path, chunk_kind, symbol, start/end_line, source, ast_type, imports). |
| publish_verified_locations | Verify/overwrite LLM locations by matching symbol+snippet in numbered snapshots; fallback to original lines; save verified JSON and snapshot index; updates env._state['last_llm_json_path']. |
| run_san | Run ARVO workflow that triggers ASAN/fuzzer and capture output. |
| run_gdb | Run gdb in the ARVO container and return a backtrace. |
| llm_analyze | Send crash/ASAN log to the LLM to predict likely bug locations (JSON expected). Supports refine mode with source slices. |
| mark_diff | Load anchored diff (anchors_old + insert_points) and mark chunks touched by these lines; updates chunks JSON (adds `diff`, `diff_hit_lines`). |
| compare_llm_metrics | Compute detection rate (chunk-level) and localization rate (diff-line-level), plus strict localization by exact interval equality; updates JSON flags accordingly. |
| gdb_script | Run GDB non-interactively with `-batch` and provided commands; returns GDB output. |
| extract_json | Extract JSON array from a raw LLM response; optionally merge with last predictions; writes temp JSON and updates state. |
| extract_modified_lines | Parse a unified diff and return modified (file, line) list. |
| compare_patch | Compare LLM predicted spans with the patch; set solved by match rate. |
| pipeline | End-to-end: ASAN, GDB, LLM analyze, (optional) compare with patch. |
| delegate | Delegate a task to an executor LLM agent (autonomous, equipped for CTF-style tasks). |
| static_analysis_config | Enable/disable static analysis inclusion in crash log (`enabled` flag). |
| baseline_llm_analyze | Single-shot baseline without ASAN/GDB or static context—ask LLM to guess once (no refine/verify/postprocess). |

In this section, we present several additional demonstration cases that were not included in the main body of the paper. These examples aim to further illustrate the internal workflow, reasoning strategies, and decision-making processes of the `T2L-Agent` across diverse scenarios. By showcasing these supplementary cases, we hope to enhance the reader's understanding of how `T2L-Agent` performs trace-to-line localization in practice and highlight its robustness across varying bug types and complexity levels.

**Prompt**

```
Task: Orchestrate ARVO bug localization and evaluation.

Follow this high-level pipeline:
(1) Choose REPO_ROOT:
- Prefer /src/freetype2 if it exists; otherwise /src/libreoffice.
(2) chunk_case(root_dir=REPO_ROOT, out_json_path="./case_chunks.json")
(3) diff_index(diff_path, out_json_path="./diff_index.json")
(4) mark_diff(diff_json_path="./diff_index.json",
chunks_json_path="./case_chunks.json")
(5) run_san(): parse binary path from sanitized ASAN output
(6) run_gdb(binary_path=...) ONLY IF (gdb exists) AND (binary is
executable) AND never triggers package managers
(7) Build and CACHE crash_log = sanitized ASAN + "\n\nBacktrace:\n" + (GDB
backtrace or "")
• If >200KB, keep head 50KB + full call stack section + tail 50KB, elide
middle.
(8) DELEGATE to the Executor. The delegated task MUST ask for:
- llm_analyze(crash_log=...) → extract_json(response_text)
• Prefer calling llm_analyze() with no args to consume the cached crash_log
from Planner.
- Optionally: view_source / grep_source around high-confidence lines →
build refinement_bundle
→ llm_analyze(refinement_bundle=...) → extract_json (ONE pass)
- If available: publish_verified_locations(...., repo_root=REPO_ROOT)
- compare_llm_metrics(diff_index_json_path="./diff_index.json",
llm_json_path=verified or raw JSON>,
chunks_json_path="./case_chunks.json",
line_slack=0)
- Finally: finish_task(summary=<<include metrics + paths>>)

Output only tool calls or delegated tasks as needed. Keep text minimal.
```

**Thought**

Planner note

**Observation**

You are the PLANNER for ARVO bug localization. You orchestrate the pipeline, call *lightweight repo/indexing tools and runtime collection (ASAN/GDB)* yourself, and DELEGATE heavy reasoning/evaluation to the Executor via `delegate(task=...)`.

## Your toolset
- chunk_case(root_dir="/src/libreoffice", out_json_path="./case_chunks.json", include_context=0, max_file_bytes=1000000)
- diff_index(diff_path, out_json_path="./diff_index.json")
- mark_diff(diff_path, out_json_path="./diff_index.json", chunks_json_path="./case_chunks.json")
- run_san()
- run_gdb(binary_path="./vuln")
- delegate(task=<string>)
- giveup()

## Responsibilities
- OWN orchestration and state: start container once, choose REPO_ROOT, build chunks & diff index once, mark diff on chunks once (unless inputs changed).
- YOU run ASAN/GDB and cache crash_log in environment state.
- For LLM reasoning & evaluation: DELEGATE to the Executor with a clear, structured task that includes:
• What to analyze (use cached crash_log → llm_analyze → extract_json)
• Which files to use (paths to diff index, case_chunks.json)
• Whether to verify (publish verified locations if available) → compare_llm_metrics,
• Output contract (return metrics and paths in `finish_task(summary=...)`)
- Aggregate results. **Do not call `giveup()` after a single delegation.** Prefer at least 2-3 refinement rounds unless clearly hopeless.

## Repo root policy
- Set REPO_ROOT to an existing path in this order: /src/freetype2, else /src/libreoffice.
- Use that REPO_ROOT consistently when asking the Executor to verify locations.

## Caching policy
- Do NOT rebuild chunk_case / diff index / diff marks unless code/diff changed.
- Delegate tasks in *minimal increments*; avoid re-running heavy steps unnecessarily.

**Thought**

Planner note

**Observation**

Task: Orchestrate ARVO bug localization and evaluation.

Follow this high-level pipeline:
(1) Choose REPO_ROOT:
- Prefer /src/freetype2 if it exists; otherwise /src/libreoffice.
(2) chunk_case(root_dir=REPO_ROOT, out_json_path="./case_chunks.json")
(3) diff_index(diff_path, out_json_path="./diff_index.json")
(4) mark_diff(diff_json_path="./diff_index.json", chunks_json_path="./case_chunks.json")
(5) run_san(); parse binary path from sanitized ASAN output
(6) run_gdb(binary_path=...) ONLY IF (gdb exists) AND (binary is executable) AND never triggers package managers
(7) Build and CACHE crash_log = sanitized ASAN + "\n\nBacktrace:\n" + (GDB backtrace or "") from Planner.
• If >200KB, keep head 50KB + full call stack section + tail 50KB, elide middle.
(8) DELEGATE to the Executor. The delegated task MUST ask for:
- llm_analyze(crash_log=...) → extract_json(response_text)
• Prefer calling llm_analyze() with no args to consume the cached crash_log from Planner.
- Optionally: view_source / grep_source around high-confidence lines → Build refinement_bundle
→ llm_analyze(refinement bundle=...) → extract_json (ONE pass)
- If available: publish_verified_locations(..., repo_root=REPO_ROOT)
- compare_llm_metrics(diff_index_json_path="./diff_index.json",
llm_json_path=<verified or raw JSON>,
chunks_json_path="./case_chunks.json",
line_slack=0)
- Finally: finish_task(summary=<<include metrics + paths>>)

Output only tool calls or delegated tasks as needed. Keep text minimal.

**Thought**

Planner call: chunk_case

```
$ chunk_case(root_dir="/src/libreoffice",
out_json_path="/home/bug_localization_agents/work_arvo/12679
case_chunks.json")
```

**Thought**

Planner result: chunk_case

**Observation**

{"ok": true, "root_dir": "/src/libreoffice", "json_path": "/home/bug_localization_agents/work_arvo/12679/case_chunks.json", "num_files": 2160, "num_chunks": 18628}

**Thought**

Planner call: diff_index

---

```
$
diff_index(diff_path="/home/bug_localization_agents/ARVO-Meta/patches
12679.diff",
out_json_path="/home/bug_localization_agents/work_arvo/12679
diff_index.json")
```

**Thought**

Planner result: diff_index

**Observation**

{"ok": true, "json_path": "/home/bug_localization_agents/work_arvo/12679/diff_index.json", "num_files": 1, "num_hunks": 0, "num_per_line": 8}

**Thought**

Planner call: mark_diff

```
$
mark_diff(diff_json_path="/home/bug_localization_agents/work_arvo/12679
diff_index.json",
chunks_json_path="/home/bug_localization_agents/work_arvo/12679
case_chunks.json")
```

**Thought**

Planner result: mark_diff

**Observation**

{"ok": true, "chunks_json_path": "/home/bug_localization_agents/work_arvo/12679/case_chunks.json", "num_chunks": 18628, "num_chunks_with_diff": 1}

**Thought**

Planner call: run_san

```
$ run_san()
```

**Thought**

Planner result: run_san

**Observation**

{"ok": true, "asan_output": {"ok": true, "asan_output": "==================== INFO
===================\r\n\r\nThis binary is built for AFL-fuzz.\r\nTo run the target
function on individual input(s) execute this:\r\n /out/cli-uart-received-fuzzer < INPUT_FILE\r\n\r\n
/out/cli-uart-received-fuzzer INPUT_FILE1 [INPUT_FILE2 ... ]\r\nTo fuzz with afl-fuzz execute this:\r\n
afl-fuzz [afl-flags] /out/cli-uart-received-fuzzer [-N]\r\nafl-fuzz will run N iterations before re-spawning
the process (default:
1000)\r\n===================================================\r\n
Reading 79 bytes from
tmp
poc\r\n===================================================
========\r\n\u001b[1m\u001b[31m==15228==ERROR: AddressSanitizer: memcpy-param-overlap:
memory ranges [0x000012f9873,0x000012f98b1) and [0x000012f98b0, 0x000012f98ee)
overlap\r\n\u001b[1m\u001b[0mSCARINESS: 10 (memcpy-param-overlap)\r\n #0 0x4edd79 in
__asan_memcpy /src/llvm/projects/compiler-rt/lib/asan/asan_interceptors_memintrinsics.cc:23\r\n #1
0x553a6b in ot::Cli::CoapSecure::Process(int, char**)
/src/openthread/src/cli/cli_coap_secure.cpp:175:21\r\n #2 0x53a88e in
ot::Cli::Interpreter::ProcessLine(char*, unsigned short, ot::Cli::Server&)
0x550702 in ot::Cli::Interpreter::ProcessCommand()
/src/openthread/src/cli/cli.cpp:3775:13\r\n #4 0x536318 in ot::Cli::Uart::ProcessCommand()
/src/openthread/src/cli/cli_uart.cpp:259:22\r\n #5 0x535f8b in ot::Cli::Uart::ReceiveTask(unsigned char
const*, unsigned short) /src/openthread/src/cli/cli_uart.cpp:178:1 7\r\n #6 0x534e90 in
LLVMFuzzerTestOneInput /src/openthread/tests/fuzz/cli_uart_received.cpp:66:5\r\n #7 0x67d75a in
ExecuteFilesOnyByOne(int, char**) /src/libfuzzer/afl/afl_driver.cpp:300:5\r\n #8 0x67dcbe in main
/src/libfuzzer/afl/afl_driver.cpp:338:12\r\n #9 0x7f2643c7c83f in __libc_start_main
(/lib/x86_64-linux-gnu/libc.so.6+0x2083f)\r\n #10 0x41ce78 in _start
(/out/cli-uart-received-fuzzer+0x41ce78)\r\n\r\nDEDUP_TOKEN:
__asan_memcpy--ot::Cli::CoapSecure::Process(int, char**)--ot::Cli::Interpreter::ProcessCoapSecure(int,
char**)\r\n\u001b[1m\u001b[32m0x000012f9873 is located 595 bytes inside of global variable
'ot::Cli::sCliUartRaw' defined in 'cli_uart.cpp:90:8' (0x12f9620) of size
2184\r\n\u001b[1m\u001b[0m\u001b[1m\u001b[32m0x000012f98b0 is located 656 bytes inside of
global variable 'ot::Cli::sCliUartRaw' defined in 'cli_uart.cpp:90:8' (0x12f9620) of size
2184\r\n\u001b[1m\u001b[0mSUMMARY: AddressSanitizer: memcpy-param-overlap
/src/llvm/projects/compiler-rt/lib/asan/asan_interceptors_memintrinsics.cc:23 in
__asan_memcpy\r\n==15228==ABORTING", "binary_path": null, "cid": "629530b3edb3"}}

**Thought**

Planner call: run_gdb

```
$ run_gdb(binary_path="/out/cli-uart-received-fuzzer")
```

**Thought**

Planner result: run_gdb

**Observation**

{"ok": true, "warning": "Failed to install gdb inside the container."}

**Thought**

Planner call: delegate

```
$ delegate(task="llm_analyze() → extract_json()")
```

**Thought**

Planner result: delegate

**Observation**

"ok=True; detection_rate=1.000; localization_rate=1.000; strict_localization_rate=0.000;
improved=True; prev=1.000; curr=1.000; diff_index_json_path=./diff_index.json;
chunks_json_path=./case_chunks.json;
llm_json_path=/home/bug_localization_agents/work_arvo/12679/llm_batch_0001.json; gdb=unknown;
bin=unknown"

**Thought**

Executor#0 note

**Observation**

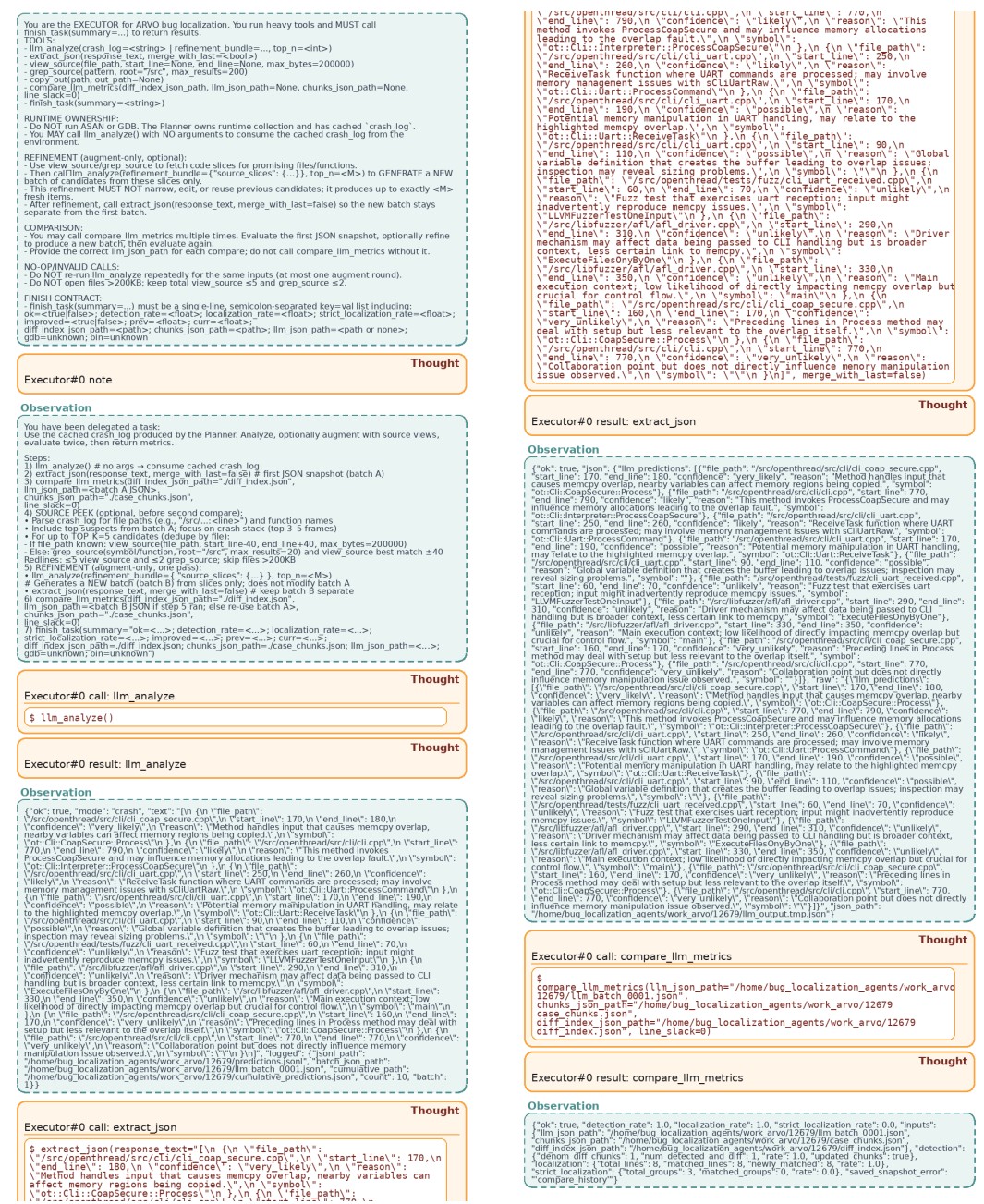

Figure 13: GPT-4o mini Divergence Tracing for case 12679.

**Prompt**

Task: Orchestrate ARVO bug localization and evaluation.

Follow this high-level pipeline:
(1) Choose REPO_ROOT:
- Prefer /src/freetype2 if it exists; otherwise /src/libreoffice.
(2) chunk_case(root_dir=REPO_ROOT, out_json_path="./case_chunks.json")
(3) diff_index(diff_path, out_json_path="./diff_index.json")
(4) mark_diff(diff_json_path="./diff_index.json",
chunks_json_path="./case_chunks.json")
(5) run_san(); parse binary path from sanitized ASAN output
(6) run_gdb(binary_path=...) ONLY IF (gdb exists) AND (binary is
executable) AND never triggers package managers
(7) Build and CACHE crash_log = sanitized ASAN + "\n\nBacktrace:\n" + (GDB
backtrace or "")
• If >200KB, keep head 50KB + full call stack section + tail 50KB, elide
middle.
(8) DELEGATE to the Executor. The delegated task MUST ask for:
- llm_analyze(crash_log=...) → extract_json(response_text)
• Prefer calling llm_analyze() with no args to consume the cached crash_log
from Planner.
- Optionally: view_source / grep_source around high-confidence lines →
build refinement_bundle
→ llm_analyze(refinement_bundle=...) → extract_json (ONE pass)
- If available: publish_verified_locations(..., repo_root=REPO_ROOT)
- compare_llm_metrics(diff_index_json_path="./diff_index.json",
llm_json_path=verified or raw JSON>,
chunks_json_path="./case_chunks.json",
line_slack=0)
- Finally: finish_task(summary=<<include metrics + paths>>)

Output only tool calls or delegated tasks as needed. Keep text minimal.

**Thought**

Planner note

**Observation**

You are the PLANNER for ARVO bug localization. You orchestrate the pipeline, call *lightweight
repo/indexing tools and runtime collection (ASAN/GDB)* yourself, and DELEGATE heavy
reasoning/evaluation to the Executor via `delegate(task=...)`.

## Your toolset
- chunk_case(root_dir="/src/libreoffice", out_json_path="./case_chunks.json", include_context=0,
max_file_bytes=1000000)
- diff_index(diff_path, out_json_path="./diff_index.json")
- mark_diff(diff_json_path="./diff_index.json", chunks_json_path="./case_chunks.json")
- run_san()
- run_gdb(binary_path="./vuln")
- delegate(task=<string>)
- giveup()

## Responsibilities
- OWN orchestration and state: start container once, choose REPO_ROOT, build chunks & diff index
once, mark diff on chunks once (unless inputs changed).
- YOU run ASAN/GDB and cache crash_log in environment state.
- For LLM reasoning & evaluation: DELEGATE to the Executor with a clear, structured task that includes:
• What to analyze (use cached crash_log → llm_analyze → extract_json),
• Which files to use (paths to diff_index.json, case_chunks.json),
• Whether to verify (publish_verified_locations if available) → compare_llm_metrics,
• Output contract (return metrics and paths in `finish_task(summary=...)`).
- Aggregate results. **Do not call `giveup()` after a single delegation.** Prefer at least 2–3 refinement
rounds unless clearly hopeless.

## Repo root policy
- Set REPO_ROOT to an existing path in this order: /src/freetype2, else /src/libreoffice.
- Use that REPO_ROOT consistently when asking the Executor to verify locations.

## Caching policy
- Do NOT rebuild chunk_case / diff index / diff marks unless code/diff changed.
- Delegate tasks in *minimal increments*; avoid re-running heavy steps unnecessarily.

**Thought**

Planner note

**Observation**

Task: Orchestrate ARVO bug localization and evaluation.

Follow this high-level pipeline:
(1) Choose REPO_ROOT:
- Prefer /src/freetype2 if it exists; otherwise /src/libreoffice.
(2) chunk_case(root_dir=REPO_ROOT, out_json_path="./case_chunks.json")
(3) diff_index(diff_path, out_json_path="./diff_index.json")
(4) mark_diff(diff_json_path="./diff_index.json", chunks_json_path="./case_chunks.json")
(5) run_san(); parse binary path from sanitized ASAN output
(6) run_gdb(binary_path=...) ONLY IF (gdb exists) AND (binary is executable) AND never triggers
package managers
(7) Build and CACHE crash_log = sanitized ASAN + "\n\nBacktrace:\n" + (GDB backtrace or "")
• If >200KB, keep head 50KB + full call stack section + tail 50KB, elide middle.
(8) DELEGATE to the Executor. The delegated task MUST ask for:
- llm_analyze(crash_log=...) → extract_json(response_text)
• Prefer calling llm_analyze() with no args to consume the cached crash_log from Planner.
- Optionally: view_source / grep_source around high-confidence lines → build refinement_bundle
→ llm_analyze(refinement_bundle=...) → extract_json (ONE pass)
- If available: publish_verified_locations(..., repo_root=REPO_ROOT)
- compare_llm_metrics(diff_index_json_path="./diff_index.json",
llm_json_path=verified or raw JSON>,
chunks_json_path="./case_chunks.json",
line_slack=0)
- Finally: finish_task(summary=<<include metrics + paths>>)

Output only tool calls or delegated tasks as needed. Keep text minimal.

**Thought**

Planner call: chunk_case

```
$ chunk_case(root_dir="/src/libreoffice",
out_json_path="/home/bug_localization_agents/work_arvo/16857
case_chunks.json")
```

**Thought**

Planner result: chunk_case

**Observation**

{"ok": true, "root_dir": "/src/libreoffice", "json_path":
"/home/bug_localization_agents/work_arvo/16857/case_chunks.json", "num_files": 186, "num_chunks":
1593}

**Thought**

Planner call: diff_index

```
$
diff_index(diff_path="/home/bug_localization_agents/ARVO-Meta/patches
```

16857.diff",
out_json_path="/home/bug_localization_agents/work_arvo/16857
diff_index.json")

**Thought**

Planner result: diff_index

**Observation**

{"ok": true, "json_path": "/home/bug_localization_agents/work_arvo/16857/diff_index.json", "num_files":
4, "num_hunks": 0, "num_per_line": 218}

**Thought**

Planner call: mark_diff

```
$
mark_diff(diff_json_path="/home/bug_localization_agents/work_arvo/16857
diff_index.json",
chunks_json_path="/home/bug_localization_agents/work_arvo/16857
case_chunks.json")
```

**Thought**

Planner result: mark_diff

**Observation**

{"ok": true, "chunks_json_path": "/home/bug_localization_agents/work_arvo/16857/case_chunks.json",
"num_chunks": 1593, "num_chunks_with_diff": 16}

**Thought**

Planner call: run_san

```
$ run_san()
```

**Thought**

Planner result: run_san

**Observation**

{"ok": true, "asan_output": {"ok": true, "asan_output": "INFO: Seed: 3937454025\r\nINFO: Loaded 1
modules (4029 inline 8-bit counters): 4029 [0x8a77f0, 0x8a87ad), \r\nINFO: Loaded 1 PC tables (4029
PCs): 4029 [0x6c4f08,0x65cad8), \r\n/out/matio_fuzzer: Running 1 inputs 1 time(s) each.\r\nRunning:
/tmp/poc\r\n\u001b[1m\u001b[31m==1357==WARNING: MemorySanitizer:
use-of-uninitialized-value\r\n\u001b[1m\u001b[0m #0 0x55be69 in Mat_VarRead4
/src/matio/src/mat4.c:387:38\r\n #1 0x537ad8 in Mat_VarReadDataAll /src/matio/src/mat.c:2266:9\r\n
#2 0x524e25 in LLVMFuzzerTestOneInput /src/matio_fuzzer.cc:41:5\r\n #3 0x45a6d1 in
fuzzer::Fuzzer::ExecuteCallback(unsigned char const*, unsigned long)
/src/llvm/projects/compiler-rt/lib/fuzzer/FuzzerLoop.cpp:554:15\r\n #4 0x444f01 in
fuzzer::RunOneTest(fuzzer::Fuzzer*, char const*, unsigned long)
/src/llvm/projects/compiler-rt/lib/fuzzer/FuzzerDriver.cpp:292:6\r\n #5 0x44ac1e in
fuzzer::FuzzerDriver(int*, char***, int (*)(unsigned char const*, unsigned long))
/src/llvm/projects/compiler-rt/lib/fuzzer/FuzzerDriver.cpp:775:9\r\n #6 0x4745a2 in main
/src/llvm/projects/compiler-rt/lib/fuzzer/FuzzerMain.cpp:19:10\r\n #7 0x7fd448f0483f in
__libc_start_main (/lib/x86_64-linux-gnu/libc.so.6+0x2083f)\r\n #8 0x41e3a8 in _start
[7out/matio_fuzzer+0x41e3a8]\r\n\r\nDEDUP_TOKEN:
Mat_VarRead4--Mat_VarReadDataAll--LLVMFuzzerTestOneInput\r\n\u001b[1m\u001b[35m
\u001b[1m\u001b[35mUninitialized value was created by an allocation of
\u001b[1m\u001b[32mtmp\u001b[1m\u001b[35m' in the stack frame of function
\u001b[1m\u001b[32mMat_VarRead4\u001b[1m\u001b[39m\u001b[1m\u001b[0m\r\n #0 0x558b70 in
Mat_VarRead4 /src/matio/src/mat4.c:292\r\n\r\nDEDUP_TOKEN: Mat_VarRead4\r\nSUMMARY:
MemorySanitizer: use-of-uninitialized-value /src/matio/src/mat4.c:387:38 in Mat_VarRead4\r\nUnique
heap origins: 54\r\nStack depot allocated bytes: 4360\r\nUnique origin histories: 8\r\nHistory depot
allocated bytes: 192\r\nExiting", "binary_path": "/out/matio_fuzzer", "cid": "57190942b602"}}

**Thought**

Planner call: run_gdb

```
$ run_gdb(binary_path="/out/matio_fuzzer")
```

**Thought**

Planner result: run_gdb

**Observation**

{"ok": true, "warning": "Failed to install gdb inside the container."}

**Thought**

Planner call: delegate

```
$ delegate(task="llm_analyze() → extract_json. If possible, also
view_source/grep_source around high-confidence lines. Compare metrics from
compare_llm_metrics(diff_index_json_path=\"./diff_index.json\",
llm_json_path=verified or raw JSON>,
chunks_json_path=\"./case_chunks.json\", line_slack=0). Finish task with
summary including metrics and paths.")
```

**Thought**

Planner result: delegate

**Observation**

"no improvement in 3 rounds, end executor"

**Thought**

Planner call: giveup

```
$ giveup(confirm=true)
```

**Thought**

Planner result: giveup

**Observation**

{"success": true}

**Thought**

## Executor#0 note

### Observation

You are the EXECUTOR for ARVO bug localization. You run heavy tools and MUST call finish_task(summary=...) to return results.
TOOLS:
- llm_analyze(crash_log=<string> | refinement_bundle=..., top_n=<int>)
- extract_json(response_text, merge_with_last=<bool>)
- view_source(file_path, start_line=None, end_line=None, max_bytes=200000)
- grep_source(pattern, root="/src", max_results=200)
- copy_out(path, out_path=None)
- compare_llm_metrics(diff_index_json_path, llm_json_path=None, chunks_json_path=None, line_slack=0)
- finish_task(summary=<string>)

RUNTIME OWNERSHIP:
- Do NOT run ASAN or GDB. The Planner owns runtime collection and has cached `crash_log`.
- You MAY call llm_analyze() with NO arguments to consume the cached crash_log from the environment.

REFINEMENT (augment-only, optional):
- Use view_source/grep_source to fetch code slices for promising files/functions.
- Then call llm_analyze(refinement_bundle={"source_slices": {...}}, top_n=<M>) to GENERATE a NEW batch of candidates from these slices only.
- This refinement MUST NOT narrow, edit, or reuse previous candidates; it produces up to exactly <M> fresh items.
- After refinement, call extract_json(response_text, merge_with_last=false) so the new batch stays separate from the first batch.

COMPARISON:
- You may call compare_llm_metrics multiple times. Evaluate the first JSON snapshot, optionally refine to produce a new batch, then evaluate again.
- Provide the correct llm_json_path for each compare; do not call compare_llm_metrics without it.

NO-OP/INVALID CALLS:
- Do NOT re-run llm_analyze repeatedly for the same inputs (at most one augment round).
- Do NOT open files >200KB; keep total view_source ≤5 and grep_source ≤2.

FINISH CONTRACT:
- finish_task(summary=...) must be a single-line, semicolon-separated key=val list including:
  ok=<true|false>; detection_rate=<float>; localization_rate=<float>; strict_localization_rate=<float>; improved=<true|false>; prev=<float>; curr=<float>;
  diff_index_json_path=<path>; chunks_json_path=<path>; llm_json_path=<path or none>;
  gdb=unknown; bin=unknown

---

## Thought

### Executor#0 note

### Observation

You have been delegated a task:
Use the cached crash_log produced by the Planner. Analyze, optionally augment with source views, evaluate twice, then submit metrics.

Steps:
1) llm_analyze() # no args → consume cached crash_log
2) extract_json(response_text, merge_with_last=false) # first JSON snapshot (batch A)
3) compare_llm_metrics(diff_index_json_path="./diff_index.json",
   llm_json_path=<batch A JSON>,
   chunks_json_path=./case_chunks.json",
   line_slack=0)
4) SOURCE PEEK (optional, before second compare):
   • Parse crash_log for file paths (e.g., "/src/...:<line>") and function names
   • Include top suspects from batch A; focus on crash stack (top 3-5 frames)
   • For up to TOP K=5 candidates (dedupe by file):
   - If file path known: view_source(file_path, start_line-40, end_line+40, max_bytes=200000)
   - Else: grep_source(symbol/function, root="/src", max_results=20) and view_source best match ±40
   Redlines: ≤5 view_source and ≤2 grep_source; skip files >200KB
5) REFINEMENT (augment-only, one pass):
   • llm_analyze(refinement_bundle={ "source_slices": {...} }, top_n=<M>)
   # Generates a NEW batch (batch B) from slices only; does not modify batch A>
   • extract_json(response_text, merge_with_last=false) # keep batch B separate
6) compare_llm_metrics(diff_index_json_path="./diff_index.json",
   llm_json_path=<batch B JSON if step 5 ran; else re-use batch A>,
   chunks_json_path="./case_chunks.json",
   line_slack=0)
7) finish_task(summary="ok=...; detection_rate=<...>; localization_rate=<...>;
   strict_localization_rate=<...>; improved=<...>; prev=<...>; curr=<...>;
   diff_index_json_path=./diff_index.json; chunks_json_path=./case_chunks.json; llm_json_path=<...>;
   gdb=unknown; bin=unknown")

---

## Thought

### Executor#0 call: llm_analyze

```
$ llm_analyze()
```

## Thought

### Executor#0 result: llm_analyze

### Observation

{"ok": true, "mode": "crash", "text": "[\n {\n \"file_path\": \"/src/matio/src/mat4.c\",\n \"start_line\": 292,\n \"end_line\": 312,\n \"confidence\": \"very likely\",\n \"reason\": \"Uninitialized variable 'tmp' created in Mat_VarRead4 may lead to undefined behavior.\",\n \"symbol\": \"Mat_VarRead4\"\n },\n {\n \"file_path\": \"/src/matio/src/mat4.c\",\n \"start_line\": 2250,\n \"end_line\": 2280,\n \"confidence\": \"likely\",\n \"reason\": \"Mat_VarReadDataAll calls Mat_VarRead4; potential issues may propagate through this function.\",\n \"symbol\": \"Mat_VarReadDataAll\"\n },\n {\n \"file_path\": \"/src/matio_fuzzer.cc\",\n \"start_line\": 36,\n \"end_line\": 46,\n \"confidence\": \"possible\",\n \"reason\": \"LLVMFuzzerTestOneInput directly invokes Mat_VarReadDataAll, which could lead to similar issues.\",\n \"symbol\": \"LLVMFuzzerTestOneInput\"\n },\n {\n \"file_path\": \"/src/matio/src/mat.c\",\n \"start_line\": 2200,\n \"end_line\": 2250,\n \"confidence\": \"possible\",\n \"reason\": \"Function boundary context around Mat_VarReadDataAll likely has relevant initialization issues.\",\n \"symbol\": \"Mat_VarReadDataAll\"\n },\n {\n \"file_path\": \"/src/matio/src/mat4.c\",\n \"start_line\": 395,\n \"end_line\": 400,\n \"confidence\": \"unlikely\",\n \"reason\": \"Specifically the crash line: checking return values might help, but low confidence due to singular nature.\",\n \"symbol\": \"Mat_VarRead4\"\n },\n {\n \"file_path\": \"/src/llvm/projects/compiler-rt/lib/fuzzer/FuzzerMain.cpp\",\n \"start_line\": 0,\n \"end_line\": 20,\n \"confidence\": \"unlikely\",\n \"reason\": \"Main function context unlikely to harbor related issues directly.\",\n \"symbol\": \"main\"\n },\n {\n \"file_path\": \"/src/llvm/projects/compiler-rt/lib/fuzzer/FuzzerLoop.cpp\",\n \"start_line\": 545,\n \"end_line\": 570,\n \"confidence\": \"very unlikely\",\n \"reason\": \"Loop handling fuzz execution; may not relate to uninitialized value directly.\",\n \"symbol\": \"fuzzer::Fuzzer::ExecuteCallback\"\n },\n {\n \"file_path\": \"/src/llvm/projects/compiler-rt/lib/fuzzer/FuzzerDriver.cpp\",\n \"start_line\": 275,\n \"end_line\": 295,\n \"confidence\": \"very unlikely\",\n \"reason\": \"Driver context might not address the core issue of variable initialization.\",\n \"symbol\": \"fuzzer::FuzzerDriver\"\n },\n {\n \"file_path\": \"/src/llvm/projects/compiler-rt/lib/fuzzer/FuzzerLoop.cpp\",\n \"start_line\": 500,\n \"end_line\": 545,\n \"confidence\": \"very unlikely\",\n \"reason\": \"Related to the loop execution but unlikely to impact the specific error directly.\",\n \"symbol\": \"fuzzer::FuzzerDriver\"\n }\n]", "logged": {"json_path": "/home/bug_localization_agents/work_arvo/16857/predictions.json", "batch_json_path": "/home/bug_localization_agents/work_arvo/16857/llm_batch_0001.json", "cumulative_path": "/home/bug_localization_agents/work_arvo/16857/cumulative_predictions.json", "count": 10, "batch": 1}}

---

## Thought

### Executor#0 call: extract_json

```
$ extract_json(response_text="[\n {\n \"file_path\":
\"/src/matio/src/mat4.c\",\n \"start_line\": 292,\n \"end_line\": 312,\n
\"confidence\": \"very likely\",\n \"reason\": \"Uninitialized variable
'tmp' created in Mat_VarRead4 may lead to undefined behavior.\",\n
\"symbol\": \"Mat_VarRead4\"\n },\n {\n \"file_path\":
\"/src/matio/src/mat4.c\",\n \"start_line\": 375,\n \"end_line\": 400,\n
\"confidence\": \"likely\",\n \"reason\": \"Mat_VarRead4 is highlighted
for use of uninitialized values; nearby context lines should also be
```

---

reviewed.\",\n \"symbol\": \"Mat_VarRead4\"\n },\n {\n {\n \"file_path\":
\"/src/matio/src/mat.c\",\n \"start_line\": 2250,\n \"end_line\": 2280,\n
\"confidence\": \"likely\",\n \"reason\": \"Mat_VarReadDataAll calls
Mat_VarRead4; potential issues may propagate through this function.\",\n
\"symbol\": \"Mat_VarReadDataAll\"\n },\n {\n \"file_path\":
\"/src/matio_fuzzer.cc\",\n \"start_line\": 36,\n \"end_line\": 46,\n
\"confidence\": \"possible\",\n \"reason\": \"LLVMFuzzerTestOneInput
directly invokes Mat_VarReadDataAll, which could lead to similar
issues.\",\n \"symbol\": \"LLVMFuzzerTestOneInput\"\n },\n {\n
\"file_path\": \"/src/matio/src/mat.c\",\n \"start_line\": 2200,\n
\"end_line\": 2250,\n \"confidence\": \"possible\",\n \"reason\":
\"Function boundary context around Mat_VarReadDataAll likely has relevant
initialization issues.\",\n \"symbol\": \"Mat_VarReadDataAll\"\n }\n]",
merge_with_last=false)

---

## Thought

### Executor#0 result: extract_json

### Observation

{"ok": true, "json": {"llm_predictions": [{"file_path": "/src/matio/src/mat4.c", "start_line": 292, "end_line": 312, "confidence": "very likely", "reason": "Uninitialized variable 'tmp' created in Mat_VarRead4 may lead to undefined behavior.", "symbol": "Mat_VarRead4"}, {"file_path": "/src/matio/src/mat4.c", "start_line": 375, "end_line": 400, "confidence": "likely", "reason": "Mat_VarRead4 is highlighted for use of uninitialized values; nearby context lines should also be reviewed.", "symbol": "Mat_VarRead4"}, {"file_path": "/src/matio/src/mat.c", "start_line": 2250, "end_line": 2280, "confidence": "likely", "reason": "Mat_VarReadDataAll calls Mat_VarRead4; potential issues may propagate through this function.", "symbol": "Mat_VarReadDataAll"}, {"file_path": "/src/matio_fuzzer.cc", "start_line": 36, "end_line": 46, "confidence": "possible", "reason": "LLVMFuzzerTestOneInput directly invokes Mat_VarReadDataAll, which could lead to similar issues.", "symbol": "LLVMFuzzerTestOneInput"}, {"file_path": "/src/matio/src/mat.c", "start_line": 2200, "end_line": 2250, "confidence": "possible", "reason": "Function boundary context around Mat_VarReadDataAll likely has relevant initialization issues.", "symbol": "Mat_VarReadDataAll"}]}, "raw": "[\n {\"llm_predictions\": [{\"file_path\": \"/src/matio/src/mat4.c\", \"start_line\": 292, \"end_line\": 312, \"confidence\": \"very likely\", \"reason\": \"Uninitialized variable 'tmp' created in Mat_VarRead4 may lead to undefined behavior.\", \"symbol\": \"Mat_VarRead4\"}, {\"file_path\": \"/src/matio/src/mat4.c\", \"start_line\": 375, \"end_line\": 400, \"confidence\": \"likely\", \"reason\": \"Mat_VarRead4 is highlighted for use of uninitialized values; nearby context lines should also be reviewed.\", \"symbol\": \"Mat_VarRead4\"}, {\"file_path\": \"/src/matio/src/mat.c\", \"start_line\": 2250, \"end_line\": 2280, \"confidence\": \"likely\", \"reason\": \"Mat_VarReadDataAll calls Mat_VarRead4; potential issues may propagate through this function.\", \"symbol\": \"Mat_VarReadDataAll\"}, {\"file_path\": \"/src/matio_fuzzer.cc\", \"start_line\": 36, \"end_line\": 46, \"confidence\": \"possible\", \"reason\": \"LLVMFuzzerTestOneInput directly invokes Mat_VarReadDataAll, which could lead to similar issues.\", \"symbol\": \"LLVMFuzzerTestOneInput\"}, {\"file_path\": \"/src/matio/src/mat.c\", \"start_line\": 2200, \"end_line\": 2250, \"confidence\": \"possible\", \"reason\": \"Function boundary context around Mat_VarReadDataAll likely has relevant initialization issues.\", \"symbol\": \"Mat_VarReadDataAll\"}]}", "json_path": "/home/bug_localization_agents/work_arvo/16857/llm_output.tmp.json"}

---

## Thought

### Executor#0 call: compare_llm_metrics

```
$
compare_llm_metrics(llm_json_path="/home/bug_localization_agents/work_arvo
16857/llm_batch_0001.json",
chunks_json_path="/home/bug_localization_agents/work_arvo/16857
case_chunks.json",
diff_index_json_path="/home/bug_localization_agents/work_arvo/16857
diff_index.json", line_slack=0)
```

---

## Thought

### Executor#0 result: compare_llm_metrics

### Observation

{"ok": true, "detection_rate": 0.0625, "localization_rate": 0.03669724770642202,
"strict_localization_rate": 0.0, "inputs": {"llm_json_path":
"/home/bug_localization_agents/work_arvo/16857/llm_batch_0001.json", "chunks_json_path":
"/home/bug_localization_agents/work_arvo/16857/case_chunks.json", "diff_index_json_path":
"/home/bug_localization_agents/work_arvo/16857/diff_index.json"}, "detection": {"total_lines":
16, "num_detected and diff": 1, "rate": 0.0625, "updated_chunks": true}, "localization": {"total_lines":
218, "matched_lines": 8, "newly_matched": 8, "rate": 0.03669724770642202}, "strict_localization":
{"total_groups": 103, "matched_groups": 0, "rate": 0.0, "saved_snapshot_error": "compare_history"}}

---

## Thought

### Executor#0 call: view_source

```
$ view_source(file_path="/src/matio/src/mat4.c", start_line=252,
end_line=352)
```

---

## Thought

### Executor#0 result: view_source

### Observation

{"ok": true, "file_path": "/src/matio/src/mat4.c", "start_line": 252, "end_line": 352, "content": "252 | <
sparse->j[c|| + 1] && | < sparse->ndata; j++ } {\n253 fwrite(re + j*stride, stride, 1,
(FILE*)mat->fp);\n254 }\n255 }\n256 fwrite(&tmp, stride, 1, (FILE*)mat->fp);\n257 for ( i = 0; i <
sparse->njc - 1; i++ ) {\n258 for ( j = sparse->jc[i];\n259 j < sparse->jc[i + 1] && j < sparse->ndata;
j++ } {\n260 fwrite(im + j*stride, stride, 1, (FILE*)mat->fp);\n261 }\n262 }\n263 } else {\n264 char
*data = (char*)sparse->data;\n265 for ( i = 0; i < sparse->njc - 1; i++ ) {\n266 fwrite(data + j*stride,
stride, 1, (FILE*)mat->fp);\n269 }\n270 }\n271 }\n272 fwrite(&tmp, stride, 1, (FILE*)mat->fp);\n273
break;\n274 }\n275 default:\n276 break;\n277 }\n278 }\n279 return 0;\n280 }\n281 }\n282 /** @if
mat_devman\n283 * @brief Reads the data of a version 4 MAT file variable\n284 *\n285 * @ingroup
mat_internal\n286 * @param mat MAT file pointer\n287 * @param matvar MAT variable pointer to read
the data\n288 * @endif\n289 */\n290 void\n291 Mat_VarRead4(mat_t *mat,matvar_t *matvar)\n292
{\n293 int err;\n294 size_t nelems = 1;\n295 \n296 err = SafeMulDims(matvar, &nelems);\n297 if ( err )
{\n298 Mat_Critical(\"Integer multiplication overflow\");\n299 return;\n300 }\n301 \n302
(void)fseek((FILE*)mat->fp,matvar->internal->datapos,SEEK_SET);\n303 \n304 switch (
matvar->class_type ) {\n305 case MAT_C_DOUBLE:\n306 matvar->data_size = sizeof(double);\n307 err
= SafeMul(&matvar->nbytes, nelems, matvar->data_size);\n308 if ( err ) {\n309 Mat_Critical(\"Integer
multiplication overflow\");\n310 return;\n311 }\n312 \n313 if ( matvar->isComplex ) {\n314
mat_complex_split_t *complex_data = ComplexMalloc(matvar->nbytes);\n315 if ( NULL !=
complex_data ) {\n316 matvar->data = complex_data;\n317 ReadDoubleData(mat,
(double*)complex_data->Re, matvar->data_type, nelems);\n318 ReadDoubleData(mat,
(double*)complex_data->Im, matvar->data_type, nelems);\n319 }\n320 else {\n321
Mat_Critical(\"Couldn't allocate memory for the complex data\");\n322 }\n323 } else {\n324
matvar->data = malloc(matvar->nbytes);\n325 if ( NULL != matvar->data ) {\n326
ReadDoubleData(mat, (double*)matvar->data, matvar->data_type, nelems);\n327 }\n328 else {\n329
Mat_Critical(\"Couldn't allocate memory for the data\");\n330 }\n331 }\n332 /* Update data type to
match format of matvar */\n333 matvar->data_type = MAT_T_DOUBLE;\n334 break;\n335 case
MAT_C_CHAR:\n336 matvar->data_size = 1;\n337 matvar->nbytes = nelems;\n338 matvar->data =
malloc(matvar->nbytes);\n339 if ( NULL != matvar->data ) {\n340 ReadUInt8Data(mat,
(mat_uint8_t*)matvar->data, matvar->data_type, nelems);\n341 }\n342 else {\n343
Mat_Critical(\"Couldn't allocate memory for the data\");\n344 }\n345 matvar->data_type =
MAT_T_UINT8;\n346 break;\n347 case MAT_C_SPARSE:\n348 matvar->data_size =
sizeof(mat_sparse_t);\n349 matvar->data = malloc(matvar->data_size);\n350 if ( NULL !=
matvar->data ) {\n351 double tmp;\n352 int i;\n", "error": null}

---

## Thought

### Executor#0 call: view_source

```
$ view_source(file_path="/src/matio/src/mat.c", start_line=2220,
end_line=2280)
```

---

## Thought

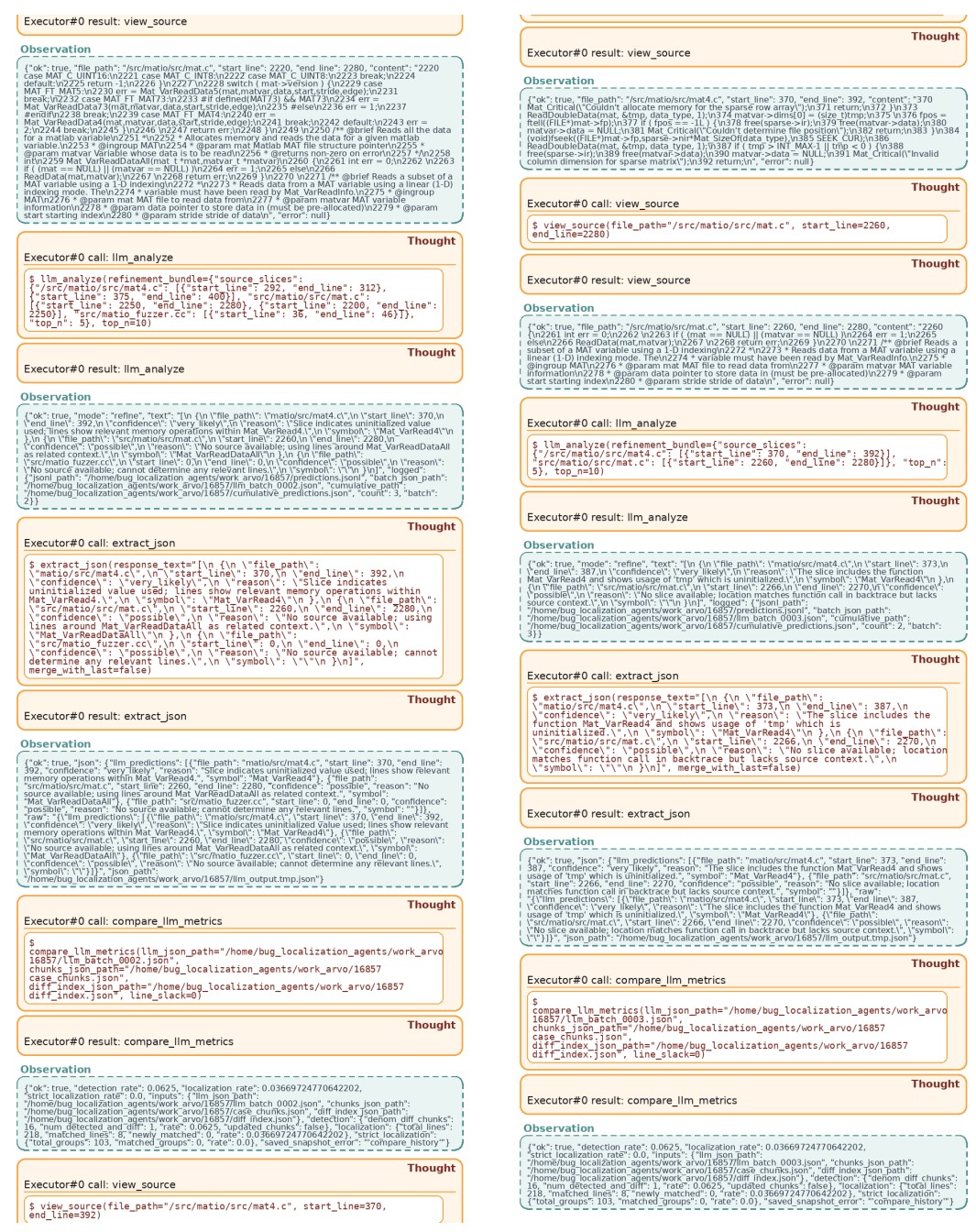

Figure 14: GPT-4o mini Divergence Tracing for case 16857.

**Thought**

Planner note

**Observation**

```
You are the PLANNER for ARVO bug localization. You orchestrate the pipeline, call *lightweight
repo/indexing tools and runtime collection (ASAN/GDB)* yourself, and DELEGATE heavy
reasoning/evaluation to the Executor via `delegate(task=...)`.

## Your toolset
- chunk_case(root_dir="/src/libreoffice", out_json_path="./case_chunks.json", include_context=0,
max_file_bytes=1000000)
- diff_index(diff_path, out_json_path="./diff_index.json")
- mark_diff(diff_json_path="./diff_index.json", chunks_json_path="./case_chunks.json")
- run_san()
- run_gdb(binary_path="./vuln")
- delegate(task=<string>)
- giveup()

## Responsibilities
- OWN orchestration and state: start container once, choose REPO_ROOT, build chunks & diff index
once, mark diff on chunks once (unless inputs changed).
- YOU run ASAN/GDB and cache crash log in environment state.
- For LLM reasoning & evaluation: DELEGATE to the Executor with a clear, structured task that includes:
• What to analyze (use cached crash_log → llm_analyze → extract_json),
• Which files to use (paths to diff index, case chunks.json),
• Whether to verify (publish_verified_locations if available) → compare_llm_metrics,
• Output contract (return metrics and paths in `finish_task(summary=...)`).
- Aggregate results. **Do not call `giveup()` after a single delegation.** Prefer at least 2-3 refinement
rounds unless clearly hopeless.

## Repo root policy
- Set REPO_ROOT to an existing path in this order: /src/freetype2, else /src/libreoffice.
- Use that REPO_ROOT consistently when asking the Executor to verify locations.

## Caching policy
- Do NOT rebuild chunk_case / diff index / diff marks unless code/diff changed.
- Delegate tasks in *minimal increments*; avoid re-running heavy steps unnecessarily.
```

**Thought**

Planner note

**Observation**

**Thought**

Planner call: chunk_case

```
$ chunk_case(root_dir="/src/libreoffice",
out_json_path="/home/bug_localization_agents/work_arvo/20112
case_chunks.json")
```

**Thought**

Planner result: chunk_case

**Observation**

```
{"ok": true, "root_dir": "/src/libreoffice", "json_path":
"/home/bug_localization_agents/work_arvo/20112/case_chunks.json", "num_files": 472, "num_chunks":
5505}
```

**Thought**

Planner call: diff_index

```
$
diff_index(diff_path="/home/bug_localization_agents/ARVO-Meta/patches
20112.diff"
out_json_path="/home/bug_localization_agents/work_arvo/20112
diff_index.json")
```

**Thought**

Planner result: diff_index

**Thought**

Planner call: mark_diff

**Observation**

```
{"ok": true, "json_path": "/home/bug_localization_agents/work_arvo/20112/diff_index.json", "num_files":
1, "num_hunks": 0, "num_per_line": 26}
```

**Thought**

Planner call: mark_diff

```
$
mark_diff(diff_json_path="/home/bug_localization_agents/work_arvo/20112
diff_index.json",
chunks_json_path="/home/bug_localization_agents/work_arvo/20112
case_chunks.json")
```

**Thought**

Planner result: mark_diff

**Observation**

```
{"ok": true, "chunks_json_path": "/home/bug_localization_agents/work_arvo/20112/case_chunks.json",
"num_chunks": 5505, "num_chunks_with_diff": 2}
```

**Thought**

Planner call: run_san

```
$ run_san()
```

**Thought**

Planner result: run_san

**Observation**

```
{"ok": true, "asan_output": {"ok": true, "asan_output": "INFO: Seed: 2174243374\r\nINFO: Loaded 1
modules (10308 inline 8-bit counters): 10308[0xa201f0, 0xa22a34), \r\nINFO: Loaded 1 PC tables
(10308 PCs): 10308 [0xa22a38,0xa4ae78), \r\n/out/fuzz_json_decode: Running 1 inputs 1 time(s)
each.\r\nRunning: /tmp/poc\r\n\u001b[1m\u001b[0m #0 0x6f4649 in jumpOverRec
use-of-uninitialized-value\r\n\u001b[1m\u001b[0m #0 0x6f4649 in jumpOverRec
/src/open62541/src/ua_types_encoding_json.c:2353:16\r\n #1 0x6f40ed in jumpOverRec
/src/open62541/src/ua_types_encoding_json.c:2339:16\r\n #2 0x6f4048 in jumpOverRec
/src/open62541/src/ua_types_encoding_json.c:2337:16\r\n #3 0x6e8ccb in jumpOverObject decodeJson
/src/open62541/src/ua_types_encoding_json.c:2370:5\r\n #4 0x6e8ccb in ExtensionObject decodeJson
/src/open62541/src/ua_types_encoding_json.c:2838:13\r\n #5 0x6f5da7 in Array_decodeJson_internal
/src/open62541/src/ua_types_encoding_json.c:3142:15\r\n #6 0x6d8ea7 in decodeFields
/src/open62541/src/ua_types_encoding_json.c:3096:23\r\n #7 0x6eb7bc in Variant_decodeJson
/src/open62541/src/ua_types_encoding_json.c\r\n #8 0x6ed5d4 in UA_decodeJson
/src/open62541/src/ua_types_encoding_json.c:3321:11\r\n #9 0x524bb4 in LLVMFuzzerTestOneInput
/src/open62541/tests/fuzz/fuzz_json_decode.cc:26:28\r\n #10 0x45ab51 in
fuzzer::Fuzzer::ExecuteCallback(unsigned char const*, unsigned long)
/src/llvm/projects/compiler-rt/lib/fuzzer/FuzzerLoop.cpp:554:15\r\n #11 0x445381 in
fuzzer::RunOneTest(fuzzer::Fuzzer*, char const*, unsigned long)
/src/llvm/projects/compiler-rt/lib/fuzzer/FuzzerDriver.cpp:292:6\r\n #12 0x44b09e in
fuzzer::FuzzerDriver(int*, char***, int (*)(unsigned char const*, unsigned long))
/src/llvm/projects/compiler-rt/lib/fuzzer/FuzzerDriver.cpp:775:9\r\n #13 0x474a22 in main
/src/llvm/projects/compiler-rt/lib/fuzzer/FuzzerMain.cpp:19:10\r\n #14 0x7fd4e6c9283f in
__libc_start_main /lib/x86_64-linux-gnu/libc.so.6+0x2083f)\r\n #15 0x41e828 in _start
/out/fuzz_json_decode+0x41e828)\r\n\nDEDUP_TOKEN:
jumpOverRec--jumpOverRec\r\n\u001b[1m\u001b[35mUninitialized value was created
by a heap allocation\u001b[1m\u001b[0m\r\n #0 0x4d552d in malloc
/src/llvm/projects/compiler-rt/lib/msan/msan_interceptors.cpp:916:3\r\n #1 0x7086cb in
UA_memoryManager_malloc /src/open62541/tests/fuzz/custom_memory_manager.c:132:18\r\n #2
0x6ed176 in UA_decodeJson /src/open62541/src/ua_types_encoding_json.c:3296:39\r\n #3 0x524bb4 in
LLVMFuzzerTestOneInput /src/open62541/tests/fuzz/fuzz_json_decode.cc:26:28\r\n #4 0x45ab51 in
fuzzer::Fuzzer::ExecuteCallback(unsigned char const*, unsigned long)
/src/llvm/projects/compiler-rt/lib/fuzzer/FuzzerLoop.cpp:554:15\r\n #5 0x445381 in
fuzzer::RunOneTest(fuzzer::Fuzzer*, char const*, unsigned long)
/src/llvm/projects/compiler-rt/lib/fuzzer/FuzzerDriver.cpp:292:6\r\n #6 0x44b09e in
fuzzer::FuzzerDriver(int*, char***, int (*)(unsigned char const*, unsigned long))
/src/llvm/projects/compiler-rt/lib/fuzzer/FuzzerDriver.cpp:775:9\r\n #7 0x474a22 in main
/src/llvm/projects/compiler-rt/lib/fuzzer/FuzzerMain.cpp:19:10\r\n #8 0x7fd4e6c9283f in
__libc_start_main /lib/x86_64-linux-gnu/libc.so.6+0x2083f)\r\n\nDEDUP_TOKEN:
malloc--UA_memoryManager_malloc--UA_decodeJson\r\nSUMMARY: MemorySanitizer:
use-of-uninitialized-value /src/open62541/src/ua_types_encoding_json.c:2353:16 in
jumpOverRec\r\n\nUnique heap origins: 2?\r\n\nStack depot allocated bytes: 1952\r\n\nUnique origin
histories: 3\r\n\nStack depot allocated bytes: 72\r\n\nExiting", "binary_path": "/out/fuzz_json_decode",
"cid": "8f87ddaf8c72"}}
```

**Thought**

Planner call: run_gdb

```
$ run_gdb(binary_path="/out/fuzz_json_decode")
```

**Thought**

Planner result: run_gdb

**Observation**

```
{"ok": true, "warning": "Failed to install gdb inside the container."}
```

**Thought**

Planner call: delegate

```
$ delegate(task="llm_analyze(crash_log='INFO: Seed: 2174243374'\nINFO:
Loaded 1 modules (10308 inline 8-bit counters): 10308 [0xa201f0,
0xa22a34), \\INFO: Loaded 1 PC tables (10308 PCs): 10308
[0xa22a38,0xa4ae78), \\n/out/fuzz_json_decode: Running 1 inputs 1 time(s)
each.\\nRunning: /tmp/poc\\\n\u001b[1m\u001b[31m=3363==WARNING:
MemorySanitizer: use-of-uninitialized-value\\n\\u001b[1m\u001b[0m #0
0x6f4649 in jumpOverRec
/src/open62541/src/ua_types_encoding_json.c:2353:16\\n #1 0x6f40ed in
jumpOverRec /src/open62541/src/ua_types_encoding_json.c:2339:16\\n #2
0x6f4048 in jumpOverRec
/src/open62541/src/ua_types_encoding_json.c:2337:16\\n #3 0x6e8ccb in
jumpOverObject /src/open62541/src/ua_types_encoding_json.c:2370:5\\n #4
0x6e8ccb in ExtensionObject decodeJson
/src/open62541/src/ua_types_encoding_json.c:2838:13\\n #5 0x6f5da7 in
Array_decodeJson_internal
/src/open62541/src/ua_types_encoding_json.c:3142:15\\n #6 0x6d8ea7 in
decodeFields /src/open62541/src/ua_types_encoding_json.c:3096:23\\n #7
0x6eb7bc in Variant_decodeJson
/src/open62541/src/ua_types_encoding_json.c\\n #8 0x6ed5d4 in
UA_decodeJson /src/open62541/src/ua_types_encoding_json.c:3321:11\\n #9
0x524bb4 in LLVMFuzzerTestOneInput
/src/open62541/tests/fuzz/fuzz_json_decode.cc:26:28\\n #10 0x45ab51 in
fuzzer::Fuzzer::ExecuteCallback(unsigned char const*, unsigned long)
/src/llvm/projects/compiler-rt/lib/fuzzer/FuzzerLoop.cpp:554:15\\n #11
0x445381 in fuzzer::RunOneTest(fuzzer::Fuzzer*, char const*, unsigned
long) /src/llvm/projects/compiler-rt/lib/fuzzer/FuzzerDriver.cpp:292:6\\n
#12 0x44b09e in fuzzer::FuzzerDriver(int*, char***, int (*)(unsigned char
const*, unsigned long))
/src/llvm/projects/compiler-rt/lib/fuzzer/FuzzerDriver.cpp:775:9\\n #13
0x474a22 in main
/src/llvm/projects/compiler-rt/lib/fuzzer/FuzzerMain.cpp:19:10\\n #14
0x7fd4e6c9283f in __libc_start_main
(/lib/x86_64-linux-gnu/libc.so.6+0x2083f)\\n #15 0x41e828 in _start
(/out/fuzz_json_decode+0x41e828)\\n\\nDEDUP_TOKEN:
jumpOverRec--jumpOverRec\\n
\\u001b[1m\\u001b[35mUninitialized value was created by a heap
allocation\\u001b[1m\\u001b[0m\\n #0 0x4d552d in malloc
/src/llvm/projects/compiler-rt/lib/msan/msan_interceptors.cpp:916:3\\n #1
0x7086cb in UA_memoryManager_malloc
/src/open62541/tests/fuzz/custom_memory_manager.c:132:18\\n #2 0x6ed176 in
UA_decodeJson /src/open62541/src/ua_types_encoding_json.c:3296:39\\n #3
```

```
UA_decodeJson /src/open62541/src/ua_types_encoding_json.c:3290.39\\n #3
0x524bb4 in LLVMFuzzerTestOneInput
/src/open62541/tests/fuzz/fuzz_json_decode.cc:26:28\\n #4 0x45ab51 in
fuzzer::Fuzzer::ExecuteCallback(unsigned char const*, unsigned long)
/src/llvm/projects/compiler-rt/lib/fuzzer/FuzzerLoop.cpp:554:15\\n #5
0x445381 in fuzzer::RunOneTest(fuzzer::Fuzzer*, char const*, unsigned
long) /src/llvm/projects/compiler-rt/lib/fuzzer/FuzzerDriver.cpp:292:6\\n
#6 0x44b09e in fuzzer::FuzzerDriver(int*, char***, int (*)(unsigned char
const*, unsigned long))
/src/llvm/projects/compiler-rt/lib/fuzzer/FuzzerDriver.cpp:775:9\\n #7
0x474a22 in main
/src/llvm/projects/compiler-rt/lib/fuzzer/FuzzerMain.cpp:19:10\\n #8
0x7fd4e6c92834 in __libc_start_main
(/lib/x86_64-linux-gnu/libc.so.6+0x2083f)\\n\\nDEDUP_TOKEN:
malloc--UA_memoryManager_malloc--UA_decodeJson\\n\\nSUMMARY: MemorySanitizer:
use-of-uninitialized-value
/src/open62541/src/ua_types_encoding_json.c:2353:16 in
jumpOverRec\\n\\nUnique heap origins: 27\\n\\nStack depot allocated bytes:
1952\\n\\nUnique origin histories: 3\\n\\nHistory depot allocated bytes:
72\\n\\nExiting', 'verified_locations=None', 'repo_root=/src
libreoffice', 'compare_llm_metrics(diff_index_json_path=".
diff_index.json", llm_json_path=None, chunks_json_path=".
case_chunks.json", line_slack=0)')")
```

**Thought**

Planner result: delegate

**Observation**

"no improvement in 3 rounds, end executor"

**Thought**

Planner call: giveup

```
$ giveup(confirm=true)
```

**Thought**

Planner result: giveup

**Observation**

{"success": true}

**Thought**

Executor#0 note

**Observation**

You are the EXECUTOR for ARVO bug localization. You run heavy tools and MUST call
finish_task(summary=...) to return results.
TOOLS:
- llm_analyze(crash_log=<string> | refinement_bundle=..., top_n=<int>)
- extract_json(response_text, merge_with_last=<bool>)
- view_source(file_path, start_line=None, end_line=None, max_bytes=200000)
- grep_source(pattern, root="/src", max_results=200)
- copy_out(path, out_path=None)
- compare_llm_metrics(diff_index_json_path=None, llm_json_path=None, chunks_json_path=None,
line_slack=0)
- finish_task(summary=<string>)

RUNTIME OWNERSHIP:
- Do NOT run ASAN or GDB. The Planner owns runtime collection and has cached `crash_log`.
- You MAY call llm_analyze() with NO arguments to consume the cached crash_log from the
environment.

REFINEMENT (augment-only, optional):
- Use view_source/grep_source to fetch code slices for promising files/functions.
- Then call llm_analyze(refinement_bundle={"source_slices": {...}}, top_n=<M>) to GENERATE a NEW
batch of candidates from these slices only.
- This refinement MUST NOT narrow, edit, or reuse previous candidates; it produces up to exactly <M>
fresh items.
- After refinement, call extract_json(response_text, merge_with_last=false) so the new batch stays
separate from the first batch.

COMPARISON:
- You may call compare_llm_metrics multiple times. Evaluate the first JSON snapshot, optionally refine
to produce a new batch, then evaluate again.
- Provide the correct llm_json_path for each compare; do not call compare_llm_metrics without it.

NO-OP/INVALID CALLS:
- Do NOT re-run llm_analyze repeatedly for the same inputs (at most one augment round).
- Do NOT open files >200KB; keep total view_source ≤5 and grep_source ≤2.

FINISH CONTRACT:
- finish_task(summary=...) must be a single-line, semicolon-separated key=val list including:
ok=<t/f>false=; detection_rate=<float>; localization_rate=<float>; strict_localization_rate=<float>;
improved=<true/false>; prev=<float>; curr=<float>;
diff_index_json_path=<path>; chunks_json_path=<path>; llm_json_path=<path or none>;
gdb=unknown; bin=unknown

**Thought**

Executor#0 note

**Observation**

You have been delegated a task:
Use the cached crash_log produced by the Planner. Analyze, optionally augment with source views,
evaluate twice, then return metrics.

Steps:
1) llm_analyze() # no args → consume cached crash_log
2) extract_json(response_text, merge_with_last=false) # first JSON snapshot (batch A)
3) compare_llm_metrics(diff_index_json_path=".diff_index.json",
llm_json_path=<batch A JSON>,
chunks_json_path="./case_chunks.json",
line_slack=0)
4) SOURCE PEEK (optional, before second compare):
- Parse crash log for file paths (e.g., "/src/...:<line>") and function names
- Include top suspects from batch A; focus on crash stack (top 3-5 frames)
- For up to TOP K=5 candidates (dedupe by file):
- If file_path known: view_source(file_path, start_line-40, end_line+40, max_bytes=200000)
- Else: grep_source(symbol/function, root="/src", max_results=20) and view_source best match ±40
Redlines: ≤5 view_source and ≤2 grep_source; skip files >200KB
5) REFINEMENT (augment-only, one pass):
- llm_analyze(refinement_bundle={"source_slices": {...}}, top_n=<M>)
# Generates a NEW batch (batch B) from slices only; does not modify batch A
- extract_json(response_text, merge_with_last=false) # keep batch B separate
6) compare_llm_metrics(diff_index_json_path="./diff_index.json",
llm_json_path=<batch B JSON if step 5 ran; else re-use batch A>,
chunks_json_path="./case_chunks.json",
line_slack=0)
7) finish_task(summary="ok=<...>; detection_rate=<...>; localization_rate=<...>;
strict_localization_rate=<...>; improved=<...>; prev=<...>; curr=<...>;
diff_index_json_path=./diff_index.json; chunks_json_path=./case_chunks.json; llm_json_path=<...>;
gdb=unknown; bin=unknown")

**Thought**

Executor#0 call: llm_analyze

```
$ llm_analyze()
```

**Thought**

Executor#0 result: llm_analyze

**Observation**

**Observation**

{"ok": true, "mode": "crash", "text": "[\n {\n \"file_path\":
\"/src/open62541/src/ua_types_encoding_json.c\",\n \"start_line\": 2335,\n \"end_line\": 2359,\n
\"confidence\": \"very_likely\",\n \"reason\": \"Involves several calls leading to uninitialized memory use
in jumpOverRec, check state before dereferencing.\",\n \"symbol\": \"jumpOverRec\"\n },\n {\n
\"file_path\": \"/src/open62541/src/ua_types_encoding_json.c\",\n \"start_line\": 2830,\n \"end_line\":
2850,\n \"confidence\": \"very_likely\",\n \"reason\": \"ExtensionObject_decodeJson is implicated in the
call chain. Inspect for memory handling issues.\",\n \"symbol\": \"ExtensionObject_decodeJson\"\n },\n
{\n \"file_path\": \"/src/open62541/src/ua_types_encoding_json.c\",\n \"start_line\": 3090,\n \"end_line\":
3110,\n \"confidence\": \"likely\",\n \"reason\": \"Potential issues in Array_decodeJson internal due to
uninitialized values propagating.\",\n \"symbol\": \"Array_decodeJson_internal\"\n },\n {\n \"file_path\":
\"/src/open62541/src/ua_types_encoding_json.c\",\n \"start_line\": 3320,\n \"end_line\": 3340,\n
\"confidence\": \"possible\",\n \"reason\": \"Possible fault in UA_decodeJson where values may be
incorrectly initialized or handled.\",\n \"symbol\": \"UA_decodeJson\"\n },\n {\n \"file_path\":
\"/src/open62541/tests/fuzz/fuzz_json_decode.cc\",\n \"start_line\": 26,\n \"end_line\": 34,\n
\"confidence\": \"possible\",\n \"reason\": \"The test input handling can influence memory allocation.
Check how inputs interact with decode functions.\",\n \"symbol\": \"LLVMFuzzerTestOneInput\"\n },\n {\n
\"file_path\": \"/src/open62541/src/ua_types_encoding_json.c\",\n \"start_line\": 3290,\n \"end_line\":
3310,\n \"confidence\": \"unlikely\",\n \"reason\": \"UA_decodeJson indirectly uses malloc; check
interactions with memory allocation semantics.\",\n \"symbol\": \"UA_decodeJson\"\n },\n {\n
\"file_path\": \"/src/open62541/tests/fuzz/custom_memory_manager.c\",\n \"start_line\": 132,\n
\"end_line\": 142,\n \"confidence\": \"unlikely\",\n \"reason\": \"Custom memory allocator may have
uninitialized states affecting memory safety.\",\n \"symbol\": \"UA_memoryManager_malloc\"\n },\n {\n
\"file_path\": \"/src/llvm/projects/compiler-rt/lib/fuzzer/FuzzerLoop.cpp\",\n \"start_line\": 550,\n
\"end_line\": 570,\n \"confidence\": \"unlikely\",\n \"reason\": \"Handles the fuzzing callback; investigate
how fuzz inputs are processed leading to memory issues.\",\n \"symbol\": \"Fuzzer::ExecuteCallback\"\n
},\n {\n \"file_path\": \"/src/llvm/projects/compiler-rt/lib/fuzzer/FuzzerDriver.cpp\",\n \"start_line\": 290,\n
\"end_line\": 310,\n \"confidence\": \"unlikely\",\n \"reason\": \"Fuzzer driver may be responsible for
initiating problematic tests. Assess test case input methods.\",\n \"symbol\": \"Fuzzer::FuzzerDriver\"\n
},\n {\n \"file_path\": \"/src/llvm/projects/compiler-rt/lib/fuzzer/FuzzerMain.cpp\",\n \"start_line\": 19,\n
\"end_line\": 29,\n \"confidence\": \"unlikely\",\n \"reason\": \"Main entry point handles overall fuzzer
lifecycle; indirect impact on memory allocation practices.\",\n \"symbol\": \"main\"\n }]", "logged":
{"json_path": "/home/bug_localization_agents/work_arvo20112/predictions.json", "batch_json_path":
"/home/bug_localization_agents/work_arvo20112/llm_batch_0001.json", "cumulative_path":
"/home/bug_localization_agents/work_arvo20112/cumulative_predictions.json", "count": 10, "batch":
1}}

**Thought**

Executor#0 call: extract_json

```
$ extract_json(response_text="[\n {\n \"file_path\":
\"/src/open62541/src/ua_types_encoding_json.c\",\n \"start_line\": 2335,\n
\"end_line\": 2359,\n \"confidence\": \"very_likely\",\n \"reason\":
\"Involves several calls leading to uninitialized memory use in
jumpOverRec, check state before dereferencing.\",\n \"symbol\":
\"jumpOverRec\"\n },\n {\n \"file_path\":
\"/src/open62541/src/ua_types_encoding_json.c\",\n \"start_line\": 2830,\n
\"end_line\": 2850,\n \"confidence\": \"very_likely\",\n \"reason\":
\"ExtensionObject_decodeJson is implicated in the call chain. Inspect for
memory handling issues.\",\n \"symbol\": \"ExtensionObject_decodeJson\"\n
},\n {\n \"file_path\": \"/src/open62541/src/ua_types_encoding_json.c\",\n
\"start_line\": 3090,\n \"end_line\": 3110,\n \"confidence\":
\"likely\",\n \"reason\": \"Potential issues in Array_decodeJson_internal
due to uninitialized values propagating.\",\n \"symbol\":
\"Array_decodeJson_internal\"\n },\n {\n \"file_path\":
\"/src/open62541/src/ua_types_encoding_json.c\",\n \"start_line\": 3320,\n
\"end_line\": 3340,\n \"confidence\": \"possible\",\n \"reason\":
\"Possible fault in UA_decodeJson where values may be incorrectly
initialized or handled.\",\n \"symbol\": \"UA_decodeJson\"\n },\n {\n
\"file_path\": \"/src/open62541/tests/fuzz/fuzz_json_decode.cc\",\n
\"start_line\": 26,\n \"end_line\": 34,\n \"confidence\": \"possible\",\n
\"reason\": \"The test input handling can influence memory allocation.
Check how inputs interact with decode functions.\",\n \"symbol\":
\"LLVMFuzzerTestOneInput\"\n },\n {\n \"file_path\":
\"/src/open62541/src/ua_types_encoding_json.c\",\n \"start_line\": 3290,\n
\"end_line\": 3310,\n \"confidence\": \"unlikely\",\n \"reason\":
\"UA_decodeJson indirectly uses malloc; check interactions with memory
allocation semantics.\",\n \"symbol\": \"UA_decodeJson\"\n },\n {\n
\"file_path\": \"/src/open62541/tests/fuzz/custom_memory_manager.c\",\n
\"start_line\": 132,\n \"end_line\": 142,\n \"confidence\":
\"unlikely\",\n \"reason\": \"Custom memory allocator may have
uninitialized states affecting memory safety.\",\n \"symbol\":
\"UA_memoryManager_malloc\"\n },\n {\n \"file_path\":
\"/src/llvm/projects/compiler-rt/lib/fuzzer/FuzzerLoop.cpp\",\n
\"start_line\": 550,\n \"end_line\": 570,\n \"confidence\":
\"unlikely\",\n \"reason\": \"Handles the fuzzing callback; investigate
how fuzz inputs are processed leading to memory issues.\",\n \"symbol\":
\"Fuzzer::ExecuteCallback\"\n },\n {\n \"file_path\":
\"/src/llvm/projects/compiler-rt/lib/fuzzer/FuzzerDriver.cpp\",\n
\"start_line\": 290,\n \"end_line\": 310,\n \"confidence\":
\"unlikely\",\n \"reason\": \"Fuzzer driver may be responsible for
initiating problematic tests. Assess test case input methods.\",\n
\"symbol\": \"Fuzzer::FuzzerDriver\"\n },\n {\n \"file_path\":
\"/src/llvm/projects/compiler-rt/lib/fuzzer/FuzzerMain.cpp\",\n
\"start_line\": 19,\n \"end_line\": 29,\n \"confidence\": \"unlikely\",\n
\"reason\": \"Main entry point handles overall fuzzer lifecycle; indirect
impact on memory allocation practices.\",\n \"symbol\": \"main\"\n }]",
merge_with_last=false)
```

**Thought**

Executor#0 result: extract_json

**Observation**

{"ok": true, "json": {"llm_predictions": [{"file_path": "/src/open62541/src/ua_types_encoding_json.c",
"start_line": 2335, "end_line": 2359, "confidence": "very_likely", "reason": "Involves several calls
leading to uninitialized memory use in jumpOverRec, check state before dereferencing.", "symbol":
"jumpOverRec"}, {"file_path": "/src/open62541/src/ua_types_encoding_json.c", "start_line": 2830,
"end_line": 2850, "confidence": "very_likely", "reason": "ExtensionObject_decodeJson is implicated in
the call chain. Inspect for memory handling issues.", "symbol": "ExtensionObject_decodeJson"},
{"file_path": "/src/open62541/src/ua_types_encoding_json.c", "start_line": 3090, "end_line": 3110,
"confidence": "likely", "reason": "Potential issues in Array_decodeJson internal due to uninitialized
values propagating.", "symbol": "Array_decodeJson_internal"}, {"file_path":
"/src/open62541/src/ua_types_encoding_json.c", "start_line": 3320, "end_line": 3340, "confidence":
"possible", "reason": "Possible fault in UA_decodeJson where values may be incorrectly initialized or
handled.", "symbol": "UA_decodeJson"}, {"file_path": "/src/open62541/tests/fuzz/fuzz_json_decode.cc",
"start_line": 26, "end_line": 34, "confidence": "possible", "reason": "The test input handling can
influence memory allocation. Check how inputs interact with decode functions.", "symbol":
"LLVMFuzzerTestOneInput"}, {"file_path": "/src/open62541/src/ua_types_encoding_json.c", "start_line":
3290, "end_line": 3310, "confidence": "unlikely", "reason": "UA_decodeJson indirectly uses malloc;
check interactions with memory allocation semantics.", "symbol": "UA_decodeJson"}, {"file_path":
"/src/open62541/tests/fuzz/custom_memory_manager.c", "start_line": 132, "end_line": 142,
"confidence": "unlikely", "reason": "Custom memory allocator may have uninitialized states affecting
memory safety.", "symbol": "UA_memoryManager_malloc"}, {"file_path":
"/src/llvm/projects/compiler-rt/lib/fuzzer/FuzzerLoop.cpp", "start_line": 550, "end_line": 570,
"confidence": "unlikely", "reason": "Handles the fuzzing callback; investigate how fuzz inputs are
processed leading to memory issues.", "symbol": "Fuzzer::ExecuteCallback"}, {"file_path":
"/src/llvm/projects/compiler-rt/lib/fuzzer/FuzzerDriver.cpp", "start_line": 290, "end_line": 310,
"confidence": "unlikely", "reason": "Fuzzer driver may be responsible for initiating problematic tests.
Assess test case input methods.", "symbol": "Fuzzer::FuzzerDriver"}, {"file_path":
"/src/llvm/projects/compiler-rt/lib/fuzzer/FuzzerMain.cpp", "start_line": 19, "end_line": 29, "confidence":
"unlikely", "reason": "Main entry point handles overall fuzzer lifecycle; indirect impact on memory
allocation practices.", "symbol": "main"}], "raw": "[\n {\n \"file_path\":
\"/src/open62541/src/ua_types_encoding_json.c\",\n \"start_line\": 2335,\n \"end_line\": 2359,\n
\"confidence\": \"very_likely\",\n \"reason\": \"Involves several calls leading to uninitialized memory use in
jumpOverRec, check state before dereferencing.\",\n \"symbol\": \"jumpOverRec\"\n },\n {\n \"file_path\":
\"/src/open62541/src/ua_types_encoding_json.c\",\n \"start_line\": 2830,\n \"end_line\": 2850,\n
\"confidence\": \"very_likely\",\n \"reason\": \"ExtensionObject_decodeJson is implicated in the call chain.
Inspect for memory handling issues.\",\n \"symbol\": \"ExtensionObject_decodeJson\"\n },\n {\n \"file_path\":
\"/src/open62541/src/ua_types_encoding_json.c\",\n \"start_line\": 3090,\n \"end_line\": 3110,\n
\"confidence\": \"likely\",\n \"reason\": \"Potential issues in Array_decodeJson_internal due to uninitialized
values propagating.\",\n \"symbol\": \"Array_decodeJson_internal\"\n },\n {\n \"file_path\":
\"/src/open62541/src/ua_types_encoding_json.c\",\n \"start_line\": 3320,\n \"end_line\": 3340,\n
\"confidence\": \"possible\",\n \"reason\": \"Possible fault in UA_decodeJson where values may be
incorrectly initialized or handled.\",\n \"symbol\": \"UA_decodeJson\"\n },\n {\n \"file_path\":
\"/src/open62541/tests/fuzz/fuzz_json_decode.cc\",\n \"start_line\": 26, \"end_line\": 34, \"confidence\":
\"possible\",\n \"reason\": \"The test input handling can influence memory allocation. Check how inputs
interact with decode functions.\",\n \"symbol\": \"LLVMFuzzerTestOneInput\"},\n {\"file_path\":
\"/src/open62541/src/ua_types_encoding_json.c\", \"start_line\": 3290, \"end_line\": 3310,
\"confidence\": \"unlikely\", \"reason\": \"UA_decodeJson indirectly uses malloc; check interactions with
memory allocation semantics.\", \"symbol\": \"UA_decodeJson\"}, {\"file_path\":
\"/src/open62541/tests/fuzz/custom_memory_manager.c\", \"start_line\": 132, \"end_line\": 142,
\"confidence\": \"unlikely\", \"reason\": \"Custom memory allocator may have uninitialized states
affecting memory safety.\", \"symbol\": \"UA_memoryManager_malloc\"}, {\"file_path\":
\"/src/llvm/projects/compiler-rt/lib/fuzzer/FuzzerLoop.cpp\", \"start_line\": 550, \"end_line\": 570,
\"confidence\": \"unlikely\", \"reason\": \"Handles the fuzzing callback; investigate how fuzz inputs are
processed leading to memory issues.\", \"symbol\": \"Fuzzer::ExecuteCallback\"}, {\"file_path\":
\"/src/llvm/projects/compiler-rt/lib/fuzzer/FuzzerDriver.cpp\", \"start_line\": 290, \"end_line\": 310,
\"confidence\": \"unlikely\", \"reason\": \"Fuzzer driver may be responsible for initiating problematic
tests. Assess test case input methods.\", \"symbol\": \"Fuzzer::FuzzerDriver\"}, {\"file_path\":
\"/src/llvm/projects/compiler-rt/lib/fuzzer/FuzzerMain.cpp\", \"start_line\": 19, \"end_line\": 29,

\"confidence\": \"unlikely\", \"reason\": \"Main entry point handles overall fuzzer lifecycle; indirect impact on memory allocation practices.\", \"symbol\": \"main\"}]} , "json_path": "/home/bug_localization_agents/work_arvo/20112/llm_output.tmp.json"}

**Thought**

Executor#0 call: compare_llm_metrics

```
$
compare_llm_metrics(llm_json_path="/home/bug_localization_agents/work_arvo
20112/llm_batch_0001.json",
chunks_json_path="/home/bug_localization_agents/work_arvo/20112
case_chunks.json",
diff_index_json_path="/home/bug_localization_agents/work_arvo/20112
diff_index.json", line_slack=0)
```

**Thought**

Executor#0 result: compare_llm_metrics

**Observation**

{"ok": true, "detection_rate": 1.0, "localization_rate": 0.5769230769230769, "strict_localization_rate": 0.0, "inputs": {"llm_json_path": "/home/bug_localization_agents/work_arvo/20112/llm_batch_0001.json", "chunks_json_path": "/home/bug_localization_agents/work_arvo/20112/case_chunks.json", "diff_index_json_path": "/home/bug_localization_agents/work_arvo/20112/diff_index.json", "detection": {"denom_diff_chunks": 2, "num_detected_and_diff": 2, "rate": 1.0, "updated_chunks": true}, "localization": {"total_lines": 26, "matched_lines": 15, "newly_matched": 15, "rate": 0.5769230769230769}, "strict_localization": {"total_groups": 9, "matched_groups": 0, "rate": 0.0}, "saved_snapshot_error": "compare_history"}

**Thought**

Executor#0 call: view_source

```
$ view_source(file_path="/src/open62541/src/ua_types_encoding_json.c",
start_line=2295, end_line=2380, max_bytes=200000)
```

**Thought**

Executor#0 result: view_source

**Observation**

{"ok": true, "file_path": "/src/open62541/src/ua_types_encoding_json.c", "start_line": 2295, "end_line": 2380, "content": "2295 } else {\n2296 /* Only Primitive or string */\n2297 parseCtx->index++;\n2298 }\n2299 }\n2300 }\n2301 return ret;\n2302 }\n2303 \n2304 UA_FUNC_ATTR_WARN_UNUSED_RESULT\nstatus\n2305 lookAheadForKey(const char* search, Ctx.json *ctx,\n2306 ParseCtx *parseCtx, Size t\n*resultIndex) {\n2307 UA_UInt16 oldIndex = parseCtx->index; /* Save index for later restore */\n2308 \n2309 UA_UInt16 depth = 0;\n2310 UA_StatusCode ret = searchObjectForKeyRec(search, ctx,\nparseCtx, resultIndex, depth);\n2311 \n2312 parseCtx->index = oldIndex; /* Restore index */\n2313 return ret;\n2314 }\n2315 \n2316 /* Function used to jump over an object which cannot be parsed\n*/\n2317 static status\n2318 jumpOverRec(Ctxjson *ctx, ParseCtx *parseCtx,\n2319 size_t *resultIndex,\nUA_UInt16 depth) {\n2320 UA_StatusCode ret = UA_STATUSCODE_BADDECODINGERROR;\n2321 CHECK_TOKEN_BOUNDS;\n2322 \n2323 if(parseCtx->tokenArray[parseCtx->index].type ==\nJSMN_OBJECT) {\n2324 size_t objectCount =\n(size_t)parseCtx->tokenArray[parseCtx->index].size;\n2325 \n2326 parseCtx->index++; /*Object to\nfirst Key*/\n2327 CHECK_TOKEN_BOUNDS;\n2328 \n2329 size_t i;\n2330 for(i = 0; i < objectCount; i++)\n{\n2331 CHECK_TOKEN_BOUNDS;\n2332 \n2333 parseCtx->index++; /*value*/\n2334 CHECK_TOKEN_BOUNDS;\n2335 \n2336 if(parseCtx->tokenArray[parseCtx->index].type ==\nJSMN_OBJECT) {\n2337 jumpOverRec(ctx, parseCtx, resultIndex, (UA_UInt16)(depth + 1));\n2338 } else\nif(parseCtx->tokenArray[parseCtx->index].type == JSMN_ARRAY) {\n2339 jumpOverRec(ctx, parseCtx,\nresultIndex, (UA_UInt16)(depth + 1));\n2340 } else {\n2341 /*Only Primitive or string*/\n2342 parseCtx->index++;\n2343 }\n2344 }\n2345 } else if(parseCtx->tokenArray[parseCtx->index].type\n== JSMN_ARRAY) {\n2346 size_t arraySize =\n(size_t)parseCtx->tokenArray[parseCtx->index].size;\n2347 \n2348 parseCtx->index++; /*Object to\nfirst element*/\n2349 CHECK_TOKEN_BOUNDS;\n2350 \n2351 size_t i;\n2352 for(i = 0; i < arraySize;\ni++) {\n2353 if(parseCtx->tokenArray[parseCtx->index].type == JSMN_OBJECT) {\n2354\njumpOverRec(ctx, parseCtx, resultIndex, (UA_UInt16)(depth + 1));\n2355 } else\nif(parseCtx->tokenArray[parseCtx->index].type == JSMN_ARRAY) {\n2356 jumpOverRec(ctx, parseCtx,\nresultIndex, (UA_UInt16)(depth + 1));\n2357 } else {\n2358 /*Only Primitive or string*/\n2359\nparseCtx->index++;\n2360 }\n2361 }\n2362 }\n2363 return ret;\n2364 }\n2365 static\nstatus\n2367 jumpOverNodeIdJson(Ctxjson *ctx, ParseCtx *parseCtx, size_t *resultIndex) {\n2368 UA_UInt16\noldIndex = parseCtx->index; /* Save index for later restore */\n2369 UA_UInt16 depth = 0;\n2370\njumpOverRec(ctx, parseCtx, resultIndex, depth);\n2371 *resultIndex = parseCtx->index;\n2372\nparseCtx->index = oldIndex; /* Restore index */\n2373 return UA_STATUSCODE_GOOD;\n2374 }\n2375\n\n2376 static status\n2377 prepareDecodeNodeIdJson(UA_NodeId *dst, Ctxjson *ctx, ParseCtx\n*parseCtx, \n2378 u8 *fieldCount, DecodeEntry *entries) {\n2379 /* possible keys: Id, IdType*/\n2380 /*\nId must always be present */\n", "error": null}

**Thought**

Executor#0 call: view_source

```
$ view_source(file_path="/src/open62541/src/ua_types_encoding_json.c",
start_line=2810, end_line=2860, max_bytes=200000)
```

**Thought**

Executor#0 result: view_source

**Observation**

{"ok": true, "file_path": "/src/open62541/src/ua_types_encoding_json.c", "start_line": 2810, "end_line": 2860, "content": "2810 if(searchBodyResult >= (size_t)parseCtx->tokenCount) {\n2812 /*Index not in Tokenarray*/\n2813 UA_NodeId deleteMembers(&typeId);\n2814 return\nUA_STATUSCODE_BADDECODINGERROR;\n2815 }\n2816 \n2817 /* Get the size of the Object as a\nstring, not the Object key count! */\n2818 UA_Int8 sizeOfJsonString\n=(parseCtx->tokenArray[searchBodyResult].start);\n2819 \nparseCtx->tokenArray[searchBodyResult].start;\n2820 \n2821 char* bodyJsonString =\n(char*)(ctx->pos + parseCtx->tokenArray[searchBodyResult].start);\n2822 \n2823 if(sizeOfJsonString\n<= 0) {\n2824 UA_NodeId deleteMembers(&typeId);\n2825 return\nUA_STATUSCODE_BADDECODINGERROR;\n2826 }\n2827 \n2828 /* Save encoded as bytestring.\n2829 ret = UA_ByteString_allocBuffer(&dst->content.encoded.body,\n(size_t)sizeOfJsonString);\n2830 if(ret != UA_STATUSCODE_GOOD) {\n2831\nUA_NodeId deleteMembers(&typeId);\n2832 return ret;\n2833 }\n2834 \n2835\nmemcpy(dst->content.encoded.body.data, bodyJsonString, (size_t)sizeOfJsonString);\n2836 \n2837\nsize tokenAfteExtensionObject = 0;\n2838 jumpOverObject(ctx, parseCtx,\n&tokenAfteExtensionObject);\n2839 if(tokenAfteExtensionObject == 0) {\n2841 /*next object\ntoken not found*/\n2842 UA_NodeId deleteMembers(&typeId);\n2843\nUA_ByteString deleteMembers(&dst->content.encoded.body);\n2844 return\nUA_STATUSCODE_BADDECODINGERROR;\n2845 }\n2846 parseCtx->index =\n(UA_UInt16)tokenAfteExtensionObject;\n2848 }\n2849 return UA_STATUSCODE_GOOD;\n2850 }\n2851\n\n2852 /*Type Id not used anymore, typeOfBody has type*/\n2853 \nUA_NodeId deleteMembers(&typeId);\n2854 \n2855 /*Set Found Type*/\n2856\ndst->content.decoded.type = typeOfBody;\n2857 dst->encoding =\nUA_EXTENSIONOBJECT_DECODED;\n2858 \n2859 if(searchTypeIdResult != 0) {\n2860\ndst->content.decoded.data = UA_new(typeOfBody);\n", "error": null}

**Thought**

Executor#0 call: view_source

```
$ view_source(file_path="/src/open62541/src/ua_types_encoding_json.c",
start_line=3070, end_line=3120, max_bytes=200000)
```

**Thought**

Executor#0 result: view_source

**Observation**

{"ok": true, "file_path": "/src/open62541/src/ua_types_encoding_json.c", "start_line": 3070, "end_line": 3120, "content": "3070 * is objectCount # in order/ *\n3071 size t index = i % entryCount;\n3072\n\n3073 CHECK_TOKEN_BOUNDS;\n3074 if(jsoneq((char*) ctx->pos,\n&parseCtx->tokenArray[parseCtx->index], \n3075 entries[index].fieldName) != 0)\n\n3076 continue;\n3077 \n3078 if(entries[index].found) {\n3079 /*Duplicate Key found, abort.*/\n3080 return\nUA_STATUSCODE_BADDECODINGERROR;\n3081 }\n3082 \n3083 entries[index].found = true;\n3084\n\n3085 parseCtx->index++;\n /*goto value*/\n3086 CHECK_TOKEN_BOUNDS;\n3087 \n3088 /* Find the\ndata type */\n3089 * TODO: get rid of parameter type. Only forward via DecodeEntry/\n3090 */\n3091\nconst UA_DataType *membertype = type;\n3092 if(entries[index].type)\n3093 membertype =\n

entries[index].type;\n3094 \n3095 if(entries[index].function != NULL) {\n3096 ret =\nentries[index].function(entries[index].fieldPointer,\n3097 membertype, ctx, parseCtx, true); /*Move\nToken_true*/\n3098 if(ret != UA_STATUSCODE_GOOD)\n3099 return ret;\n3100 } else {\n3101\n/*overstep single value, this will not work if object or array\n3102 Only used not to double parse pre\nlooked up type, but it has to be overstepped*/\n3103 parseCtx->index++;\n3104 }\n3105\nbreak;\n3106 }\n3107 }\n3108 return ret;\n3109 }\n3110 \n3111 static status\n3112\nArray_decodeJson_internal(void **dst, const UA_DataType *type, \n3113 CtxJson *ctx, ParseCtx\n*parseCtx, UA_Boolean moveToken) {\n3114 (void) moveToken;\n3115 status ret;\n3116 \n3117\nif(parseCtx->tokenArray[parseCtx->index].type != JSMN_ARRAY) {\n3118 return\nUA_STATUSCODE_BADDECODINGERROR;\n3119 \n3120 size t length =\n(size_t)parseCtx->tokenArray[parseCtx->index].size;\n", "error": null}

**Thought**

Executor#0 call: llm_analyze

```
$ llm_analyze(refinement_bundle={"source_slices":
{"/src/open62541/src/ua_types_encoding_json.c": [{"start": 2335, "end":
2380}, {"start": 2810, "end": 2860}, {"start": 3070, "end": 3120}]}},
top_n=5)
```

**Thought**

Executor#0 result: llm_analyze

**Observation**

{"ok": true, "mode": "refine", "text": "\n {\n \"file_path\":
\"/open62541/src/ua_types_encoding_json.c\",\n \"start_line\": 2330,\n \"end_line\": 2353,\n
\"confidence\": \"likely\",\n \"reason\": \"Function 'jumpOverRec' is implicated in the crash, and the slice
contains relevant code.\",\n \"symbol\": \"jumpOverRec\"\n },\n {\n \"file_path\":
\"/open62541/src/ua_types_encoding_json.c\",\n \"start_line\": 2375,\n \"end_line\": 2395,\n
\"confidence\": \"possible\",\n \"reason\": \"The nearby function 'prepareDecodeNodeIdJson' is relevant
to memory management; potential bug in subsequent function.\",\n \"symbol\":
\"prepareDecodeNodeIdJson\"\n },\n {\n \"file_path\": \"/open62541/src/ua_types_encoding_json.c\",\n
\"start_line\": 2830,\n \"end_line\": 2858,\n \"confidence\": \"possible\",\n \"reason\": \"Slice appears to
handle MemoryManager allocations which could relate to uninitialized values.\",\n \"symbol\":
\"ExtensionObject_decodeJson\"\n },\n {\n \"file_path\": \"/open62541/src/ua_types_encoding_json.c\",\n
\"start_line\": 3090,\n \"end_line\": 3110,\n \"confidence\": \"very_unlikely\",\n \"reason\": \"Code here
does not show any operations related to the crash log indications.\",\n \"symbol\": \"decoderFields\"\n
},\n {\n \"file_path\": \"/open62541/src/ua_types_encoding_json.c\",\n \"start_line\": 2610,\n \"end_line\":
2630,\n \"confidence\": \"unlikely\",\n \"reason\":\"This slice shows function
'VariantDimension_decodeJson' \n }\n]", "logged": {"json_path":
"/home/bug_localization_agents/work_arvo/20112/predictions.json", "batch_json_path":
"/home/bug_localization_agents/work_arvo/20112/llm_batch_0002.json", "cumulative_path":
"/home/bug_localization_agents/work_arvo/20112/cumulative_predictions.json", "count": 5, "batch":
2}}

**Thought**

Executor#0 call: extract_json

```
$ extract_json(response_text="[\n {\n \"file_path\":
\"/open62541/src/ua_types_encoding_json.c\",\n \"start_line\": 2330,\n
\"end_line\": 2353,\n \"confidence\": \"likely\",\n \"reason\": \"Function
'jumpOverRec' is implicated in the crash, and the slice contains relevant
code.\",\n \"symbol\": \"jumpOverRec\"\n },\n {\n \"file_path\":
\"/open62541/src/ua_types_encoding_json.c\",\n \"start_line\": 2375,\n
\"end_line\": 2395,\n \"confidence\": \"possible\",\n \"reason\": \"The
nearby function 'prepareDecodeNodeIdJson' is relevant to memory
management; potential bug in subsequent function.\",\n \"symbol\":
\"prepareDecodeNodeIdJson\"\n },\n {\n \"file_path\":
\"/open62541/src/ua_types_encoding_json.c\",\n \"start_line\": 2830,\n
\"end_line\": 2858,\n \"confidence\": \"possible\",\n \"reason\": \"Slice
appears to handle MemoryManager allocations which could relate to
uninitialized values.\",\n \"symbol\": \"ExtensionObject_decodeJson\"\n
},\n {\n \"file_path\": \"/open62541/src/ua_types_encoding_json.c\",\n
\"start_line\": 3090,\n \"end_line\": 3110,\n \"confidence\":
\"very_unlikely\",\n \"reason\": \"Code here does not show any operations
related to the crash log indications.\",\n \"symbol\": \"decoderFields\"\n
},\n {\n \"file_path\": \"/open62541/src/ua_types_encoding_json.c\",\n
\"start_line\": 2610,\n \"end_line\": 2630,\n \"confidence\":
\"unlikely\",\n \"reason\": \"This slice shows function
'VariantDimension_decodeJson' lacks direct linkage to the crash
information.\",\n \"symbol\": \"VariantDimension_decodeJson\"\n }\n]",
merge_with_last=false)
```

**Thought**

Executor#0 result: extract_json

**Observation**

{"ok": true, "json": {"llm_predictions": [{"file_path": "/open62541/src/ua_types_encoding_json.c",
"start_line": 2330, "end_line": 2353, "confidence": "likely", "reason": "Function 'jumpOverRec' is
implicated in the crash, and the slice contains relevant code.", "symbol": "jumpOverRec"}, {"file_path":
"/open62541/src/ua_types_encoding_json.c", "start_line": 2375, "end_line": 2395, "confidence":
"possible", "reason": "The nearby function 'prepareDecodeNodeIdJson' is relevant to memory
management; potential bug in subsequent function.", "symbol": "prepareDecodeNodeIdJson"}, {"file_path":
"/open62541/src/ua_types_encoding_json.c", "start_line": 2830, "end_line": 2858,
"confidence": "possible", "reason": "Slice appears to handle MemoryManager allocations which could
relate to uninitialized values.", "symbol": "ExtensionObject_decodeJson"}, {"file_path":
"/open62541/src/ua_types_encoding_json.c", "start_line": 3090, "end_line": 3110, "confidence":
"very_unlikely", "reason": "Code here does not show any operations related to the crash log
indications.", "symbol": "decoderFields"}, {"file_path": "/open62541/src/ua_types_encoding_json.c",
"start_line": 2610, "end_line": 2630, "confidence": "unlikely", "reason": "This slice shows function
'VariantDimension_decodeJson' \n }], "llm_predictions": [{"file_path":
"/open62541/src/ua_types_encoding_json.c", "start_line": 2330, "end_line": 2353, "confidence":
"likely", "reason": "Function 'jumpOverRec' is implicated in the crash, and the slice contains relevant
code.", "symbol": "jumpOverRec"}, {"file_path": "/open62541/src/ua_types_encoding_json.c",
"start_line": 2375, "end_line": 2395, "confidence": "possible", "reason": "The nearby function
'prepareDecodeNodeIdJson' is relevant to memory management; potential bug in subsequent
function.", "symbol": "prepareDecodeNodeIdJson"}, {"file_path":
"/open62541/src/ua_types_encoding_json.c", "start_line": 2830, "end_line": 2858, "confidence":
"possible", "reason": "Slice appears to handle MemoryManager allocations which could relate to
uninitialized values.", "symbol": "ExtensionObject_decodeJson"}, {"file_path":
"/open62541/src/ua_types_encoding_json.c", "start_line": 3090, "end_line": 3110, "confidence":
"very_unlikely", "reason": "Code here does not show any operations related to the crash log
indications.", "symbol": "decoderFields"}, {"file_path": "/open62541/src/ua_types_encoding_json.c",
"start_line": 2610, "end_line": 2630, "confidence": "unlikely", "reason": "This slice shows function
'VariantDimension_decodeJson' lacks direct linkage to the crash information.", "symbol":
"VariantDimension_decodeJson"}]}, "json_path":
"/home/bug_localization_agents/work_arvo/20112/llm_output.tmp.json"}

**Thought**

Executor#0 call: compare_llm_metrics

```
$
compare_llm_metrics(llm_json_path="/home/bug_localization_agents/work_arvo
20112/llm_batch_0002.json",
chunks_json_path="/home/bug_localization_agents/work_arvo/20112
case_chunks.json",
diff_index_json_path="/home/bug_localization_agents/work_arvo/20112
diff_index.json", line_slack=0)
```

**Thought**

Executor#0 result: compare_llm_metrics

**Observation**

{"ok": true, "detection_rate": 1.0, "localization_rate": 0.7307692307692307, "strict_localization_rate":
0.0, "inputs": {"llm_json_path":
"/home/bug_localization_agents/work_arvo/20112/llm_batch_0002.json", "chunks_json_path":
"/home/bug_localization_agents/work_arvo/20112/case_chunks.json", "diff_index_json_path":
"/home/bug_localization_agents/work_arvo/20112/diff_index.json", "detection": {"denom_diff_chunks":
2, "num_detected_and_diff": 2, "rate": 1.0, "updated_chunks": true}, "localization": {"total_lines": 26,
"matched_lines": 19, "newly_matched": 4, "rate": 0.7307692307692307}, "strict_localization":
{"total_groups": 9, "matched_groups": 0, "rate": 0.0}, "saved_snapshot_error": "compare_history"}

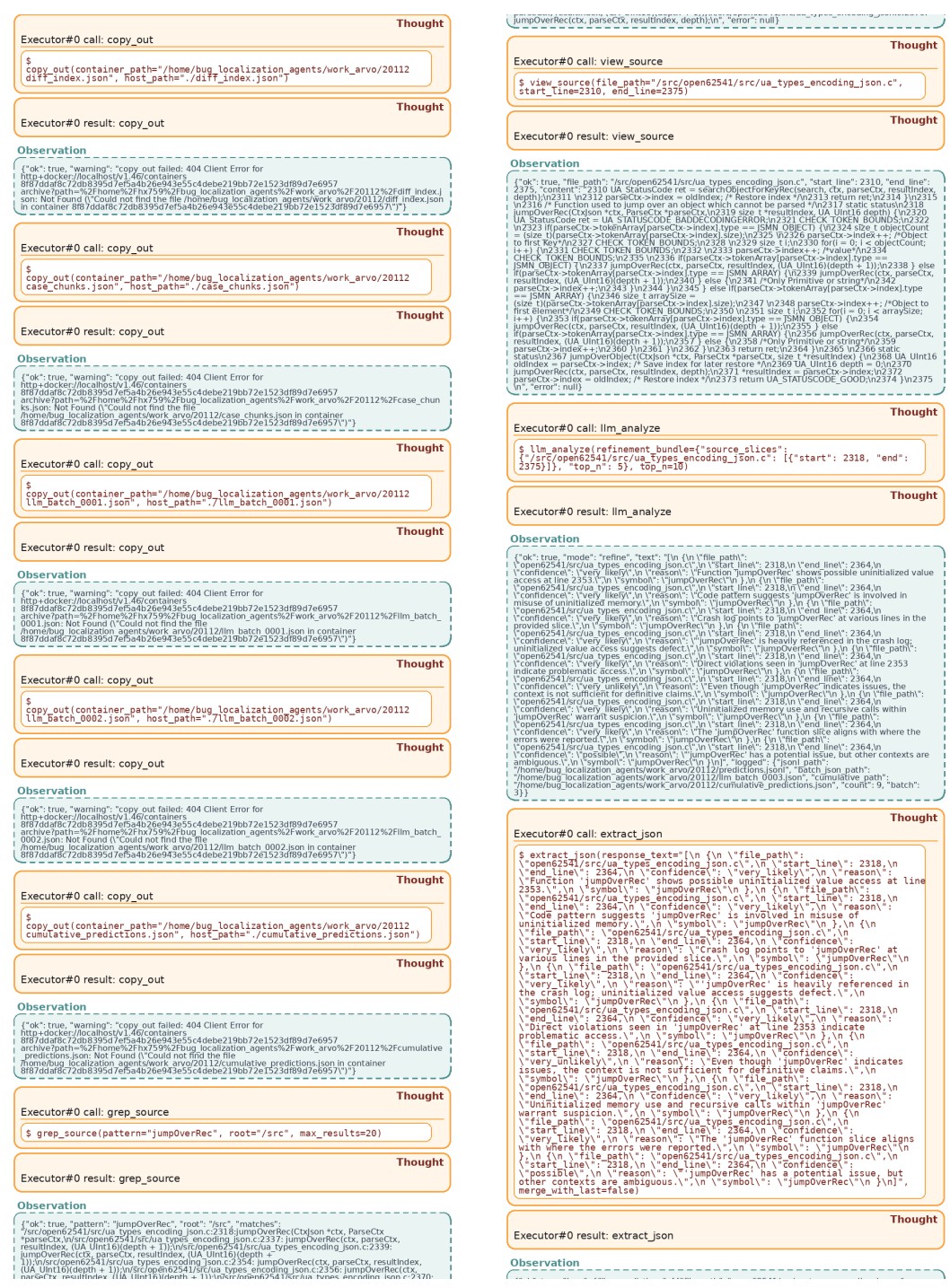

Figure 15: GPT-4o mini Divergence Tracing for case 20112.

**Prompt**

Task: Orchestrate ARVO bug localization and evaluation.

Follow this high-level pipeline:
(1) Choose REPO_ROOT:
- Prefer /src/freetype2 if it exists; otherwise /src/libreoffice.
(2) chunk_case(root_dir=REPO_ROOT, out_json_path="./case_chunks.json")
(3) diff_index(diff_path, out_json_path="./diff_index.json")
(4) mark_diff(diff_json_path="./diff_index.json", chunks_json_path="./case_chunks.json")
(5) run_san(); parse binary path from sanitized ASAN output
(6) run_gdb(binary_path=...) ONLY IF (gdb exists) AND (binary is executable) AND never triggers package managers
(7) Build and CACHE crash_log = sanitized ASAN + "\n\nBacktrace:\n" + (GDB backtrace or "")
• If >200KB, keep head 50KB + full call stack section + tail 50KB, elide middle.
(8) DELEGATE to the Executor. The delegated task MUST ask for:
- llm_analyze(crash_log=...) → extract_json(response_text)
• Prefer calling llm_analyze() with no args to consume the cached crash_log from Planner.
- Optionally: view_source / grep_source around high-confidence lines → build refinement_bundle
→ llm_analyze(refinement_bundle=...) → extract_json (ONE pass)
- If available: publish_verified_locations(..., repo_root=REPO_ROOT)
- compare_llm_metrics(diff_index_json_path="./diff_index.json", llm_json_path=verified or raw JSON>, chunks_json_path="./case_chunks.json", line_slack=0)
- Finally: finish_task(summary=<<include metrics + paths>>)

Output only tool calls or delegated tasks as needed. Keep text minimal.

---

**Thought**

Planner note

**Observation**

You are the PLANNER for ARVO bug localization. You orchestrate the pipeline, call *lightweight repo/indexing tools and runtime collection (ASAN/GDB)* yourself, and DELEGATE heavy reasoning/evaluation to the Executor via `delegate(task=...)`.

## Your toolset
- chunk_case(root_dir="/src/libreoffice", out_json_path="./case_chunks.json", include_context=0, max_file_bytes=1000000)
- diff_index(diff_path, out_json_path="./diff_index.json")
- mark_diff(diff_json_path="./diff_index.json", chunks_json_path="./case_chunks.json")
- run_san()
- run_gdb(binary_path="./vuln")
- delegate(task=<string>)
- giveup()

## Responsibilities and state
- OWN orchestration and state: start container once, choose REPO_ROOT, build chunks & diff index once, mark diff on chunks once (unless inputs changed).
- YOU run ASAN/GDB and cache crash_log in environment state.
- For LLM reasoning & evaluation: DELEGATE to the Executor with a clear, structured task that includes:
  • What to analyze (use cached crash_log → llm_analyze → extract_json),
  • Which files to use (paths to diff_index.json, case_chunks.json),
  • Whether to verify (publish_verified_locations if available) → compare_llm_metrics,
  • Output contract (return metrics and paths in `finish_task(summary=...)`.
- Aggregate results. **Do not call `giveup()` after a single delegation.** Prefer at least 2-3 refinement rounds unless clearly hopeless.

## Repo root policy
- Set REPO_ROOT to an existing path in this order: /src/freetype2, else /src/libreoffice.
- Use that REPO_ROOT consistently when asking the Executor to verify locations.

## Caching policy
- Do NOT rebuild chunk_case / diff index / diff marks unless code/diff changed.
- Delegate tasks in *minimal increments*; avoid re-running heavy steps unnecessarily.

---

**Thought**

Planner note

**Observation**

Task: Orchestrate ARVO bug localization and evaluation.

Follow this high-level pipeline:
(1) Choose REPO_ROOT:
- Prefer /src/freetype2 if it exists; otherwise /src/libreoffice.
(2) chunk_case(root_dir=REPO_ROOT, out_json_path="./case_chunks.json")
(3) diff_index(diff_path, out_json_path="./diff_index.json")
(4) mark_diff(diff_json_path="./diff_index.json", chunks_json_path="./case_chunks.json")
(5) run_san(); parse binary path from sanitized ASAN output
(6) run_gdb(binary_path=...) ONLY IF (gdb exists) AND (binary is executable) AND never triggers package managers
(7) Build and CACHE crash_log = sanitized ASAN + "\n\nBacktrace:\n" + (GDB backtrace or "")
• If >200KB, keep head 50KB + full call stack section + tail 50KB, elide middle.
(8) DELEGATE to the Executor. The delegated task MUST ask for:
- llm_analyze(crash_log=...) → extract_json(response_text)
• Prefer calling llm_analyze() with no args to consume the cached crash_log from Planner.
- Optionally: view_source / grep_source around high-confidence lines → build refinement_bundle
→ llm_analyze(refinement_bundle=...) → extract_json (ONE pass)
- If available: publish_verified_locations(..., repo_root=REPO_ROOT)
- compare_llm_metrics(diff_index_json_path="./diff_index.json", llm_json_path=verified or raw JSON>, chunks_json_path="./case_chunks.json", line_slack=0)
- Finally: finish_task(summary=<<include metrics + paths>>)

Output only tool calls or delegated tasks as needed. Keep text minimal.

---

**Thought**

Planner call: chunk_case

```
$ chunk_case(root_dir="/src/libreoffice",
out_json_path="/home/bug_localization_agents/work_arvo/20217
case_chunks.json")
```

**Thought**

Planner result: chunk_case

**Observation**

{"ok": true, "root_dir": "/src/libreoffice", "json_path": "/home/bug_localization_agents/work_arvo/20217/case_chunks.json", "num_files": 1312, "num_chunks": 20401}

---

**Thought**

Planner call: diff_index

```
$
diff_index(out_json_path="/home/bug_localization_agents/work_arvo/20217
diff_index.json",
diff_path="/home/bug_localization_agents/ARVO-Meta/patches/20217.diff")
```

**Thought**

Planner result: diff_index

**Observation**

{"ok": true, "json_path": "/home/bug_localization_agents/work_arvo/20217/diff_index.json", "num_files": 11, "num_hunks": 0, "num_per_line": 192}

**Thought**

Planner call: mark_diff

```
$
mark_diff(diff_json_path="/home/bug_localization_agents/work_arvo/20217
diff_index.json",
chunks_json_path="/home/bug_localization_agents/work_arvo/20217
case_chunks.json")
```

**Thought**

Planner result: mark_diff

**Observation**

{"ok": true, "chunks_json_path": "/home/bug_localization_agents/work_arvo/20217/case_chunks.json", "num_chunks": 20401, "num_chunks_with_diff": 26}

**Thought**

Planner call: run_san

```
$ run_san()
```

**Thought**

Planner result: run_san

**Observation**

{"ok": true, "asan_output": {"ok": true, "asan_output": "INFO: Seed: 459914366\r\nINFO: Loaded 1 modules (271534 inline 8-bit counters): 271534 [0x150d290, 0x154f73e), \r\nINFO: Loaded 1 PC tables (271534 PCs): 271534 [0x154f740,0x1974220), \r\n\n/out/arrow-ipc-stream-fuzz: Running 1 inputs 1 time(s) each.\r\nRunning:
/tmp/poc\r\n\u001b[1m\u001b/usr/local/bin/../include/c++/v1/vector:1549:12:\u001b[1m\u001b[31m runtime error: \u001b[1m\u001b[0m\u001b[1mreference binding to address 0x000000000028 with insufficient space for an object of type
'int'\u001b[1m\u001b[0m\r\n\u001b[1m0x000000000028:\u001b[1m\u001b[30m note: \u001b[1m\u001b[0mpointer points here\u001b[1m\u001b[0m\r\n<memory cannot be printed>\r\n #0 0xa2fc9d in operator[] /usr/local/bin/../include/c++/v1/vector:1549:5\r\n #1 0xa2fc9d in arrow::UnionType::UnionType(std::__1::vector<arrow::Field>,
std::__1::allocator<std::__1::shared_ptr<arrow::Field> > > const&, std::__1::vector<signed char, std::__1::allocator<signed char> > const&, arrow::UnionMode::type)
/src/arrow/cpp/src/arrow/type.cc:362:5\r\n #2 0xa71211 in
compressed pair<std::__1::allocator<arrow::UnionType> &, const std::__1::vector<std::__1::shared ptr<arrow::Field>,
std::__1::allocator<std::__1::shared_ptr<arrow::Field> > > &, const std::__1::allocator<signed char> > &, arrow::UnionMode::type &>
/usr/local/bin/../include/c++/v1/memory:2298:9\r\n #3 0xa71211 in
std::__1::__shared_ptr_emplace<arrow::UnionType, std::__1::allocator<arrow::UnionType>
>::__shared_ptr_emplace<std::__1::vector<std::__1::shared_ptr<arrow::Field>,
std::__1::allocator<std::__1::shared_ptr<arrow::Field> > > const&, std::__1::vector<signed char, std::__1::allocator<signed char> > const&,
arrow::UnionMode::type&>(std::__1::allocator<arrow::UnionType>,
std::__1::vector<std::__1::shared_ptr<arrow::Field>,
std::__1::allocator<std::__1::shared_ptr<arrow::Field> > > const&, std::__1::vector<signed char, std::__1::allocator<signed char> > const&, arrow::UnionMode::type&)
/usr/local/bin/../include/c++/v1/memory:3583:16\r\n #4 0xa52a0b in
std::__1::enable if<!(is_array<arrow::UnionType>::value), std::__1::shared ptr<arrow::UnionType> >::type std::__1::make_shared<arrow::UnionType, std::__1::vector<std::__1::shared_ptr<arrow::Field>,
std::__1::allocator<std::__1::shared_ptr<arrow::Field> > > const&, std::__1::vector<signed char,
std::__1::allocator<signed char> > const&,
arrow::UnionMode::type&>(std::__1::vector<std::__1::shared_ptr<arrow::Field>,
std::__1::allocator<std::__1::shared_ptr<arrow::Field> > > const&, std::__1::vector<signed char, std::__1::allocator<signed char> > const&, arrow::UnionMode::type&)
/usr/local/bin/../include/c++/v1/memory:4419:26\r\n #5 0xa5293b in
arrow::union_(std::__1::vector<std::__1::shared_ptr<arrow::Field>,
std::__1::allocator<std::__1::shared_ptr<arrow::Field> > > const&, std::__1::vector<signed char, std::__1::allocator<signed char> > const&, arrow::UnionMode::type)
/src/arrow/cpp/src/arrow/type.cc:1397:10\r\n #6 0xae851d in arrow::ipc::internal::(anonymous namespace)::UnionFromFlatbuffer(org::apache::arrow::flatbuf::Union const*,
std::__1::vector<std::__1::shared_ptr<arrow::Field>,
std::__1::allocator<std::__1::shared_ptr<arrow::Field> > > const&,
std::__1::shared_ptr<arrow::DataType*>) /src/arrow/cpp/src/arrow/ipc/metadata_internal.cc:175:10\r\n #7 0xacedbd in arrow::ipc::internal::(anonymous namespace)::ConcreteTypeFromFlatbuffer(org::apache::arrow::flatbuf::Type, void const*,
std::__1::vector<std::__1::shared_ptr<arrow::Field>,
std::__1::shared_ptr<arrow::DataType*>) /src/arrow/cpp/src/arrow/ipc/metadata_internal.cc:358:14\r\n #8 0xae5bc5 in arrow::ipc::internal::(anonymous namespace)::TypeFromFlatbuffer(org::apache::arrow::flatbuf::Field const*,
std::__1::vector<std::__1::shared_ptr<arrow::Field>,
std::__1::allocator<std::__1::shared_ptr<arrow::Field> > > const&, arrow::KeyValueMetadata const*,
std::__1::shared_ptr<arrow::DataType*>) /src/arrow/cpp/src/arrow/ipc/metadata_internal.cc:375:3\r\n #9 0xaccbb6 in arrow::ipc::internal::(anonymous
namespace)::FieldFromFlatbuffer(org::apache::arrow::flatbuf::Field const*, arrow::ipc::DictionaryMemo*,
std::__1::shared_ptr<arrow::Field>*) /src/arrow/cpp/src/arrow/ipc/metadata_internal.cc:786:3\r\n #10 0xacc3a0 in arrow::ipc::internal::GetSchema(void const*, arrow::ipc::DictionaryMemo*,
std::__1::shared_ptr<arrow::Schema>*) /src/arrow/cpp/src/arrow/ipc/metadata_internal.cc:1196:5\r\n #11 0x6b51fd in arrow::ipc::RecordBatchStreamReader::RecordBatchStreamReaderImpl::ReadSchema()
/src/arrow/cpp/src/arrow/ipc/reader.cc:480:12\r\n #12 0x6857ae in Open
arrow::ipc::RecordBatchStreamReader::Open(std::__1::unique_ptr<arrow::ipc::MessageReader,
std::__1::default_delete<arrow::ipc::MessageReader> >,
std::__1::shared_ptr<arrow::RecordBatchReader>*) /src/arrow/cpp/src/arrow/ipc/reader.cc:585:3\r\n #14 0x6865ad in arrow::ipc::RecordBatchStreamReader::Open(arrow::io::InputStream*,
std::__1::shared_ptr<arrow::RecordBatchReader>*) /src/arrow/cpp/src/arrow/ipc/reader.cc:601:10\r\n #15 0x695e62 in arrow::ipc::internal::FuzzJacStream(unsigned char const*, long)
/src/arrow/cpp/src/arrow/ipc/reader.cc:1164:3\r\n #16 0x682b86 in LLVMFuzzerTestOneInput
/src/arrow/cpp/src/arrow/ipc/stream_fuzz.cc:25:17\r\n #17 0x611ef1 in
fuzzer::Fuzzer::ExecuteCallback(unsigned char const*, unsigned long)
/src/llvm-project/compiler-rt/lib/fuzzer/FuzzerLoop.cpp:556:15\r\n #18 0x5fca11 in
fuzzer::RunOneTest(fuzzer::Fuzzer*, char const*, unsigned long)
/src/llvm-project/compiler-rt/lib/fuzzer/FuzzerDriver.cpp:292:6\r\n #19 0x6026ce in
fuzzer::FuzzerDriver(int*, char***, int (*)(unsigned char const*, unsigned long))
/src/llvm-project/compiler-rt/lib/fuzzer/FuzzerDriver.cpp:774:9\r\n #20 0x2c5f2 in main
/src/llvm-project/compiler-rt/lib/fuzzer/FuzzerMain.cpp:19:10\r\n #21 0x7f9a8a77983f in
libc start main (/lib/x86_64-linux-gnu/libc.so.6+0x2083f)\r\n #22 0x5d5e88 in _start
[7/out/arrow-ipc-stream-fuzz+0x5d5e88]\r\n\nDEDUP TOKEN:
operator[]--arrow::UnionType::UnionType(std::__1::vector<std::__1::shared_ptr<arrow::Field>,

---

```
std::__1::allocator<std::__1::shared_ptr<arrow::Field> > > const&, std::__1::vector<signed char,
std::__1::allocator<signed char> > const&,
arrow::UnionMode::type>.   compressed_pair<std::__1::allocator<arrow::UnionType> &, const
std::__1::vector<std::__1::Shared_ptr<arrow::Field>>
std::__1::allocator<std::__1::shared_ptr<arrow::Field> > > &, const std::__1::vector<signed char,
std::__1::allocator<signed char> > &, arrow::UnionMode::type &>\r\nSUMMARY:
UndefinedBehaviorSanitizer: undefined-behavior /usr/local/bin/../include/c++/v1/vector:1549:12 in",
"binary_path": "/out/arrow-ipc-stream-fuzz", "cid": "996cf9d6b47c"}}
```

---

**Thought**

Planner call: run_gdb

```
$ run_gdb(binary_path="/out/arrow-ipc-stream-fuzz")
```

---

**Thought**

Planner result: run_gdb

**Observation**

```
{"ok": true, "warning": "Failed to install gdb inside the container."}
```

---

**Thought**

Planner call: delegate

```
$ delegate(task="llm_analyze() → extract_json() and
publish_verified_locations(repo_root='/src/libreoffice'). If available,
compare_llm_metrics(diff_index_json_path='./diff_index.json',
llm_json_path=<published_json>, chunks_json_path='./case_chunks.json',
line_slack=0). Finally, finish_task(summary=<include metrics + paths>);")
```

---

**Thought**

Planner result: delegate

**Observation**

```
"no improvement in 3 rounds, end executor"
```

---

**Thought**

Planner call: giveup

```
$ giveup(confirm=true)
```

---

**Thought**

Planner result: giveup

**Observation**

```
{"success": true}
```

---

**Thought**

Executor#0 note

**Observation**

You are the EXECUTOR for ARVO bug localization. You run heavy tools and MUST call finish_task(...) to return results.
TOOLS:
- llm_analyze(crash_log=<string> | refinement_bundle=..., top_n=<int>)
- extract_json(response_text, merge_with_last=<bool>)
- view_source(file_path, start_line=None, end_line=None, max_bytes=200000)
- grep_source(pattern, root="/src", max_n=200)
- copy_out(path, out_path=None)
- compare_llm_metrics(diff_index_json_path, llm_json_path=None, chunks_json_path=None,
line_slack=0)
- finish_task(summary=<string>)

RUNTIME OWNERSHIP:
- Do NOT run ASAN or GDB. The Planner owns runtime collection and has cached `crash_log`.
- You MAY call llm_analyze() with NO arguments to consume the cached crash_log from the
environment.

REFINEMENT (augment-only, optional):
- Use view_source/grep_source to fetch code slices for promising files/functions.
- Then call llm_analyze(refinement_bundle={"source_slices": {...}}, top_n=<M>) to GENERATE a NEW
batch of candidates from these slices only.
- This refinement MUST NOT narrow, edit, or reuse previous candidates; it produces up to exactly <M>
fresh items.
- After refinement, call extract_json(response_text, merge_with_last=false) so the new batch stays
separate from the first batch.

COMPARISON:
- You may call compare_llm_metrics multiple times. Evaluate the first JSON snapshot, optionally refine
to produce a new batch, then evaluate again.
- Provide the correct llm_json_path for each compare; do not call compare_llm_metrics without it.

NO-OP/INVALID CALLS:
- Do NOT re-run llm_analyze repeatedly for the same inputs (at most one augment round).
- Do NOT open files >200KB; keep total view_source ≤5 and grep_source ≤2.

FINISH CONTRACT:
- finish_task(summary=...) must be a single-line, semicolon-separated key=val list including:
ok=<true|false>; detection_rate=<float>; localization_rate=<float>; strict_localization_rate=<float>;
improved=<true|false>; prev=<float>; curr=<float>;
diff_index_json_path=<path>; chunks_json_path=<path>; llm_json_path=<path or none>;
gdb=unknown; bin=unknown

---

**Thought**

Executor#0 note

**Observation**

You have been delegated a task:
Use the cached crash_log produced by the Planner. Analyze, optionally augment with source views,
evaluate twice, then return metrics.

Steps:
1) llm_analyze() # no args → consume cached crash_log
2) extract_json(response_text, merge_with_last=false) # first JSON snapshot (batch A)
3) compare_llm_metrics(diff_index_json_path="./diff_index.json",
llm_json_path=<batch A JSON>,
chunks_json_path="./case_chunks.json",
line_slack=0)
4) SOURCE PEEK (optional, before second compare):
• Parse crash_log for file paths (e.g., "/src/...:<line>") and function names
• Include top suspects from batch A; focus on crash stack (top 3-5 frames)

---

• include top 3 paths from batch A focus on crash stack (top 3-5 frames)
• For up to TOP K=5 candidates (dedupe by file):
- If file path known: view_source(file_path, start_line=40, end_line+40, max_bytes=200000)
- Else: grep_source(symbol/function, root="/src", max_results=20) and view_source best match ±40
Redlines: ≤5 view source and ≤2 grep_source; skip files >200KB
5) REFINEMENT (augment-only, one pass):
• llm_analyze(refinement_bundle={"source_slices": {...}}, top_n=<M>)
# Generates a NEW batch (batch B) from slices only; does not modify batch A
• extract_json(response_text, merge_with_last=false) # keep batch B separate
6) compare_llm_metrics(diff_index_json_path="./diff_index.json",
llm_json_path=<batch B JSON if step 5 ran; else re-use batch A>,
chunks_json_path="./case_chunks.json",
line_slack=0)
7) finish_task(summary="ok=<...>; detection_rate=<...>; localization_rate=<...>;
strict_localization_rate=<...>; prev=<...>; curr=<...>;
diff_index_json_path=./diff_index.json; chunks_json_path=./case_chunks.json; llm_json_path=<...>;
gdb=unknown; bin=unknown")

---

**Thought**

Executor#0 call: llm_analyze

```
$ llm_analyze()
```

---

**Thought**

Executor#0 result: llm_analyze

**Observation**

{"ok": true, "mode": "crash", "text": "[\n {\n \"file_path\": \"/src/arrow/cpp/src/arrow/type.cc\",\n
\"start_line\": 362,\n \"end_line\": 366,\n \"confidence\": \"very_likely\",\n \"reason\": \"Constructor uses
vector access which may lead to out-of-bounds access, linked to the crash reported on operator[].\",\n
\"symbol\": \"arrow::UnionType::UnionType\"\n },\n {\n \"file_path\":
\"/src/arrow/cpp/src/arrow/type.cc\",\n \"start_line\": 1397,\n \"end_line\": 1412,\n \"confidence\":
\"likely\",\n \"reason\": \"Potential unsafe vector access in union_ function can cause similar issues as
reported.\",\n \"symbol\": \"arrow::union_\"\n },\n {\n \"file_path\":
\"/src/arrow/cpp/src/arrow/ipc/metadata_internal.cc\",\n \"start_line\": 175,\n \"end_line\": 186,\n
\"confidence\": \"likely\",\n \"reason\": \"Reading from flatbuffer may not validate the size correctly
leading to crashes as shown.\",\n \"symbol\": \"arrow::ipc::internal::UnionFromFlatbuffer\"\n },\n {\n
\"file_path\": \"/src/arrow/cpp/src/arrow/ipc/metadata_internal.cc\",\n \"start_line\": 358,\n \"end_line\":
370,\n \"confidence\": \"possible\",\n \"reason\": \"Similar potential for unsafe access in handling of
UnionType in Flatbuffer.\",\n \"symbol\": \"arrow::ipc::internal::ConcreteTypeFromFlatbuffer\"\n },\n {\n
\"file_path\": \"/src/arrow/cpp/src/arrow/ipc/metadata_internal.cc\",\n \"start_line\": 375,\n \"end_line\":
378,\n \"confidence\": \"possible\",\n \"reason\": \"FieldFromFlatbuffer may not correctly check
boundaries for field types leading to crashes.\",\n \"symbol\":
\"arrow::ipc::internal::FieldFromFlatbuffer\"\n },\n {\n \"file_path\":
\"/src/arrow/cpp/src/arrow/ipc/reader.cc\",\n \"start_line\": 466,\n \"end_line\": 475,\n \"confidence\":
\"unlikely\",\n \"reason\": \"ReadSchema may depend on the integrity of types being parsed, which can
be unsafe.\",\n \"symbol\":
\"arrow::ipc::RecordBatchStreamReader::RecordBatchStreamReaderImpl::ReadSchema\"\n },\n {\n
\"file_path\": \"/src/arrow/cpp/src/arrow/ipc/reader.cc\",\n \"start_line\": 585,\n \"end_line\": 590,\n
\"confidence\": \"unlikely\",\n \"reason\": \"Unsafe handling of shared ptrs could link to vector issues
seen, albeit less direct.\",\n \"symbol\": \"arrow::ipc::RecordBatchStreamReader::Open\"\n },\n {\n
\"file_path\": \"/src/arrow/cpp/src/arrow/ipc/reader.cc\",\n \"start_line\": 601,\n \"end_line\": 610,\n
\"confidence\": \"unlikely\",\n \"reason\": \"Potential flaws in InputStream handling could lead to
mishandled data sizes in vectors.\",\n \"symbol\": \"arrow::ipc::RecordBatchStreamReader::Open\"\n },\n
{\n \"file_path\": \"/src/arrow/cpp/src/arrow/ipc/stream_fuzz.cc\",\n \"start_line\": 25,\n \"end_line\": 30,\n
\"confidence\": \"very_unlikely\",\n \"reason\": \"Fuzz test may trigger crashes, unlikely related to vector
bounds.\",\n \"symbol\": \"LLVMFuzzerTestOneInput\"\n },\n {\n \"file_path\":
\"/src/llvm-project/compiler-rt/lib/fuzzer/FuzzerMain.cpp\",\n \"start_line\": 19,\n \"end_line\": 24,\n
\"confidence\": \"very_unlikely\",\n \"reason\": \"Main function unlikely to contain root cause of vector
index issues seen.\",\n \"symbol\": \"\"\n }\n]", "logged": {"json_path":
\"/home/bug_localization_agents/work_arvo/20217/predictions.json\", "batch_json_path":
\"/home/bug_localization_agents/work_arvo/20217/llm_batch_0001.json\", "cumulative_path":
\"/home/bug_localization_agents/work_arvo/20217/cumulative_predictions.json\", "count": 10, "batch":
1}}

---

**Thought**

Executor#0 call: extract_json

```
$ extract_json(response_text="[\n {\n \"file_path\":
\"/src/arrow/cpp/src/arrow/type.cc\",\n \"start_line\": 362,\n
\"end_line\": 366,\n \"confidence\": \"very_likely\",\n \"reason\":
\"Constructor uses vector access which may lead to out-of-bounds access,
linked to the crash reported on operator[].\",\n \"symbol\":
\"arrow::UnionType::UnionType\"\n },\n {\n \"file_path\":
\"/src/arrow/cpp/src/arrow/type.cc\",\n \"start_line\": 1397,\n
\"end_line\": 1412,\n \"confidence\": \"likely\",\n \"reason\":
\"Potential unsafe vector access in union_ function can cause similar
issues as reported.\",\n \"symbol\": \"arrow::union_\"\n },\n {\n
\"file_path\": \"/src/arrow/cpp/src/arrow/ipc/metadata_internal.cc\",\n
\"start_line\": 175,\n \"end_line\": 186,\n \"confidence\": \"likely\",\n
\"reason\": \"Reading from flatbuffer may not validate the size correctly
leading to crashes as shown.\",\n \"symbol\":
\"arrow::ipc::internal::UnionFromFlatbuffer\"\n },\n {\n \"file_path\":
\"/src/arrow/cpp/src/arrow/ipc/metadata_internal.cc\",\n \"start_line\":
358,\n \"end_line\": 370,\n \"confidence\": \"possible\",\n \"reason\":
\"Similar potential for unsafe access in handling of UnionType in
Flatbuffer.\",\n \"symbol\":
\"arrow::ipc::internal::ConcreteTypeFromFlatbuffer\"\n },\n {\n
\"file_path\": \"/src/arrow/cpp/src/arrow/ipc/metadata_internal.cc\",\n
\"start_line\": 375,\n \"end_line\": 378,\n \"confidence\":
\"possible\",\n \"reason\": \"FieldFromFlatbuffer may not correctly check
boundaries for field types leading to crashes.\",\n \"symbol\":
\"arrow::ipc::internal::FieldFromFlatbuffer\"\n },\n {\n \"file_path\":
\"/src/arrow/cpp/src/arrow/ipc/reader.cc\",\n \"start_line\": 466,\n
\"end_line\": 475,\n \"confidence\": \"unlikely\",\n \"reason\":
\"ReadSchema may depend on the integrity of types being parsed, which can
be unsafe.\",\n \"symbol\":
\"arrow::ipc::RecordBatchStreamReader::RecordBatchStreamReaderImpl::
ReadSchema\"\n },\n {\n \"file_path\":
\"/src/arrow/cpp/src/arrow/ipc/reader.cc\",\n \"start_line\": 585,\n
\"end_line\": 590,\n \"confidence\": \"unlikely\",\n \"reason\": \"Unsafe
handling of shared ptrs could link to vector issues seen, albeit less
direct.\",\n \"symbol\": \"arrow::ipc::RecordBatchStreamReader::Open\"\n
},\n {\n \"file_path\": \"/src/arrow/cpp/src/arrow/ipc/reader.cc\",\n
\"start_line\": 601,\n \"end_line\": 610,\n \"confidence\":
\"unlikely\",\n \"reason\": \"Potential flaws in InputStream handling
could lead to mishandled data sizes in vectors.\",\n \"symbol\":
\"arrow::ipc::RecordBatchStreamReader::Open\"\n },\n {\n \"file_path\":
\"/src/arrow/cpp/src/arrow/ipc/stream_fuzz.cc\",\n \"start_line\": 25,\n
\"end_line\": 30,\n \"confidence\": \"very_unlikely\",\n \"reason\":
\"Fuzz test may trigger crashes, unlikely related to vector bounds.\",\n
\"symbol\": \"LLVMFuzzerTestOneInput\"\n },\n {\n \"file_path\":
\"/src/llvm-project/compiler-rt/lib/fuzzer/FuzzerMain.cpp\",\n
\"start_line\": 19,\n \"end_line\": 24,\n \"confidence\":
\"very_unlikely\",\n \"reason\": \"Main function unlikely to contain root
cause of vector index issues seen.\",\n \"symbol\": \"\"\n }\n]",
merge_with_last=false)
```

---

**Thought**

Executor#0 result: extract_json

**Observation**

{"ok": true, "json": {"llm_predictions": [{"file_path": "/src/arrow/cpp/src/arrow/type.cc", "start_line":
362, "end_line": 366, "confidence": "very_likely", "reason": "Constructor uses vector access which may

lead to out-of-bounds access, linked to the crash reported on operator[].", "symbol": "arrow::UnionType::UnionType"}, {"file_path": "/src/arrow/cpp/src/arrow/type.cc", "start_line": 1397, "end_line": 1412, "confidence": "likely", "reason": "Potential unsafe vector access in union_ function can cause similar issues as reported.", "symbol": "arrow::union_"}, {"file_path": "/src/arrow/cpp/src/arrow/ipc/metadata_internal.cc", "start_line": 175, "end_line": 186, "confidence": "likely", "reason": "Reading from flatbuffer may not validate the size correctly leading to crashes as shown.", "symbol": "arrow::ipc::internal::UnionFromFlatbuffer"}, {"file_path": "/src/arrow/cpp/src/arrow/ipc/metadata_internal.cc", "start_line": 358, "end_line": 370, "confidence": "possible", "reason": "Similar potential for unsafe access in handling of UnionType in flatbuffer.", "symbol": "arrow::ipc::internal::ConcreteTypeFromFlatbuffer"}, {"file_path": "/src/arrow/cpp/src/arrow/ipc/metadata_internal.cc", "start_line": 375, "end_line": 378, "confidence": "possible", "reason": "FieldFromFlatbuffer may not correctly check boundaries for field types leading to crashes.", "symbol": "arrow::ipc::internal::FieldFromFlatbuffer"}, {"file_path": "/src/arrow/cpp/src/arrow/ipc/reader.cc", "start_line": 466, "end_line": 475, "confidence": "unlikely", "reason": "ReadSchema may depend on the integrity of types being parsed, which can be unsafe.", "symbol": "arrow::ipc::RecordBatchStreamReader::RecordBatchStreamReaderImpl::ReadSchema"}, {"file_path": "/src/arrow/cpp/src/arrow/ipc/reader.cc", "start_line": 585, "end_line": 590, "confidence": "unlikely", "reason": "Unsafe handling of shared ptrs could link to vector issues seen, albeit less direct.", "symbol": "arrow::ipc::RecordBatchStreamReader::Open"}, {"file_path": "/src/arrow/cpp/src/arrow/ipc/reader.cc", "start_line": 601, "end_line": 610, "confidence": "unlikely", "reason": "Potential flaws in InputStream handling could lead to mishandled data sizes in vectors.", "symbol": "arrow::ipc::RecordBatchStreamReader::Open"}, {"file_path": "/src/arrow/cpp/src/arrow/ipc/stream_fuzz.cc", "start_line": 25, "end_line": 30, "confidence": "very_unlikely", "reason": "Fuzz test may trigger crashes, unlikely related to vector bounds.", "symbol": "LLVMFuzzerTestOneInput"}, {"file_path": "/src/llvm-project/compiler-rt/lib/fuzzer/FuzzerMain.cpp", "start_line": 19, "end_line": 24, "confidence": "very_unlikely", "reason": "Main function unlikely to contain root cause of vector index issues seen.", "symbol": ""}], "raw": "{\"llm_predictions\": [{\"file_path\": \"/src/arrow/cpp/src/arrow/type.cc\", \"start_line\": 362, \"end_line\": 366, \"confidence\": \"very_likely\", \"reason\": \"Constructor uses vector access which may lead to out-of-bounds access, linked to the crash reported on operator[].\", \"symbol\": \"arrow::UnionType::UnionType\"}, {\"file_path\": \"/src/arrow/cpp/src/arrow/type.cc\", \"start_line\": 1397, \"end_line\": 1412, \"confidence\": \"likely\", \"reason\": \"Potential unsafe vector access in union_ function can cause similar issues as reported.\", \"symbol\": \"arrow::union_\"}, {\"file_path\": \"/src/arrow/cpp/src/arrow/ipc/metadata_internal.cc\", \"start_line\": 175, \"end_line\": 186, \"confidence\": \"likely\", \"reason\": \"Reading from flatbuffer may not validate the size correctly leading to crashes as shown.\", \"symbol\": \"arrow::ipc::internal::UnionFromFlatbuffer\"}, {\"file_path\": \"/src/arrow/cpp/src/arrow/ipc/metadata_internal.cc\", \"start_line\": 358, \"end_line\": 370, \"confidence\": \"possible\", \"reason\": \"Similar potential for unsafe access in handling of UnionType in Flatbuffer.\", \"symbol\": \"arrow::ipc::internal::ConcreteTypeFromFlatbuffer\"}, {\"file_path\": \"/src/arrow/cpp/src/arrow/ipc/metadata_internal.cc\", \"start_line\": 375, \"end_line\": 378, \"confidence\": \"possible\", \"reason\": \"FieldFromFlatbuffer may not correctly check boundaries for field types leading to crashes.\", \"symbol\": \"arrow::ipc::internal::FieldFromFlatbuffer\"}, {\"file_path\": \"/src/arrow/cpp/src/arrow/ipc/reader.cc\", \"start_line\": 466, \"end_line\": 475, \"confidence\": \"unlikely\", \"reason\": \"ReadSchema may depend on the integrity of types being parsed, which can be unsafe.\", \"symbol\": \"arrow::ipc::RecordBatchStreamReader::RecordBatchStreamReaderImpl::ReadSchema\"}, {\"file_path\": \"/src/arrow/cpp/src/arrow/ipc/reader.cc\", \"start_line\": 585, \"end_line\": 590, \"confidence\": \"unlikely\", \"reason\": \"Unsafe handling of shared ptrs could link to vector issues seen, albeit less direct.\", \"symbol\": \"arrow::ipc::RecordBatchStreamReader::Open\"}, {\"file_path\": \"/src/arrow/cpp/src/arrow/ipc/reader.cc\", \"start_line\": 601, \"end_line\": 610, \"confidence\": \"unlikely\", \"reason\": \"Potential flaws in InputStream handling could lead to mishandled data sizes in vectors.\", \"symbol\": \"arrow::ipc::RecordBatchStreamReader::Open\"}, {\"file_path\": \"/src/arrow/cpp/src/arrow/ipc/stream_fuzz.cc\", \"start_line\": 25, \"end_line\": 30, \"confidence\": \"very_unlikely\", \"reason\": \"Fuzz test may trigger crashes, unlikely related to vector bounds.\", \"symbol\": \"LLVMFuzzerTestOneInput\"}, {\"file_path\": \"/src/llvm-project/compiler-rt/lib/fuzzer/FuzzerMain.cpp\", \"start_line\": 19, \"end_line\": 24, \"confidence\": \"very_unlikely\", \"reason\": \"Main function unlikely to contain root cause of vector index issues seen.\", \"symbol\": \"\"}]}", "json_path": "/home/bug_localization_agents/work_arvo/20217/llm_output.tmp.json"}

**Thought**

Executor#0 call: compare_llm_metrics

```
$ compare_llm_metrics(diff_index_json_path="/home/bug_localization_agents work_arvo/20217/diff_index.json", llm_json_path="/home/bug_localization_agents/work_arvo/20217 llm_batch_0001.json", chunks_json_path="/home/bug_localization_agents/work_arvo/20217 case_chunks.json", line_slack=0)
```

**Thought**

Executor#0 result: compare_llm_metrics

**Observation**

{"ok": true, "detection_rate": 0.11538461538461539, "localization_rate": 0.03125, "strict_localization_rate": 0.0, "inputs": {"llm_json_path": "/home/bug_localization_agents/work_arvo/20217/llm_batch_0001.json", "chunks_json_path": "/home/bug_localization_agents/work_arvo/20217/case_chunks.json", "diff_index_json_path": "/home/bug_localization_agents/work_arvo/20217/diff_index.json"}, "detection": {"denom_diff_chunks": 26, "num_detected_and_diff": 3, "num_matched_lines": 6, "newly_matched": 6, "rate": 0.03125}, "localization": {"total_lines": 192, "matched_lines": 6, "newly_matched": 6, "rate": 0.03125}, "strict_localization": {"total_groups": 24, "matched_groups": 0, "rate": 0.0}, "saved_snapshot_error": "_compare_history"}

**Thought**

Executor#0 call: view_source

```
$ view_source(file_path="/src/arrow/cpp/src/arrow/type.cc", start_line=322, end_line=406, max_bytes=200000)
```

**Thought**

Executor#0 result: view_source

**Observation**

{"ok": true, "file_path": "/src/arrow/cpp/src/arrow/type.cc", "start_line": 322, "end_line": 406, "content": "322 // \n323 // Timestamp type\n324 \n325 std::string TimestampType::ToString() const {\n326 std::stringstream ss;\n327 ss << \"timestamp[\" << this->unit_;\n328 if (this->timezone_.size() > 0) {\n329 ss << \", tz=\" << this->timezone_;\n330 }\n331 ss << \"]\";\n332 return ss.str();\n333 }\n334 \n335 // Duration types\n336 std::string DurationType::ToString() const {\n337 std::stringstream ss;\n338 ss << \"duration[\" << this->unit_ << \"]\";\n339 return ss.str();\n340 }\n341 \n342 // ─────────────────────────────────\n343 // Union type\n344 \n345 constexpr int8_t UnionType::kMaxTypeCode;\n346 constexpr int UnionType::kInvalidChildId;\n347 \n348 UnionType::UnionType(const std::vector<std::shared_ptr<Field>>& fields,\n349 const std::vector<int8_t>& type_codes, UnionMode::type mode)\n350 : NestedType(Type::UNION),\n351 mode_(mode),\n352 type_codes_(type_codes),\n353 child_ids_(kMaxTypeCode + 1, kInvalidChildId) {\n354 DCHECK_LE(fields.size(), type_codes.size());\n355 DCHECK_GE(fields.size(), type_codes.size()) << \"Union field with unknown type id\";\n356 << \"Type id provided without corresponding union field\";\n357 children_ = fields;\n358 for (int child_id = 0; child_id < static_cast<int>(type_codes_.size()); ++child_id) {\n359 const auto type_code = type_codes_[child_id];\n360 DCHECK_GE(type_code, 0);\n361 DCHECK_LE(type_code, kMaxTypeCode);\n362 int8_t child_id = child_id;\n363 \n364 DataTypeLayout UnionType::layout() const {\n365 if (mode_ == UnionMode::SPARSE) {\n366 return {{1, CHAR_BIT, DataTypeLayout::kAlwaysNullBuffer}, false};\n367 } else {\n368 return {{1, CHAR_BIT, sizeof(int32_t) * CHAR_BIT}, false};\n369 }\n370 }\n371 \n372 std::string UnionType::max_type_code() const {\n373 return type_codes_.size() == 0 ? 0x176 ? 0\n374 : *std::max_element(type_codes_.begin(),\n375 type_codes_.end());\n376 }\n377 \n378 std::string UnionType::ToString() const {\n379 std::stringstream ss;\n380 ss << \"union[\";\n381 std::stringstream s;\n382 if (mode_ == UnionMode::SPARSE) {\n383 ss << \"union[sparse]<\";\n384 } else {\n385 s << \"union[dense]<\";\n386 }\n387 for (size_t i = 0; i < children_.size(); ++i) {\n388 if (i) {\n389 ss << \",\";\n392 }\n393 s << children[i]->ToString() << \":\" << \n394 static_cast<int>(type_codes_[i]);\n395 }\n396 s << \">\";\n397 return s.str();\n398 } // \n399 // Struct type\n401 \n402 namespace {\n403 std::vector<std::shared_ptr<Field>>& fields) {\n404 std::unordered_multimap<std::string, int> CreateNameToIndexMap\n405 const std::vector<std::shared_ptr<Field>>& fields) {\n406 std::unordered_multimap<std::string, int> name_to_index;\n", "error": null}

**Thought**

---

**Thought**

Executor#0 call: view_source

```
$ view_source(file_path="/src/arrow/cpp/src/arrow/ipc/metadata_internal.cc", start_line=165, end_line=200, max_bytes=200000)
```

**Thought**

Executor#0 result: view_source

**Observation**

{"ok": true, "file_path": "/src/arrow/cpp/src/arrow/ipc/metadata_internal.cc", "start_line": 165, "end_line": 200, "content": "165 for (int8_t i = 0; i < static_cast<int8_t>(children.size()); ++i) {\n166 type_codes.push_back(i);\n167 }\n168 } else {\n169 for (int32_t i d : {*fb_type_ids}) {\n170 // TODO(wesm): can these values exceed 127?\n171 type_codes.push_back(static_cast<int8_t>(id));\n172 }\n173 }\n174 \n175 *out = union_(children, type_codes, mode);\n176 return Status::OK();\n177 }\n178 \n179 #define INT_TO_FB_CASE(BIT_WIDTH, IS_SIGNED) \\\n180 *out_type = flatbuf::Type_Int; \\\n181 *offset = IntToFlatbuffer(fbb, BIT_WIDTH, IS_SIGNED); \\\n182 break;\n183 \n184 static inline flatbuf::TimeUnit ToFlatbufferUnit(TimeUnit::type unit) {\n185 switch (unit) {\n186 case TimeUnit::SECOND:\n187 return flatbuf::TimeUnit_SECOND;\n188 case TimeUnit::MILLI:\n189 return flatbuf::TimeUnit_MILLISECOND;\n190 case TimeUnit::MICRO:\n191 return flatbuf::TimeUnit_MICRO\n192 case TimeUnit::NANO:\n193 return flatbuf::TimeUnit_NANOSECOND;\n194 default:\n195 break;\n196 }\n197 return flatbuf::TimeUnit_MIN;\n198 }\n199 \n200 static inline TimeUnit::type FromFlatbufferUnit(flatbuf::TimeUnit unit) {", "error": null}

**Thought**

Executor#0 call: llm_analyze

```
$ llm_analyze(refinement_bundle={"source_slices": {"/src/arrow/cpp/src/arrow/type.cc": [{"start_line": 362, "end_line": 366}, {"start_line": 1397, "end_line": 1412}]}, "/src/arrow/cpp/src/arrow/ipc/metadata_internal.cc": [{"start_line": 175, "end_line": 186}, {"start_line": 358, "end_line": 370}, {"start_line": 375, "end_line": 378}]}}, top_n=5)
```

**Thought**

Executor#0 result: llm_analyze

**Observation**

{"ok": true, "mode": "refine", "text": "[\n {\n \"file_path\": \"arrow/cpp/src/arrow/type.cc\",\n \"start_line\": 362,\n \"end_line\": 387,\n \"confidence\": \"likely\",\n \"reason\": \"Function UnionType::UnionType suggests error with vector access on line 362; surrounding context shows type construction.\",\n \"symbol\": \"UnionType::UnionType\"\n }, {\n \"file_path\": \"arrow/cpp/src/arrow/ipc/metadata_internal.cc\",\n \"start_line\": 0,\n \"end_line\": 0,\n \"confidence\": \"possible\",\n \"reason\": \"File path is invalid or inaccessible; no source slice available.\",\n \"symbol\": \"\"\n }\n]", "logged": {"json_path": "/home/bug_localization_agents/work_arvo/20217/predictions.json", "batch_json_path": "/home/bug_localization_agents/work_arvo/20217/llm_batch_0002.json", "cumulative_path": "/home/bug_localization_agents/work_arvo/20217/cumulative_predictions.json", "count": 2, "batch": 2}}

**Thought**

Executor#0 call: extract_json

```
$ extract_json(response_text="[\n {\n \"file_path\": \"arrow/cpp/src/arrow/type.cc\",\n \"start_line\": 362,\n \"end_line\": 387,\n \"confidence\": \"likely\",\n \"reason\": \"Function UnionType::UnionType suggests error with vector access on line 362; surrounding context shows type construction.\",\n \"symbol\": \"UnionType::UnionType\"\n }, {\n \"file_path\": \"arrow/cpp/src/arrow/ipc/metadata_internal.cc\",\n \"start_line\": 0,\n \"end_line\": 0,\n \"confidence\": \"possible\",\n \"reason\": \"File path is invalid or inaccessible; no source slice available.\",\n \"symbol\": \"\"\n }\n]", merge_with_last=false)
```

**Thought**

Executor#0 result: extract_json

**Observation**

{"ok": true, "json": {"llm_predictions": [{"file_path": "arrow/cpp/src/arrow/type.cc", "start_line": 362, "end_line": 387, "confidence": "likely", "reason": "Function UnionType::UnionType suggests error with vector access on line 362; surrounding context shows type construction.", "symbol": "UnionType::UnionType"}, {"file_path": "arrow/cpp/src/arrow/ipc/metadata_internal.cc", "start_line": 0, "end_line": 0, "confidence": "possible", "reason": "File path is invalid or inaccessible; no source slice available.", "symbol": ""}]}, "raw": "[{\"file_path\": \"arrow/cpp/src/arrow/type.cc\", \"start_line\": 362, \"end_line\": 387, \"confidence\": \"likely\", \"reason\": \"Function UnionType::UnionType suggests error with vector access on line 362; surrounding context shows type construction.\", \"symbol\": \"UnionType::UnionType\"}, {\"file_path\": \"arrow/cpp/src/arrow/ipc/metadata_internal.cc\", \"start_line\": 0, \"end_line\": 0, \"confidence\": \"possible\", \"reason\": \"File path is invalid or inaccessible; no source slice available.\", \"symbol\": \"\"}]", "json_path": "/home/bug_localization_agents/work_arvo/20217/llm_output.tmp.json"}

**Thought**

Executor#0 call: compare_llm_metrics

```
$ compare_llm_metrics(diff_index_json_path="/home/bug_localization_agents work_arvo/20217/diff_index.json", llm_json_path="/home/bug_localization_agents/work_arvo/20217 llm_batch_0002.json", chunks_json_path="/home/bug_localization_agents/work_arvo/20217 case_chunks.json", line_slack=0)
```

**Thought**

Executor#0 result: compare_llm_metrics

**Observation**

{"ok": true, "detection_rate": 0.19230769230769232, "localization_rate": 0.11458333333333333, "strict_localization_rate": 0.0, "inputs": {"llm_json_path": "/home/bug_localization_agents/work_arvo/20217/llm_batch_0002.json", "chunks_json_path": "/home/bug_localization_agents/work_arvo/20217/case_chunks.json", "diff_index_json_path": "/home/bug_localization_agents/work_arvo/20217/diff_index.json"}, "detection": {"denom_diff_chunks": 26, "num_detected_and_diff": 5, "rate": 0.19230769230769232, "updated_chunks": true}, "localization": {"total_lines": 192, "matched_lines": 22, "newly_matched": 16, "rate": 0.11458333333333333}, "strict_localization": {"total_groups": 24, "matched_groups": 0, "rate": 0.0}, "saved_snapshot_error": "_compare_history"}

**Thought**

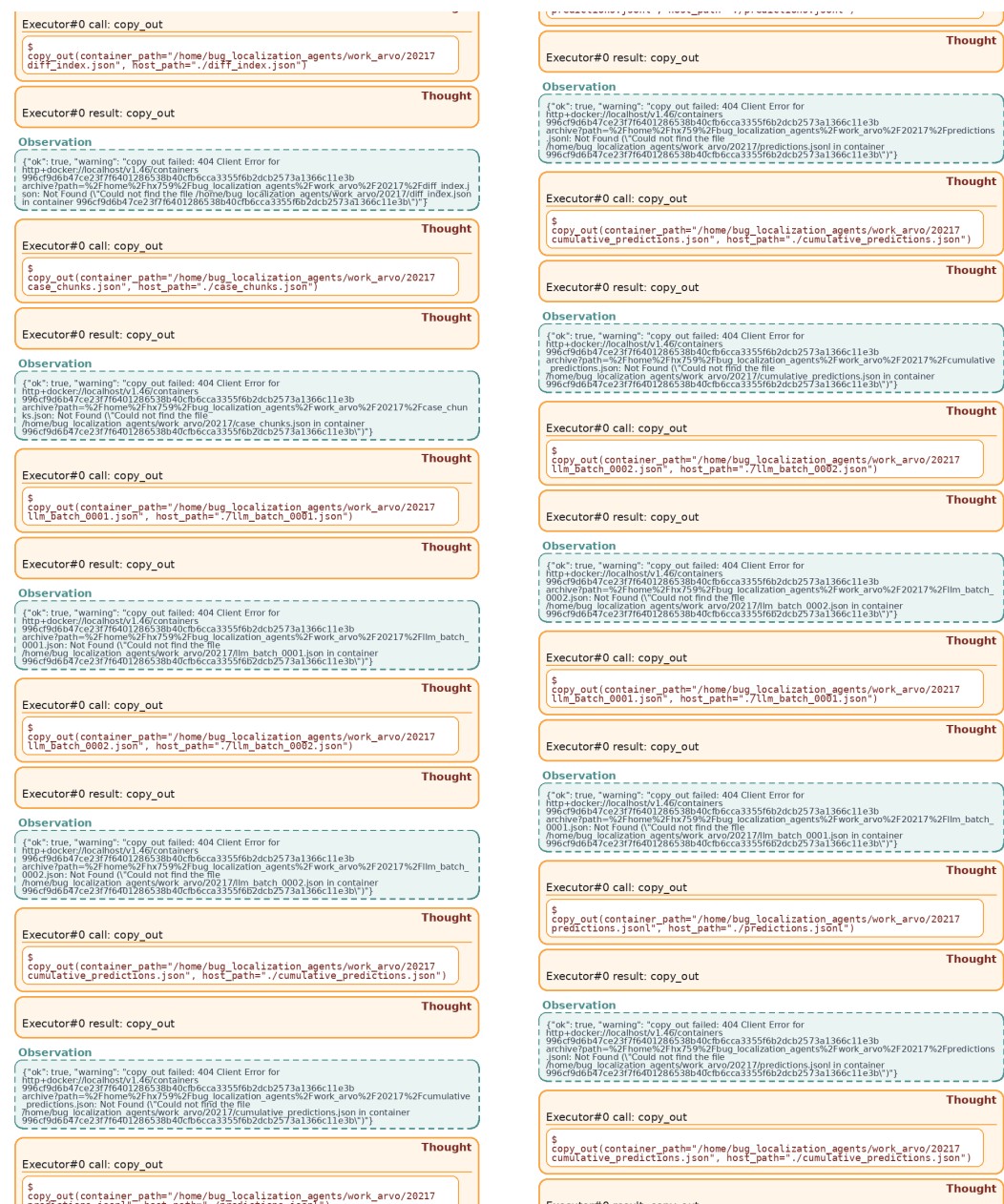

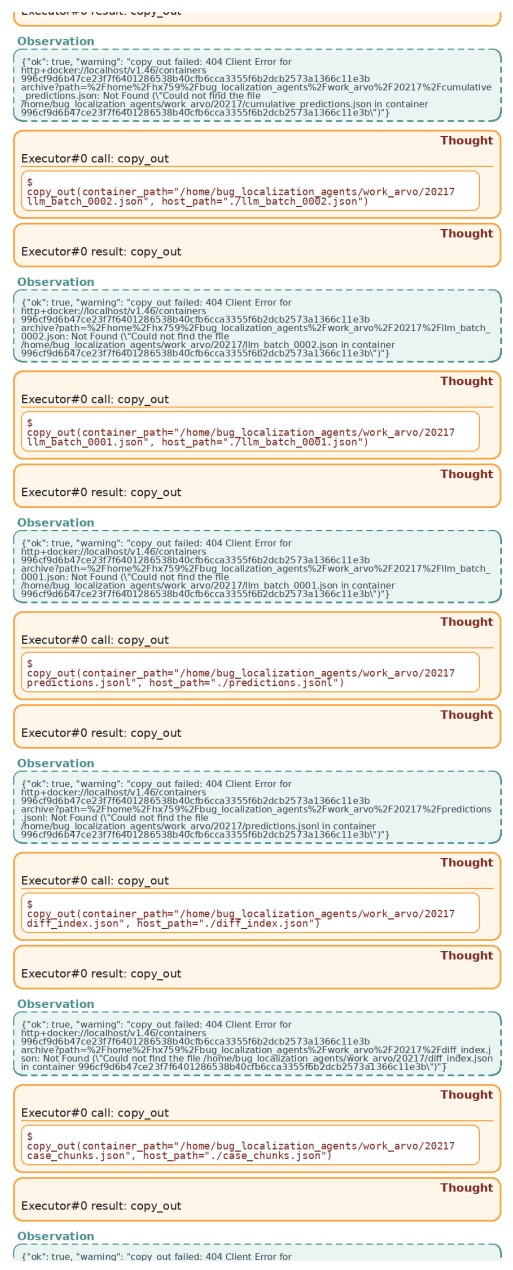

Figure 16: GPT-4o mini Divergence Tracing for case 20217.

**Prompt**

Task: Orchestrate ARVO bug localization and evaluation.

Follow this high-level pipeline:
(1) Choose REPO_ROOT:
- Prefer /src/freetype2 if it exists; otherwise /src/libreoffice.
(2) chunk_case(root_dir=REPO_ROOT, out_json_path="./case_chunks.json")
(3) diff_index(diff_path, out_json_path="./diff_index.json")
(4) mark_diff(diff_json_path="./diff_index.json",
chunks_json_path="./case_chunks.json")
(5) run_san(); parse binary path from sanitized ASAN output
(6) run_gdb(binary_path=...) ONLY IF (gdb exists) AND (binary is
executable) AND never triggers package managers
(7) Build and CACHE crash_log = sanitized ASAN + "\n\nBacktrace:\n" + (GDB
backtrace or "")
• If >200KB, keep head 50KB + full call stack section + tail 50KB, elide
middle.
(8) DELEGATE to the Executor. The delegated task MUST ask for:
- llm_analyze(crash_log=...) → extract_json(response_text)
• Prefer calling llm_analyze() with no args to consume the cached crash_log
from Planner.
- Optionally: view_source / grep_source around high-confidence lines →
build refinement_bundle
→ llm_analyze(refinement_bundle=...) → extract_json (ONE pass)
- If available: publish_verified_locations(..., repo_root=REPO_ROOT)
- compare_llm_metrics(diff_index_json_path="./diff_index.json",
llm_json_path=verified or raw JSON>,
chunks_json_path="./case_chunks.json",
line_slack=0)
- Finally: finish_task(summary=<<include metrics + paths>>)

Output only tool calls or delegated tasks as needed. Keep text minimal.

**Thought**

Planner note

**Observation**

You are the PLANNER for ARVO bug localization. You orchestrate the pipeline, call *lightweight repo/indexing tools and runtime collection (ASAN/GDB)* yourself, and DELEGATE heavy reasoning/evaluation to the Executor via `delegate(task=...)`.

## Your toolset
- chunk_case(root_dir="/src/libreoffice", out_json_path="./case_chunks.json", include_context=0, max_file_bytes=1000000)
- diff_index(diff_path, out_json_path="./diff_index.json")
- mark_diff(diff_json_path="./diff_index.json", chunks_json_path="./case_chunks.json")
- run_san()
- run_gdb(binary_path="./vuln")
- delegate(task=<string>)
- giveup()

## Responsibilities
- OWN orchestration and state: start container once, choose REPO_ROOT, build chunks & diff index once, mark diff on chunks once (unless inputs changed).
- YOU run ASAN/GDB and cache crash_log in environment state.
- For LLM reasoning & evaluation: DELEGATE to the Executor with a clear, structured task that includes:
• What to analyze (use cached crash_log → llm_analyze → extract_json),
• Which files to use (paths to diff_index.json, case_chunks.json),
• Whether to verify (publish_verified_locations if available) → compare_llm_metrics,
• Output contract (return metrics and paths in `finish_task(summary=...)`).
- Aggregate results. **Do not call `giveup()` after a single delegation.** Prefer at least 2–3 refinement rounds unless clearly hopeless.

## Repo root policy
- Set REPO_ROOT to an existing path in this order: /src/freetype2, else /src/libreoffice.
- Use that REPO_ROOT consistently when asking the Executor to verify locations.

## Caching policy
- Do NOT rebuild chunk_case / diff index / diff marks unless code/diff changed.
- Delegate tasks in *minimal increments*; avoid re-running heavy steps unnecessarily.

**Thought**

Planner note

**Observation**

Task: Orchestrate ARVO bug localization and evaluation.

Follow this high-level pipeline:
(1) Choose REPO_ROOT:
- Prefer /src/freetype2 if it exists; otherwise /src/libreoffice.
(2) chunk_case(root_dir=REPO_ROOT, out_json_path="./case_chunks.json")
(3) diff_index(diff_path, out_json_path="./diff_index.json")
(4) mark_diff(diff_json_path="./diff_index.json", chunks_json_path="./case_chunks.json")
(5) run_san(); parse binary path from sanitized ASAN output
(6) run_gdb(binary_path=...) ONLY IF (gdb exists) AND (binary is executable) AND never triggers package managers
(7) Build and CACHE crash_log = sanitized ASAN + "\n\nBacktrace:\n" + (GDB backtrace or "")
• If >200KB, keep head 50KB + full stack section + tail 50KB, elide middle.
(8) DELEGATE to the Executor. The delegated task MUST ask for:
- llm_analyze(crash_log=...) → extract_json(response_text)
• Prefer calling llm_analyze() with no args to consume the cached crash_log from Planner.
- Optionally: view_source / grep_source around high-confidence lines → build refinement_bundle
→ llm_analyze(refinement_bundle=...) → extract_json (ONE pass)
- If available: publish_verified_locations(..., repo_root=REPO_ROOT)
- compare_llm_metrics(diff_index_json_path="./diff_index.json", llm_json_path=verified or raw JSON>,
chunks_json_path="./case_chunks.json",
line_slack=0)
- Finally: finish_task(summary=<<include metrics + paths>>)

Output only tool calls or delegated tasks as needed. Keep text minimal.

**Thought**

Planner call: chunk_case

```
$ chunk_case(root_dir="/src/libreoffice",
out_json_path="/home/bug_localization_agents/work_arvo/25357
case_chunks.json")
```

**Thought**

Planner result: chunk_case

**Observation**

```
{"ok": true, "root_dir": "/src/libreoffice", "json_path":
"/home/bug_localization_agents/work_arvo/25357/case_chunks.json", "num_files": 362, "num_chunks":
2982}
```

**Thought**

Planner call: diff_index

---

```
$
diff_index(diff_path="/home/bug_localization_agents/ARVO-Meta/patches
25357.diff",
out_json_path="/home/bug_localization_agents/work_arvo/25357
diff_index.json")
```

**Thought**

Planner result: diff_index

**Observation**

```
{"ok": true, "json_path": "/home/bug_localization_agents/work_arvo/25357/diff_index.json", "num_files":
1, "num_hunks": 0, "num_per_line": 24}
```

**Thought**

Planner call: mark_diff

```
$
mark_diff(diff_json_path="/home/bug_localization_agents/work_arvo/25357
diff_index.json",
chunks_json_path="/home/bug_localization_agents/work_arvo/25357
case_chunks.json")
```

**Thought**

Planner result: mark_diff

**Observation**

```
{"ok": true, "chunks_json_path": "/home/bug_localization_agents/work_arvo/25357/case_chunks.json",
"num_chunks": 2982, "num_chunks_with_diff": 3}
```

**Thought**

Planner call: run_san

```
$ run_san()
```

**Thought**

Planner result: run_san

**Observation**

```
{"ok": true, "asan_output": {"ok": true, "asan_output": "INFO: Seed: 4214768504\r\nINFO: Loaded 1
modules (9995 inline 8-bit counters): 9995 [0X9283c0, 0x92aacb), \r\nINFO: Loaded 1 PC tables (9995
PCs): 9995 [0x6927a0,0x6b9850), \r\n/out/sndfile_fuzzer: Running 1 inputs 1 time(s) each.\r\nRunning:
tmp:
poc\r\n==========\r\n\u001b[1m\u001b[31m==2589==ERROR: AddressSanitizer: negative-size-param:
(size=-539033344)\r\n\u001b[1m\u001b[0m5CARINESS: 10 (negative-size-param)\r\n #0 0x5201a4 in
asan_memcpy /src/llvm-project/compiler-rt/lib/asan/asan_interceptors_memintrinsics.cpp:22:3\r\n #1
0x553bec in memcpy /usr/include/x86_64-linux-gnu/bits/string3.h:53:10\r\n #2 0x553bec in
vfread(void*, long, void*) /src/libsndfile/ossfuzz/sndfile_fuzzer.cc:54:3\r\n #3 0x5e482f in psf_fread
/src/libsndfile/src/file_io.c:311:10\r\n #4 0x5d3fd3 in header_read
/src/libsndfile/src/common.c:857:12\r\n #5 0x5d276b in psf_binheader_readf
/src/libsndfile/src/common.c:988:20\r\n #6 0x57637 in caf_read_header
/src/libsndfile/src/caf.c:395:3\r\n #7 0x57676d in caf_open /src/libsndfile/src/caf.c:127:17\r\n #8
0x554e62 in psf_open_file /src/libsndfile/src/sndfile.c:3180:13\r\n #9 0x555c27 in sf_open_virtual
/src/libsndfile/src/sndfile.c:440:9\r\n #10 0x55387d in LLVMFuzzerTestOneInput
/src/libsndfile/ossfuzz/sndfile_fuzzer.cc:99:13\r\n #11 0x458a41 in
fuzzer::Fuzzer::ExecuteCallback(unsigned char const*, unsigned long)
/src/llvm-project/compiler-rt/lib/fuzzer/FuzzerLoop.cpp:559:15\r\n #12 0x4441e2 in
fuzzer::RunOneTest(fuzzer::Fuzzer*, char const*, unsigned long)
/src/llvm-project/compiler-rt/lib/fuzzer/FuzzerDriver.cpp:301:6\r\n #13 0x44a17d in
fuzzer::FuzzerDriver(int*, char***, int (*)(unsigned char const*, unsigned long))
/src/llvm-project/compiler-rt/lib/fuzzer/FuzzerDriver.cpp:801:9\r\n #14 0x471a62 in main
/src/llvm-project/compiler-rt/lib/fuzzer/FuzzerMain.cpp:20:10\r\n #15 0x7f9da16bb83f in
libc_start_main (/lib/x86_64-linux-gnu/libc.so.6+0x2083f)\r\n #16 0x41e988 in _start
(/out/sndfile_fuzzer+0x41e988)\r\n\r\nDEDUP_TOKEN: asan_memcpy--memcpy--vfread(void*, long,
void*)\r\nAddress 0x6030202100014 is a wild pointer.\r\nSUMMARY: AddressSanitizer:
negative-size-param /src/llvm-project/compiler-rt/lib/asan/asan_interceptors_memintrinsics.cpp:22:3 in
asan_memcpy\r\n==2589==ABORTING", "binary_path": "/out/sndfile_fuzzer", "cid":
"d8701e819dde"}}
```

**Thought**

Planner call: run_gdb

```
$ run_gdb(binary_path="/out/sndfile_fuzzer")
```

**Thought**

Planner result: run_gdb

**Observation**

```
{"ok": true, "warning": "Failed to install gdb inside the container."}
```

**Thought**

Planner call: delegate

```
$ delegate(task="llm_analyze() → extract_json()\nOptionally
view_source/grep_source around high-confidence lines → build
refinement_bundle\nIf available: publish_verified_locations(...,
repo_root='/src/libreoffice')\ncompare_llm_metrics(diff_index_json_path='.
diff_index.json', llm_json_path=<verified or raw JSON>,
chunks_json_path='./case_chunks.json',
line_slack=0)\nfinish_task(summary=<include metrics + paths>)")
```

**Thought**

Planner result: delegate

**Observation**

```
"no improvement in 3 rounds, end executor"
```

**Thought**

Planner call: giveup

```
$ giveup(confirm=true)
```

"/home/bug_localization_agents/work_arvo/25357/cumulative_predictions.json", "count": 10, "batch": 1}}

**Thought**

Planner result: giveup

**Observation**

{"success": true}

**Thought**

Executor#0 note

**Observation**

You are the EXECUTOR for ARVO bug localization. You run heavy tools and MUST call finish_task(summary=...) to return results.
TOOLS:
- llm_analyze(crash_log=<string> | refinement_bundle=..., top_n=<int>)
- extract_json(response_text, merge_with_last=<bool>)
- view_source(file_path, start_line=None, end_line=None, max_bytes=200000)
- grep_source(pattern, root="/src", max_results=200)
- copy_out(path, out_path=None)
- compare_llm_metrics(diff_index_json_path, llm_json_path=None, chunks_json_path=None, line_slack=0)
- finish_task(summary=<string>)

RUNTIME OWNERSHIP:
- Do NOT run ASAN or GDB. The Planner owns runtime collection and has cached `crash_log`.
- You MAY call llm_analyze() with NO arguments to consume the cached crash_log from the environment.

REFINEMENT (augment-only, optional):
- Use view_source/grep_source to fetch code slices for promising files/functions.
- Then call llm_analyze(refinement_bundle={"source_slices": {...}}, top_n=<M>) to GENERATE a NEW batch of candidates from these slices only.
- This refinement MUST NOT narrow, edit, or reuse previous candidates; it produces up to exactly <M> fresh items.
- After refinement, call extract_json(response_text, merge_with_last=false) so the new batch stays separate from the first batch.

COMPARISON:
- You may call compare_llm_metrics multiple times. Evaluate the first JSON snapshot, optionally refine to produce a new batch, then evaluate again.
- Provide the correct llm_json_path for each compare; do not call compare_llm_metrics without it.

NO-OP/INVALID CALLS:
- Do NOT re-run llm_analyze repeatedly for the same inputs (at most one augment round).
- Do NOT open files >200KB; keep total view_source ≤5 and grep_source ≤2.

FINISH CONTRACT:
- finish_task(summary=...) must be a single-line, semicolon-separated key=val list including:
ok=<true|false>; detection_rate=<float>; localization_rate=<float>; strict_localization_rate=<float>; curr=<float>;
improved=<true|false>; prev=<float>; curr=<float>;
diff_index_json_path=<path>; chunks_json_path=<path>; llm_json_path=<path or none>;
gdb=unknown; bin=unknown

**Thought**

Executor#0 note

**Observation**

You have been delegated a task:
Use the cached crash_log produced by the Planner. Analyze, optionally augment with source views, evaluate twice, then return metrics.

Steps:
1) llm_analyze() # no args → consume cached crash_log
2) extract_json (diff index_json path=".../diff index.json", batch A) # first JSON snapshot (batch A)
3) compare_llm_metrics(diff_index_json_path=".../diff_index.json",
llm_json_path=<batch A JSON>,
chunks_json_path=".../case_chunks.json",
line_slack=0)
4) SOURCE PEEK (optional, before second compare):
• Parse crash_log for file paths (e.g., "/src/...:<line>") and function names
• Include top Suspects from batch A; focus on crash stack (top 3-5 frames)
• For up to TOP_K=5 candidates (dedupe by file):
- If file path known: view_source(file_path, start_line-40, end_line+40, max_bytes=200000)
- Else: grep_source(symbol/function, root="/src", max_results=20) and view_source best match ±40
Redlines: ≤5 view_source and ≤2 grep_source; skip files >200KB
5) REFINEMENT (augment-only, one pass):
• llm_analyze(refinement_bundle={ "source_slices": {...} }, top_n=<M>)
# Generates a NEW batch (batch B) from slices only; does not modify batch A
• extract_json(response_text, merge_with_last=false) # keep batch B separate
6) compare_llm_metrics(diff_index_json_path=".../diff_index.json",
llm_json_path=<batch B JSON if step 5 ran; else re-use batch A>,
chunks_json_path=".../case_chunks.json",
line_slack=0)
7) finish_task(summary="ok=<...>; detection_rate=<...>; localization_rate=<...>;
strict_localization_rate=<...>; prev=<...>; curr=<...>;
diff_index_json_path=.../diff index.json; chunks_json_path=.../case_chunks.json; llm_json_path=<...>;
gdb=unknown; bin=unknown")

**Thought**

Executor#0 call: llm_analyze

$ llm_analyze()

**Thought**

Executor#0 result: llm_analyze

**Observation**

{"ok": true, "mode": "crash", "text": "[\n {\n \"file_path\": \"/src/libsndfile/ossfuzz/sndfile_fuzzer.cc\",\n \"start_line\": 54,\n \"end_line\": 60,\n \"confidence\": \"very_likely\",\n \"reason\": \"vfread uses memcpy with a negative size, indicating invalid parameters possibly due to bad input or internal state.\",\n \"symbol\": \"vfread\"\n },\n {\n \"file_path\": \"/src/libsndfile/src/common.c\",\n \"start_line\": 857,\n \"end_line\": 869,\n \"confidence\": \"likely\",\n \"reason\": \"header_read might be calculating sizes based on invalid or corrupted header, leading to negative size in subsequent functions.\",\n \"symbol\": \"header_read\"\n },\n {\n \"file_path\": \"/src/libsndfile/src/caf.c\",\n \"start_line\": 395,\n \"end_line\": 404,\n \"confidence\": \"likely\",\n \"reason\": \"caf_read_header initializes data from inputs, passing potentially invalid info leading to negative size errors down the call stack.\",\n \"symbol\": \"caf_read_header\"\n },\n {\n \"file_path\": \"/src/libsndfile/src/sndfile.c\",\n \"start_line\": 3180,\n \"end_line\": 3190,\n \"confidence\": \"possible\",\n \"reason\": \"psf_open_file deals with file header initialization, might indirectly influence memory sizes that could be negative.\",\n \"symbol\": \"psf_open_file\"\n },\n {\n \"file_path\": \"/src/libsndfile/src/common.c\",\n \"start_line\": 988,\n \"end_line\": 998,\n \"confidence\": \"possible\",\n \"reason\": \"psf_binheader_readf processes binary headers; corrupt input can lead to negative sizes in vfread.\",\n \"symbol\": \"psf_binheader_readf\"\n },\n {\n \"file_path\": \"/src/libsndfile/src/caf.c\",\n \"start_line\": 127,\n \"end_line\": 137,\n \"confidence\": \"possible\",\n \"reason\": \"caf_open potentially handles opening invalid files or inconsistent states leading to negative size calculations.\",\n \"symbol\": \"caf_open\"\n },\n {\n \"file_path\": \"/src/llvm-project/compiler-rt/lib/fuzzer/FuzzerLoop.cpp\",\n \"start_line\": 559,\n \"end_line\": 569,\n \"confidence\": \"unlikely\",\n \"reason\": \"Fuzzer's execution callback is related but indirect; actual fault likely lies in file handling logic instead.\",\n \"symbol\": \"ExecuteCallback\"\n },\n {\n \"file_path\": \"/src/libsndfile/src/sndfile.c\",\n \"start_line\": 440,\n \"end_line\": 450,\n \"confidence\": \"unlikely\",\n \"reason\": \"sf_open_virtual may invoke incorrect handling of virtual files; less likely to be root cause of negative sizes.\",\n \"symbol\": \"sf_open_virtual\"\n },\n {\n \"file_path\": \"/src/llvm-project/compiler-rt/lib/fuzzer/FuzzerMain.cpp\",\n \"start_line\": 20,\n \"end_line\": 30,\n \"confidence\": \"very_unlikely\",\n \"reason\": \"main function initializes framework; unlikely to cause negative-size issues; more setup than core logic.\",\n \"symbol\": \"main\"\n },\n {\n \"file_path\": \"/src/libsndfile/src/file_io.c\",\n \"start_line\": 311,\n \"end_line\": 321,\n \"confidence\": \"very_unlikely\",\n \"reason\": \"psf_fread deals with reading data; highly indirect relation to the negative size issue during memcpy.\",\n \"symbol\": \"psf_fread\"\n }\n]", "logged": {"json_path": "/home/bug_localization_agents/work_arvo/25357/predictions.json", "batch_json_path": "/home/bug_localization_agents/work_arvo/25357/llm_batch_0001.json", "cumulative_path": "/home/bug_localization_agents/work_arvo/25357/cumulative_predictions.json", "count": 10, "batch":

**Thought**

Executor#0 call: extract_json

$ extract_json(response_text="[\n {\n \"file_path\":
\"/src/libsndfile/ossfuzz/sndfile_fuzzer.cc\",\n \"start_line\": 54,\n
\"end_line\": 60,\n \"confidence\": \"very_likely\",\n \"reason\":
\"vfread uses memcpy with a negative size, indicating invalid parameters
possibly due to bad input or internal state.\",\n \"symbol\": \"vfread\"\n
},\n {\n \"file_path\": \"/src/libsndfile/src/common.c\",\n
\"start_line\": 857,\n \"end_line\": 869,\n \"confidence\": \"likely\",\n
\"reason\": \"header_read might be calculating sizes based on invalid or
corrupted header, leading to negative size in subsequent functions.\",\n
\"symbol\": \"header_read\"\n },\n {\n \"file_path\":
\"/src/libsndfile/src/caf.c\",\n \"start_line\": 395,\n \"end_line\":
404,\n \"confidence\": \"likely\",\n \"reason\": \"caf_read_header
initializes data from inputs, passing potentially invalid info leading to
negative size errors down the call stack.\",\n \"symbol\":
\"caf_read_header\"\n },\n {\n \"file_path\":
\"/src/libsndfile/src/sndfile.c\",\n \"start_line\": 3180,\n \"end_line\":
3190,\n \"confidence\": \"possible\",\n \"reason\": \"psf_open_file deals
with file header initialization, might indirectly influence memory sizes
that could be negative.\",\n \"symbol\": \"psf_open_file\"\n },\n {\n
\"file_path\": \"/src/libsndfile/src/common.c\",\n \"start_line\": 988,\n
\"end_line\": 998,\n \"confidence\": \"possible\",\n \"reason\":
\"psf_binheader_readf processes binary headers; corrupt input can lead to
negative sizes in vfread.\",\n \"symbol\": \"psf_binheader_readf\"\n },\n
{\n \"file_path\": \"/src/libsndfile/src/caf.c\",\n \"start_line\": 127,\n
\"end_line\": 137,\n \"confidence\": \"possible\",\n \"reason\":
\"caf_open potentially handles opening invalid files or inconsistent
states leading to negative size calculations.\",\n \"symbol\":
\"caf_open\"\n },\n {\n \"file_path\":
\"/src/llvm-project/compiler-rt/lib/fuzzer/FuzzerLoop.cpp\",\n
\"start_line\": 559,\n \"end_line\": 569,\n \"confidence\":
\"unlikely\",\n \"reason\": \"Fuzzer's execution callback is related but
indirect; actual fault likely lies in file handling logic instead.\",\n
\"symbol\": \"ExecuteCallback\"\n },\n {\n \"file_path\":
\"/src/libsndfile/src/sndfile.c\",\n \"start_line\": 440,\n \"end_line\":
450,\n \"confidence\": \"unlikely\",\n \"reason\": \"sf_open_virtual may
invoke incorrect handling of virtual files; less likely to be root cause
of negative sizes.\",\n \"symbol\": \"sf_open_virtual\"\n },\n {\n
\"file_path\":
\"/src/llvm-project/compiler-rt/lib/fuzzer/FuzzerMain.cpp\",\n
\"start_line\": 20,\n \"end_line\": 30,\n \"confidence\":
\"very_unlikely\",\n \"reason\": \"main function initializes framework;
unlikely to cause negative-size issues; more setup than core logic.\",\n
\"symbol\": \"main\"\n },\n {\n \"file_path\":
\"/src/libsndfile/src/file_io.c\",\n \"start_line\": 311,\n \"end_line\":
321,\n \"confidence\": \"very_unlikely\",\n \"reason\": \"psf_fread deals
with reading data; highly indirect relation to the negative size issue
during memcpy.\",\n \"symbol\": \"psf_fread\"\n }]",
merge_with_last=false)

**Thought**

Executor#0 result: extract_json

**Observation**

{"ok": true, "json": {"llm_predictions": [{"file_path": "/src/libsndfile/ossfuzz/sndfile_fuzzer.cc", "start_line": 54, "end_line": 60, "confidence": "very_likely", "reason": "vfread uses memcpy with a negative size, indicating invalid parameters possibly due to bad input or internal state.", "symbol": "vfread"}, {"file_path": "/src/libsndfile/src/common.c", "start_line": 857, "end_line": 869, "confidence": "likely", "reason": "header_read might be calculating sizes based on invalid or corrupted header, leading to negative size in subsequent functions.", "symbol": "header_read"}, {"file_path": "/src/libsndfile/src/caf.c", "start_line": 395, "end_line": 404, "confidence": "likely", "reason": "caf_read_header initializes data from inputs, passing potentially invalid info leading to negative size errors down the call stack.", "symbol": "caf_read_header"}, {"file_path": "/src/libsndfile/src/sndfile.c", "start_line": 3180, "end_line": 3190, "confidence": "possible", "reason": "psf_open_file deals with file header initialization, might indirectly influence memory sizes that could be negative.", "symbol": "psf_open_file"}, {"file_path": "/src/libsndfile/src/common.c", "start_line": 988, "end_line": 998, "confidence": "possible", "reason": "psf_binheader_readf processes binary headers; corrupt input can lead to negative sizes in vfread.", "symbol": "psf_binheader_readf"}, {"file_path": "/src/libsndfile/src/caf.c", "start_line": 127, "end_line": 137, "confidence": "possible", "reason": "caf_open potentially handles opening invalid files or inconsistent states leading to negative size calculations.", "symbol": "caf_open"}, {"file_path": "/src/llvm-project/compiler-rt/lib/fuzzer/FuzzerLoop.cpp", "start_line": 559, "end_line": 569, "confidence": "unlikely", "reason": "Fuzzer's execution callback is related but indirect; actual fault likely lies in file handling logic instead.", "symbol": "ExecuteCallback"}, {"file_path": "/src/libsndfile/src/sndfile.c", "start_line": 440, "end_line": 450, "confidence": "unlikely", "reason": "sf_open_virtual may invoke incorrect handling of virtual files; less likely to be root cause of negative sizes.", "symbol": "sf_open_virtual"}, {"file_path": "/src/llvm-project/compiler-rt/lib/fuzzer/FuzzerMain.cpp", "start_line": 20, "end_line": 30, "confidence": "very_unlikely", "reason": "main function initializes framework; unlikely to cause negative-size issues; more setup than core logic.", "symbol": "main"}, {"file_path": "/src/libsndfile/src/file_io.c", "start_line": 311, "end_line": 321, "confidence": "very_unlikely", "reason": "psf_fread deals with reading data; highly indirect relation to the negative size issue during memcpy.", "symbol": "psf_fread"}], "raw": "[\n {\"llm_predictions\": [{\"file_path\": \"/src/libsndfile/ossfuzz/sndfile_fuzzer.cc\", \"start_line\": 54, \"end_line\": 60, \"confidence\": \"very_likely\", \"reason\": \"vfread uses memcpy with a negative size, indicating invalid parameters possibly due to bad input or internal state.\", \"symbol\": \"vfread\"}, {\"file_path\": \"/src/libsndfile/src/common.c\", \"start_line\": 857, \"end_line\": 869, \"confidence\": \"likely\", \"reason\": \"header_read might be calculating sizes based on invalid or corrupted header, leading to negative size in subsequent functions.\", \"symbol\": \"header_read\"}, {\"file_path\": \"/src/libsndfile/src/caf.c\", \"start_line\": 395, \"end_line\": 404, \"confidence\": \"likely\", \"reason\": \"caf_read_header initializes data from inputs, passing potentially invalid info leading to negative size errors down the call stack.\", \"symbol\": \"caf_read_header\"}, {\"file_path\": \"/src/libsndfile/src/sndfile.c\", \"start_line\": 3180, \"end_line\": 3190, \"confidence\": \"possible\", \"reason\": \"psf_open_file deals with file header initialization, might indirectly influence memory sizes that could be negative.\", \"symbol\": \"psf_open_file\"}, {\"file_path\": \"/src/libsndfile/src/common.c\", \"start_line\": 988, \"end_line\": 998, \"confidence\": \"possible\", \"reason\": \"psf_binheader_readf processes binary headers; corrupt input can lead to negative sizes in vfread.\", \"symbol\": \"psf_binheader_readf\"}, {\"file_path\": \"/src/libsndfile/src/caf.c\", \"start_line\": 127, \"end_line\": 137, \"confidence\": \"possible\", \"reason\": \"caf_open potentially handles opening invalid files or inconsistent states leading to negative size calculations.\", \"symbol\": \"caf_open\"}, {\"file_path\": \"/src/llvm-project/compiler-rt/lib/fuzzer/FuzzerLoop.cpp\", \"start_line\": 559, \"end_line\": 569, \"confidence\": \"unlikely\", \"reason\": \"Fuzzer's execution callback is related but indirect; actual fault likely lies in file handling logic instead.\", \"symbol\": \"ExecuteCallback\"}, {\"file_path\": \"/src/libsndfile/src/sndfile.c\", \"start_line\": 440, \"end_line\": 450, \"confidence\": \"unlikely\", \"reason\": \"sf_open_virtual may invoke incorrect handling of virtual files; less likely to be root cause of negative sizes.\", \"symbol\": \"sf_open_virtual\"}, {\"file_path\": \"/src/llvm-project/compiler-rt/lib/fuzzer/FuzzerMain.cpp\", \"start_line\": 20, \"end_line\": 30, \"confidence\": \"very_unlikely\", \"reason\": \"main function initializes framework; unlikely to cause negative-size issues; more setup than core logic.\", \"symbol\": \"main\"}, {\"file_path\": \"/src/libsndfile/src/file_io.c\", \"start_line\": 311, \"end_line\": 321, \"confidence\": \"very_unlikely\", \"reason\": \"psf_fread deals with reading data; highly indirect relation to the negative size issue during memcpy.\", \"symbol\": \"psf_fread\"}]}", "json_path": "/home/bug_localization_agents/work_arvo/25357/llm_output.tmp.json"}

**Thought**

Executor#0 call: compare_llm_metrics

$
compare_llm_metrics(llm_json_path="/home/bug_localization_agents/work_arvo
25357/llm_batch_0001.json",
chunks_json_path="/home/bug_localization_agents/work_arvo/25357
case_chunks.json",
diff_index_json_path="/home/bug_localization_agents/work_arvo/25357
diff_index.json", line_slack=0)

**Thought**

Executor#0 result: compare_llm_metrics

Executor#0 result: compare_llm_metrics

**Observation**

{"ok": true, "detection_rate": 0.6666666666666666, "localization_rate": 0.2916666666666667, "strict_localization_rate": 0.0, "inputs": {"llm_json_path": "/home/bug_localization_agents/work_arvo/25357/llm_batch_0001.json", "chunks_json_path": "/home/bug_localization_agents/work_arvo/25357/case_chunks.json", "diff_index_json_path": "/home/bug_localization_agents/work_arvo/25357/diff_index.json"}, "detection": {"denom_diff_chunks": 3, "num_detected_and_diff": 2, "rate": 0.6666666666666666, "updated_chunks": true}, "localization": {"total_lines": 24, "matched_lines": 7, "newly_matched": 7, "rate": 0.2916666666666667}, "strict_localization": {"total_groups": 6, "matched_groups": 0, "rate": 0.0}, "saved_snapshot_error": "'compare_history'"}

**Thought**

Executor#0 call: view_source

```
$ view_source(file_path="/src/libsndfile/ossfuzz/sndfile_fuzzer.cc",
start_line=14, end_line=80, max_bytes=200000)
```

**Thought**

Executor#0 result: view_source

**Observation**

{"ok": true, "file_path": "/src/libsndfile/ossfuzz/sndfile_fuzzer.cc", "start_line": 14, "end_line": 80, "content": "14 static sf_count_t vfget_filelen (void *user_data)\n15 {\n16 VIO_DATA *vf = (VIO_DATA *)user_data;\n17 return vf->length;\n18 }\n19 \n20 static sf_count_t vfseek (sf_count_t offset, int whence, void *user_data)\n21 {\n22 VIO_DATA *vf = (VIO_DATA *)user_data;\n23 \n24 switch (whence)\n25 {\n26 case SEEK_SET:\n27 vf->offset = offset;\n28 break ;\n29 \n30 case SEEK_CUR:\n31 vf->offset = vf->offset + offset;\n32 break ;\n33 \n34 case SEEK_END:\n35 vf->offset = vf->length + offset;\n36 break;\n37 \n38 default:\n39 break;\n40 }\n41 \n42 return vf->offset;\n43 }\n44 \n45 static sf_count_t vfread (void *ptr, sf_count_t count, void *user_data)\n46 {\n47 VIO_DATA *vf = (VIO_DATA *)user_data;\n48 \n49 if (vf->offset + count > vf->length)\n50 {\n51 count = vf->length - vf->offset;\n52 }\n53 \n54 memcpy(ptr, vf->data + vf->offset, count);\n55 vf->offset += count;\n56 \n57 return count;\n58 }\n59 \n60 static sf_count_t vfwrite (const void *ptr, sf_count_t count, void *user_data)\n61 {\n62 (void)ptr;\n63 (void)count;\n64 (void)user_data;\n65 \n66 // Cannot write to this virtual file.\n67 return 0;\n68 }\n69 \n70 static sf_count_t vftell (void *user_data)\n71 { VIO_DATA *vf = (VIO_DATA *)user_data;\n72 \n73 return vf->offset;\n74 }\n75 \n76 extern \"C\" int LLVMFuzzerTestOneInput(const uint8_t* data, size_t size)\n77 {\n78 VIO_DATA vio_data;\n79 SF_VIRTUAL_IO vio;\n80 SF_INFO sndfile_info;\n", "error": null}

**Thought**

Executor#0 call: view_source

```
$ view_source(file_path="/src/libsndfile/src/common.c", start_line=840,
end_line=880, max_bytes=200000)
```

**Thought**

Executor#0 result: view_source

**Observation**

{"ok": true, "file_path": "/src/libsndfile/src/common.c", "start_line": 840, "end_line": 880, "content": "840 \t\t\t\t\t\t\t\t(((sf_count_t) (ptr) [1]) << 8)\t\t| ((ptr) [0]))\n841 \n842 #define\tGET_BE_8BYTE(ptr)\t(\t(((sf_count_t) (ptr) [0]) << 56)\t| (((sf_count_t) (ptr) [1]) << 48) |\t\\\n843 \t\t\t\t\t\t\t\t(((sf_count_t) (ptr) [2]) << 40)\t| (((sf_count_t) (ptr) [3]) << 32) |\t\\\n844 \t\t\t\t\t\t\t\t(((sf_count_t) (ptr) [4]) << 24)\t| (((sf_count_t) (ptr) [5]) << 16) |\t\\\n845 \t\t\t\t\t\t\t\t(((sf_count_t) (ptr) [6]) << 8)\t\t| ((ptr) [7]))\n846 \n847 \n848 \n849 static int\n850 header_read (SF_PRIVATE *psf, void *ptr, int bytes)\n851 {\tint count = 0 ;\n852 \n853 \tif (psf->header.indx + bytes >= psf->header.len && psf_bump_header_allocation (psf, bytes))\n854 \t\treturn count ;\n855 \n856 \tif (psf->header.indx + bytes > psf->header.end)\n857 \t{\tcount = psf_fread (psf->header.ptr + psf->header.end, 1, bytes - (psf->header.end - psf->header.indx), psf) ;\n858 \t\tif (count != bytes - (int) (psf->header.end - psf->header.indx))\n859 \t\t{\tpsf_log_printf (psf, \"Error : psf_fread returned short count.\\n\") ;\n860 \t\t\treturn count ;\n861 \t\t\t} ;\n862 \t\tpsf->header.end += count ;\n863 \t\t} ;\n864 \n865 \tmemcpy (ptr, psf->header.ptr + psf->header.indx, bytes) ;\n866 \tpsf->header.indx += bytes ;\n867 \n868 \treturn bytes ;\n869 } /* header_read */\n870 \n871 static void\n872 header_seek (SF_PRIVATE *psf, sf_count_t position, int whence)\n873 {\n874 \tswitch (whence)\n875 \t{\tcase SEEK_SET :\n876 \t\t\tif (psf->header.indx + position >= psf->header.len)\n877 \t\t\t\tpsf_bump_header_allocation (psf, position) ;\n878 \t\t\tif (position > psf->header.len)\n879 \t\t\t{\t/* Too much header to cache so just seek instead. */\n880 \t\t\t\tpsf_fseek (psf, position, whence) ;\n", "error": null}

**Thought**

Executor#0 call: view_source

```
$ view_source(file_path="/src/libsndfile/src/caf.c", start_line=385,
end_line=415, max_bytes=200000)
```

**Thought**

Executor#0 result: view_source

**Observation**

{"ok": true, "file_path": "/src/libsndfile/src/caf.c", "start_line": 385, "end_line": 415, "content": "385 \n386 \tif (chunk_size > SIGNED_SIZEOF (DESC_CHUNK))\n387 \t\tpsf_binheader_readf (psf, \"j\", (int) (chunk_size - sizeof (DESC_CHUNK))) ;\n388 \n389 \tpsf->sf.channels = desc.channels_per_frame ;\n390 \n391 \twhile (1)\n392 \t{\tmarker = 0 ;\n393 \t\tchunk_size = 0 ;\n394 \n395 \t\tpsf_binheader_readf (psf, \"mE8\", &marker, &chunk_size) ;\n396 \t\tif (marker == 0)\n397 \t\t{\tsf_count_t pos = psf_ftell (psf) ;\n398 \t\t\tpsf_log_printf (psf, \"Have 0 marker at position %D (0x%x).\\n\", pos, pos) ;\n399 \t\t\tbreak ;\n400 \t\t\t} ;\n401 \t\tif (chunk_size < 0)\n402 \t\t{\tpsf_log_printf (psf, \"%M : %D *** Should be >= 0 ***\\n\", marker, chunk_size) ;\n403 \t\t\tbreak ;\n404 \t\t\t} ;\n405 \t\tif (chunk_size > psf->filelength)\n406 \t\t\tbreak ;\n407 \n408 \t\tpsf_store_read_chunk_u32 (&psf->rchunks, marker, psf_ftell (psf), chunk_size) ;\n409 \n410 \t\tswitch (marker)\n411 \t\t{\tcase peak_MARKER :\n412 \t\t\t\tpsf_log_printf (psf, \"%M : %D\\n\", marker, chunk_size) ;\n413 \t\t\t\tif (chunk_size != CAF_PEAK_CHUNK_SIZE (psf->sf.channels))\n414 \t\t\t\t{\tpsf_binheader_readf (psf, \"j\", make_size_t (chunk_size)) ;\n415 \t\t\t\t\tpsf_log_printf (psf, \"*** File PEAK chunk %D should be %d.\\n\", chunk_size, CAF_PEAK_CHUNK_SIZE (psf->sf.channels)) ;\n", "error": null}

**Thought**

Executor#0 call: llm_analyze

```
$ llm_analyze(refinement_bundle={"source_slices":
{"/src/libsndfile/ossfuzz/sndfile_fuzzer.cc": {"start_line": 54,
"end_line": 60}, "/src/libsndfile/src/common.c": {"start_line": 857,
"end_line": 869}, "/src/libsndfile/src/caf.c": {"start_line": 395,
"end_line": 404}}}, top_n=5)
```

**Thought**

Executor#0 result: llm_analyze

**Observation**

{"ok": true, "mode": "refine", "text": "[\n {\n \"file_path\": \"libsndfile/ossfuzz/sndfile_fuzzer.cc\",\n \"start_line\": 54,\n \"end_line\": 58,\n \"confidence\": \"very_likely\",\n \"reason\": \"Slice includes vfread which triggers the negative-size-param error.\",\n \"symbol\": \"vfread\"\n },\n {\n \"file_path\": \"libsndfile/src/common.c\",\n \"start_line\": 857,\n \"end_line\": 869,\n \"confidence\": \"possible\",\n

\"reason\": \"Slice indicates potential error in psf_fread, may relate to vfread usage.\",\n \"symbol\": \"header_read\"\n },\n {\n \"file_path\": \"libsndfile/src/caf.c\",\n \"start_line\": 395,\n \"end_line\": 404,\n \"confidence\": \"possible\",\n \"reason\": \"Slice shows boundary checks, relates to psf_binheader_readf which may lead into vfread.\",\n \"symbol\": \"caf_read_header\"\n }\n]", "logged": {"jsonl_path": "/home/bug_localization_agents/work_arvo/25357/predictions.jsonl", "batch_json_path": "/home/bug_localization_agents/work_arvo/25357/llm_batch_0002.json", "cumulative_path": "/home/bug_localization_agents/work_arvo/25357/cumulative_predictions.json", "count": 3, "batch": 2}}

**Thought**

Executor#0 call: extract_json

```
$ extract_json(response_text="[\n {\n \"file_path\":
\"libsndfile/ossfuzz/sndfile_fuzzer.cc\",\n \"start_line\": 54,\n
\"end_line\": 58,\n \"confidence\": \"very_likely\",\n \"reason\": \"Slice
includes vfread which triggers the negative-size-param error.\",\n
\"symbol\": \"vfread\"\n },\n {\n \"file_path\":
\"libsndfile/src/common.c\",\n \"start_line\": 857,\n \"end_line\": 869,\n
\"confidence\": \"possible\",\n \"reason\": \"Slice indicates potential
error in psf_fread, may relate to vfread usage.\",\n \"symbol\":
\"header_read\"\n },\n {\n \"file_path\": \"libsndfile/src/caf.c\",\n
\"start_line\": 395,\n \"end_line\": 404,\n \"confidence\":
\"possible\",\n \"reason\": \"Slice shows boundary checks, relates to
psf_binheader_readf which may lead into vfread.\",\n \"symbol\":
\"caf_read_header\"\n }\n]", merge_with_last=false)
```

**Thought**

Executor#0 result: extract_json

**Observation**

{"ok": true, "json": {"llm_predictions": [{"file_path": "libsndfile/ossfuzz/sndfile_fuzzer.cc", "start_line": 54, "end_line": 58, "confidence": "very_likely", "reason": "Slice includes vfread which triggers the negative-size-param error.", "symbol": "vfread"}, {"file_path": "libsndfile/src/common.c", "start_line": 857, "end_line": 869, "confidence": "possible", "reason": "Slice indicates potential error in psf_fread, may relate to vfread usage.", "symbol": "header_read"}, {"file_path": "libsndfile/src/caf.c", "start_line": 395, "end_line": 404, "confidence": "possible", "reason": "Slice shows boundary checks, relates to psf_binheader_readf which may lead into vfread.", "symbol": "caf_read_header"}]}, "raw": "{\"llm_predictions\": [{\"file_path\": \"libsndfile/ossfuzz/sndfile_fuzzer.cc\", \"start_line\": 54, \"end_line\": 58, \"confidence\": \"very_likely\", \"reason\": \"Slice includes vfread which triggers the negative-size-param error.\", \"symbol\": \"vfread\"}, {\"file_path\": \"libsndfile/src/common.c\", \"start_line\": 857, \"end_line\": 869, \"confidence\": \"possible\", \"reason\": \"Slice indicates potential error in psf_fread, may relate to vfread usage.\", \"symbol\": \"header_read\"}, {\"file_path\": \"libsndfile/src/caf.c\", \"start_line\": 395, \"end_line\": 404, \"confidence\": \"possible\", \"reason\": \"Slice shows boundary checks, relates to psf_binheader_readf which may lead into vfread.\", \"symbol\": \"caf_read_header\"}]}", "json_path": "/home/bug_localization_agents/work_arvo/25357/llm_output.tmp.json"}

**Thought**

Executor#0 call: compare_llm_metrics

```
$
compare_llm_metrics(llm_json_path="/home/bug_localization_agents/work_arvo
25357/llm_batch_0002.json",
chunks_json_path="/home/bug_localization_agents/work_arvo/25357
case_chunks.json",
diff_index_json_path="/home/bug_localization_agents/work_arvo/25357
diff_index.json", line_slack=0)
```

**Thought**

Executor#0 result: compare_llm_metrics

**Observation**

{"ok": true, "detection_rate": 0.6666666666666666, "localization_rate": 0.2916666666666667, "strict_localization_rate": 0.0, "inputs": {"llm_json_path": "/home/bug_localization_agents/work_arvo/25357/llm_batch_0002.json", "chunks_json_path": "/home/bug_localization_agents/work_arvo/25357/case_chunks.json", "diff_index_json_path": "/home/bug_localization_agents/work_arvo/25357/diff_index.json"}, "detection": {"denom_diff_chunks": 3, "num_detected_and_diff": 2, "rate": 0.6666666666666666, "updated_chunks": false}, "localization": {"total_lines": 24, "matched_lines": 7, "newly_matched": 0, "rate": 0.2916666666666667}, "strict_localization": {"total_groups": 6, "matched_groups": 0, "rate": 0.0}, "saved_snapshot_error": "'compare_history'"}

**Thought**

Executor#0 call: llm_analyze

```
$ llm_analyze()
```

**Thought**

Executor#0 result: llm_analyze

**Observation**

{"ok": true, "mode": "crash", "text": "[\n {\n \"file_path\": \"/src/libsndfile/ossfuzz/sndfile_fuzzer.cc\",\n \"start_line\": 50,\n \"end_line\": 60,\n \"confidence\": \"very_likely\",\n \"reason\": \"Crashed in memcpy due to negative size from vfread; investigate size handling in vfread.\",\n \"symbol\": \"vfread\"\n },\n {\n \"file_path\": \"/src/libsndfile/src/file_io.c\",\n \"start_line\": 300,\n \"end_line\": 320,\n \"confidence\": \"likely\",\n \"reason\": \"Error passed from vfread indicates possible flaws in file reading logic, check variable sizes before memcpy.\",\n \"symbol\": \"psf_fread\"\n },\n {\n \"file_path\": \"/src/libsndfile/src/common.c\",\n \"start_line\": 850,\n \"end_line\": 870,\n \"confidence\": \"likely\",\n \"reason\": \"Header read process may set up incorrect sizes; high potential for parameters affecting memcpy size.\",\n \"symbol\": \"header_read\"\n },\n {\n \"file_path\": \"/src/libsndfile/src/common.c\",\n \"start_line\": 980,\n \"end_line\": 1000,\n \"confidence\": \"possible\",\n \"reason\": \"Function likely prepares data read for copying; correlation with the error seen in line 54 of vfread.\",\n \"symbol\": \"psf_binheader_readf\"\n },\n {\n \"file_path\": \"/src/libsndfile/src/caf.c\",\n \"start_line\": 390,\n \"end_line\": 410,\n \"confidence\": \"possible\",\n \"reason\": \"CAF header reading; inconsistencies there may impact following reads.\",\n \"symbol\": \"caf_read_header\"\n },\n {\n \"file_path\": \"/src/libsndfile/src/caf.c\",\n \"start_line\": 120,\n \"end_line\": 140,\n \"confidence\": \"possible\",\n \"reason\": \"Opening of CAF files likely interacts with header reads; check for corrupted or malformed inputs.\",\n \"symbol\": \"caf_open\"\n },\n {\n \"file_path\": \"/src/libsndfile/src/sndfile.c\",\n \"start_line\": 3170,\n \"end_line\": 3190,\n \"confidence\": \"possible\",\n \"reason\": \"Function interfaces with file I/O procedures; errors could propagate affecting input sizes.\",\n \"symbol\": \"psf_open_file\"\n },\n {\n \"file_path\": \"/src/libsndfile/ossfuzz/sndfile_fuzzer.cc\",\n \"start_line\": 95,\n \"end_line\": 105,\n \"confidence\": \"unlikely\",\n \"reason\": \"Test case execution context could contribute; minimal impact if fault is in called functions.\",\n \"symbol\": \"LLVMFuzzerTestOneInput\"\n },\n {\n \"file_path\": \"/src/llvm-project/compiler-rt/lib/fuzzer/FuzzerLoop.cpp\",\n \"start_line\": 550,\n \"end_line\": 570,\n \"confidence\": \"very_unlikely\",\n \"reason\": \"Fuzzer's execution callback may process inputs incorrectly, though main errors stem from lower functions.\",\n \"symbol\": \"fuzzer::Fuzzer::ExecuteCallback\"\n },\n {\n \"file_path\": \"/src/llvm-project/compiler-rt/lib/fuzzer/FuzzerDriver.cpp\",\n \"start_line\": 295,\n \"end_line\": 315,\n \"confidence\": \"very_unlikely\",\n \"reason\": \"Driver setups likely; not directly implicated in memory size error from read functions.\",\n \"symbol\": \"fuzzer::FuzzerDriver\"\n }\n]", "logged": {"jsonl_path": "/home/bug_localization_agents/work_arvo/25357/predictions.jsonl", "batch_json_path": "/home/bug_localization_agents/work_arvo/25357/llm_batch_0003.json", "cumulative_path": "/home/bug_localization_agents/work_arvo/25357/cumulative_predictions.json", "count": 10, "batch": 3}}

**Thought**

Executor#0 call: extract_json

```
$ extract_json(response_text="[\n {\n \"file_path\":
```

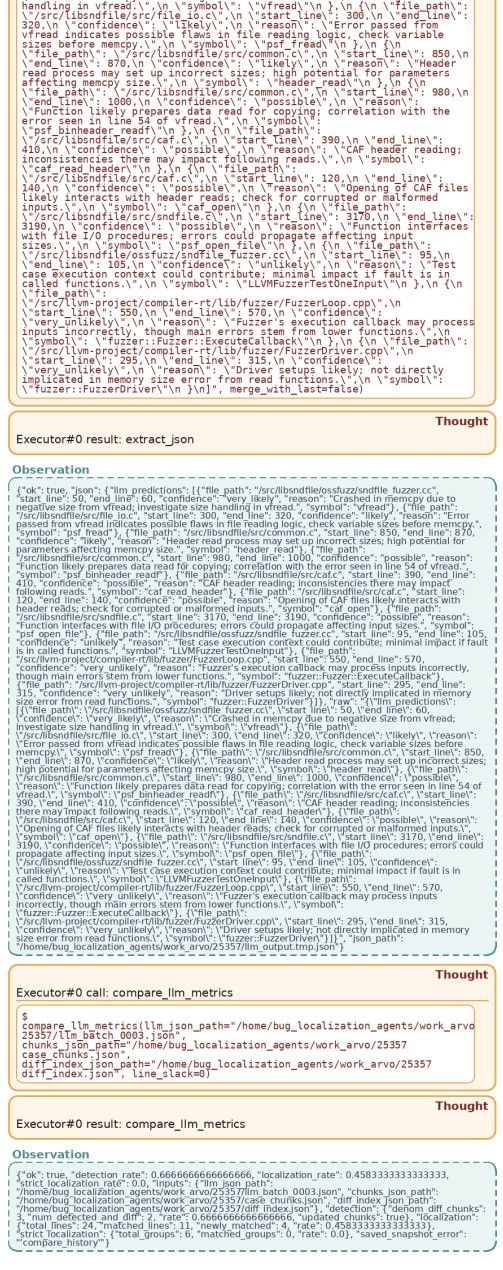

Figure 17: GPT-4o mini Divergence Tracing for case 25357.

{"ok": true, "json_path": "/home/bug_localization_agents/work_arvo/12679/diff_index.json", "num_files": 1, "num_hunks": 0, "num_per_line": 8}

**Prompt**

Task: Orchestrate ARVO bug localization and evaluation.

Follow this high-level pipeline:
(1) Choose REPO_ROOT:
- Prefer /src/freetype2 if it exists; otherwise /src/libreoffice.
(2) chunk_case(root_dir=REPO_ROOT, out_json_path="./case_chunks.json")
(3) diff_index(diff_path, out_json_path="./diff_index.json")
(4) mark_diff(diff_json_path="./diff_index.json",
chunks_json_path="./case_chunks.json")
(5) run_san(); parse binary path from sanitized ASAN output
(6) run_gdb(binary_path=...) ONLY IF (gdb exists) AND (binary is
executable) AND never triggers package managers
(7) Build and CACHE crash_log = sanitized ASAN + "\n\nBacktrace:\n" + (GDB
backtrace or "")
• If >200KB, keep head 50KB + full call stack section + tail 50KB, elide
middle.
(8) DELEGATE to the Executor. The delegated task MUST ask for:
- llm_analyze(crash_log=...) → extract_json(response_text)
• Prefer calling llm_analyze() with no args to consume the cached crash_log
from Planner.
- Optionally: view_source / grep_source around high-confidence lines →
build refinement_bundle
→ llm_analyze(refinement_bundle=...) → extract_json (ONE pass)
- If available: publish_verified_locations(..., repo_root=REPO_ROOT)
- compare_llm_metrics(diff_index_json_path="./diff_index.json",
llm_json_path=<verified or raw JSON>,
chunks_json_path="./case_chunks.json",
line_slack=0)
- Finally: finish_task(summary=<<include metrics + paths>>)

Output only tool calls or delegated tasks as needed. Keep text minimal.

---

**Thought**

Planner note

---

**Observation**

You are the PLANNER for ARVO bug localization. You orchestrate the pipeline, call *lightweight repo/indexing tools and runtime collection (ASAN/GDB)* yourself, and DELEGATE heavy reasoning/evaluation to the Executor via `delegate(task=...)`.

## Your toolset
- chunk_case(root_dir="/src/libreoffice", out_json_path="./case_chunks.json", include_context=0, max_file_bytes=1000000)
- diff_index(diff_path, out_json_path="./diff_index.json")
- mark_diff(diff_json_path="./diff_index.json", chunks_json_path="./case_chunks.json")
- run_san()
- run_gdb(binary_path="./vuln")
- delegate(task=<string>)
- giveup()

## Responsibilities
- OWN orchestration and state: start container once, choose REPO_ROOT, build chunks & diff index once, mark diff on chunks once (unless inputs changed).
- YOU run ASAN/GDB and cache crash_log in environment state.
- For LLM reasoning & evaluation: DELEGATE to the Executor with a clear, structured task that includes:
• What to analyze (use cached crash_log → llm_analyze → extract_json),
• Which files to use (paths to diff index, case_chunks.json),
• Whether to verify (publish_verified_locations if available) → compare_llm_metrics,
• Output contract (return metrics and paths in `finish_task(summary=...)`).
- Aggregate results. **Do not call `giveup()` after a single delegation.** Prefer at least 2-3 refinement rounds unless clearly hopeless.

## Repo root policy
- Set REPO_ROOT to an existing path in this order: /src/freetype2, else /src/libreoffice.
- Use that REPO_ROOT consistently when asking the Executor to verify locations.

## Caching policy
- Do NOT rebuild chunk / diff index / diff marks unless code/diff changed.
- Delegate tasks in *minimal increments*; avoid re-running heavy steps unnecessarily.

---

**Thought**

Planner note

---

**Observation**

---

**Thought**

Planner call: chunk_case

```
$ chunk_case(root_dir="/src/libreoffice",
out_json_path="/home/bug_localization_agents/work_arvo/12679
case_chunks.json")
```

---

**Thought**

Planner result: chunk_case

---

**Observation**

{"ok": true, "root_dir": "/src/libreoffice", "json_path": "/home/bug_localization_agents/work_arvo/12679/case_chunks.json", "num_files": 2160, "num_chunks": 18595}

---

**Thought**

Planner call: diff_index

```
$
diff_index(diff_path="/home/bug_localization_agents/ARVO-Meta/patches
12679.diff",
out_json_path="/home/bug_localization_agents/work_arvo/12679
diff_index.json")
```

---

**Thought**

Planner result: diff_index

---

**Observation**

---

**Thought**

Planner call: mark_diff

```
$
mark_diff(diff_json_path="/home/bug_localization_agents/work_arvo/12679
diff_index.json",
chunks_json_path="/home/bug_localization_agents/work_arvo/12679
case_chunks.json")
```

---

**Thought**

Planner result: mark_diff

---

**Observation**

{"ok": true, "chunks_json_path": "/home/bug_localization_agents/work_arvo/12679/case_chunks.json", "num_chunks": 18595, "num_chunks_with_diff": 1}

---

**Thought**

Planner call: run_san

```
$ run_san()
```

---

**Thought**

Planner result: run_san

---

**Observation**

{"ok": true, "asan_output": {"ok": true, "asan_output": "====================== INFO
====================\r\nThis binary is built for AFL-fuzz\r\nTo run the target
function on individual input(s) execute this:\r\n /out/cli-uart-received-fuzzer < INPUT_FILE\r\nor\r\n
/out/cli-uart-received-fuzzer INPUT_FILE1 [INPUT_FILE2 ...]\r\nTo fuzz with afl-fuzz execute this:\r\n
afl-fuzz [afl-flags] /out/cli-uart-received-fuzzer [-N]\r\nafl-fuzz will run N iterations before re-spawning
the process (default:
1000)\r\n=================================================\r\n
Reading 79 bytes from
tmp
poc\r\n==================================================================
=======\r\n\u001b[1m\u001b[31m==17370==ERROR: AddressSanitizer: memcpy-param-overlap:
memory ranges [0x000012f9873,0x000012f98b1) and [0x000012f98b0, 0x000012f98ee)
overlap\r\n\u001b[1m\u001b[0mSCARINESS: 10 (memcpy-param-overlap)\r\n #0 0x4edd79 in
__asan_memcpy /src/llvm/projects/compiler-rt/lib/asan/asan_interceptors_memintrinsics.cc:23\r\n #1
0x553a6b in ot::Cli::CoapSecure::Process(int, char**)
/src/openthread/src/cli/cli_coap_secure.cpp:175:21\r\n #2 0x53a88e in
ot::Cli::Interpreter::ProcessCoapSecure(int, char**) /src/openthread/src/cli/cli.cpp:781:25\r\n #3
0x550702 in ot::Cli::Interpreter::ProcessLine(char*, unsigned short, ot::Cli::Server&)
/src/openthread/src/cli/cli.cpp:3775:13\r\n #4 0x536318 in ot::Cli::Uart::ProcessCommand()
/src/openthread/src/cli/cli_uart.cpp:259:22\r\n #5 0x535f8b in ot::Cli::Uart::ReceiveTask(unsigned char
const*, unsigned short) /src/openthread/src/cli/cli_uart.cpp:178:17\r\n #6 0x534e90 in
LLVMFuzzerTestOneInput /src/openthread/tests/fuzz/cli_uart_received.cpp:66:5\r\n #7 0x67d75a in
ExecuteFilesOnyByOne(int, char**) /src/libfuzzer/afl/afl_driver.cpp:300:5\r\n #8 0x67dcbe in main
/src/libfuzzer/afl/afl_driver.cpp:338:12\r\n #9 0x7f552b6bbb8f in __libc_start_main
(/lib/x86_64-linux-gnu/libc.so.6+0x2083f)\r\n #10 0x41ce78 in _start
(/out/cli-uart-received-fuzzer+0x41ce78)\r\n\r\nDEDUP_TOKEN:
__asan_memcpy--ot::Cli::CoapSecure::Process(int, char**)--ot::Cli::Interpreter::ProcessCoapSecure(int,
char**)\r\n\u001b[1m\u001b[32m0x000012f9873 is located 595 bytes inside of global variable
'ot::Cli::sCliUartRaw' defined in 'cli_uart.cpp:90:8' (0x12f9620) of size
2184\r\n\u001b[1m\u001b[0m\u001b[32m0x000012f98b0 is located 656 bytes inside of
global variable 'ot::Cli::sCliUartRaw' defined in 'cli_uart.cpp:90:8' (0x12f9620) of size
2184\r\n\u001b[1m\u001b[0mSUMMARY: AddressSanitizer: memcpy-param-overlap
/src/llvm/projects/compiler-rt/lib/asan/asan_interceptors_memintrinsics.cc:23 in
__asan_memcpy\r\n==17370==ABORTING", "binary_path": null, "cid": "2d9d1b41ea26"}}

---

**Thought**

Planner call: run_gdb

```
$ run_gdb(binary_path="/out/cli-uart-received-fuzzer")
```

---

**Thought**

Planner result: run_gdb

---

**Observation**

{"ok": true, "skipped": true, "reason": "gdb not found in container; skipping", "binary_path": "/out/cli-uart-received-fuzzer"}

---

**Thought**

Planner call: delegate

```
$ delegate(task="llm_analyze() → extract_json")
```

---

**Thought**

Planner result: delegate

---

**Observation**

"this round has not found the bug, please continue based on your best judge"

---

**Thought**

Planner call: delegate

```
$ delegate(task="llm_analyze() → extract_json; view_source around lines of
interest; publish_verified_locations(repo_root='/src/libreoffice');
compare_llm_metrics(diff_index_json_path='./diff_index.json',
llm_json_path='raw_json', chunks_json_path='./case_chunks.json',
line_slack=0); finish_task(summary='provide aggregated metrics and paths
for detailed review.')")
```

---

**Thought**

Planner result: delegate

---

**Observation**

"this round has not found the bug, please continue based on your best judge"

---

**Thought**

Planner call: giveup

```
$ giveup(confirm=true)
```

---

**Thought**

Planner result: giveup

---

**Observation**

{"success": true}

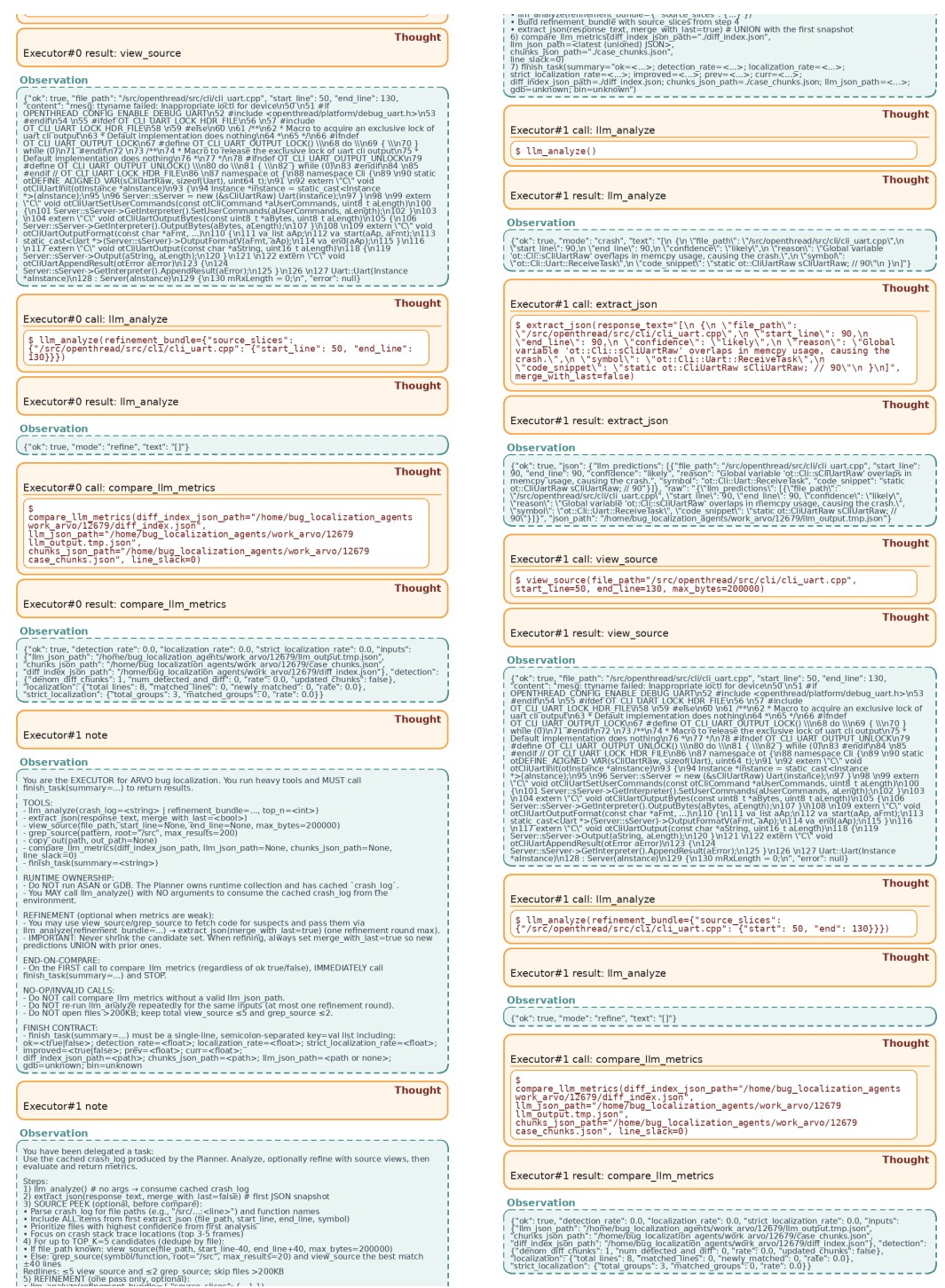

Figure 18: GPT-4o mini Baseline for case 12679.

