# OpenReview forum: "From Trace to Line: LLM Agent for Real-World OSS Vulnerability Localization"
_ICLR.cc/2026/Conference — Submitted to ICLR 2026_

### Official Review · Reviewer_3t1k · 2025-10-14

**Soundness:** 2
**Presentation:** 2
**Contribution:** 3
**Rating:** 2
**Confidence:** 4

**Summary:**

This paper addresses line-level bug localization, proposing both a benchmark of 50 manually annotated real-world cases and a multi-stage T2L-Agent framework that combines dynamic and static analysis with LLM reasoning.

The agent achieves 58% chunk-level detection and 54.8% line-level localization accuracy.
T2L-Agent consists of a Trace Analyzer that collects crash and runtime evidence using sanitizers / debuggers, and a Detection module where an LLM generates and iteratively tests root-cause hypotheses.
The benchmark is derived from the ARVO dataset of 4 900 reproducible C/C++ vulnerabilities and provides line-level ground truth and LLM-estimated difficulty labels.

**Strengths:**

1. This paper focuses on line-level bug localization, which is a critical stage bridging bug analysis and bug repair. The ability to accurately locate the faulty line provides essential context for debugging, program understanding, and subsequent automated repair tasks.

2. The authors construct a fine-grained benchmark based on real-world projects to evaluate line-level localization performance.
This benchmark includes detailed annotations from real C/C++ vulnerabilities, providing valuable data for the community to measure progress in this direction.

3. The proposed T2L-Agent framework adopts a multi-stage design that integrates multiple types of static and dynamic analysis tools, including sanitizers and debuggers. Such integration enriches the runtime and structural information available to the LLM, effectively assisting in identifying root-cause locations and improving localization precision.

**Weaknesses:**

1. The Approach section is not clearly written. It lacks a step-by-step description of what each stage takes as input and produces as output, and several key terms—such as crash signature and dynamic evidence graph—are not explicitly defined.
Furthermore, the example figures meant to help readers understand the workflow are difficult to follow: the visualizations remain in their original JSON format and lack explanations of individual fields, which can easily cause readers to get lost.

2. The definition of the benchmark’s ground truth is missing. Since the paper’s central goal is line-level localization, it is crucial to clarify how the “ground-truth line number” is derived. Without this explanation, it is hard to assess how faithfully the evaluation reflects real localization accuracy.

3. The evaluation lacks comparison with other agentic solutions. The experiments only include ablation studies, which are insufficient to contextualize performance. It would be more convincing to include comparisons with existing LLM-based agentic systems capable of bug localization, such as SWE-Agent (Zhang et al., 2024).

**Questions:**

### Agent Framework

**Trace Analyzer**
- What exactly is the output of this analyzer? Is it the *crash signature*? If so, how is that signature obtained or synthesized?
- How are the **chunks** in the codebase defined or segmented?
- What is the **dynamic evidence graph**? How does it correlate runtime observations with static code features?
- Figure 4 is intended to provide examples for readers to understand the workflow, yet the figure itself is difficult to interpret. The results remain in the original JSON format without explanations of the fields, and the tools invoked in the examples are not described. Although the appendix lists the toolkits used in the experiments, the main text and figures provide no reference or explanation, making them hard to follow.

**Detection**
- The method description states that it *“selects source code slices based on crash signatures, stack trace information, and vulnerability patterns.”* How is this selection performed concretely? The authors should first clarify what outputs are produced by the Trace Analyzer and then explain how the Detection stage consumes these outputs.
- What is the **termination condition** of the iterative detection process? Is it based on confidence thresholds, hypothesis convergence, or a fixed iteration limit?

### Benchmark
- How is the **ground-truth line number** defined? The paper states that “Localization requires exact line matches to ground-truth patches,” but for a given bug, there may exist multiple valid patch locations. The patch itself may not uniquely represent the *root-cause line*. Please clarify whether the ground truth corresponds to the first faulty statement, the patched line, or a specific heuristic rule.

### Evaluation
- In Table 3, what does “Runtime = 20.2g” mean? Is this a typo or a specific metric unit?
- Sections 6.1 and 6.2 essentially constitute an **ablation study**, but the evaluation still lacks comparison with other **agentic baselines** capable of bug localization. Ideally, the baseline section should include comparisons to systems like **SWE-Agent**, which are designed for similar reasoning-driven localization tasks.
- The *best results* reported in the abstract (58.0% detection and 54.8% line-level localization) appear only in Table 2 (ablation). It would be clearer and more impactful to highlight these key results earlier in the main tables. Moreover, Table 1 does not explicitly indicate whether the reported numbers belong to baselines or to T2L-Agent itself, which can be misleading and make it appear as though they represent the best results of the proposed approach.

---

> ### Author Response · Authors · 2025-11-21
>
> 1. The Trace Analyzer does not synthesize a “crash signature.” Its output is simply the raw runtime evidence collected during execution—crash traces, sanitizer logs, and relevant static alerts—normalized into a unified textual form. One of the core challenges of the task we introduce is enabling the agent to understand binary programs in an encapsulated environment, together with their execution traces. As shown in Table 2, the trace analyzer is essential to this new setting because current LLM-based vulnerability detection methods focus almost entirely on plain source-code inputs. By shifting the problem toward binary-level reasoning enriched with runtime signals, our work not only proposes a new agent but also defines a new task formulation and demonstrates how agents can operate effectively under this more realistic and demanding scenario. These signals are passed directly to the LLM; the analyzer performs only evidence collection and cleanup, not signature generation. The crash trace itself is obtained from executing the buggy program under the corresponding runtime environment, not produced or inferred by the analyzer.
>
> 2. Chunks are not arbitrary line windows. As described in §4.1 and §4.2, we first parse the repository with Tree-sitter and perform code-structure analysis that “partitions the codebase into function-aligned, semantically coherent units that preserve syntactic relationships while fitting LLM context windows” and “semantically meaningful chunks” for precise slice extraction. The chunk_case tool stores each chunk as a JSON entry with file_path, symbol, and start/end_line fields, derived from AST nodes. Thus, chunk boundaries follow syntactic units like functions, and are sized to LLM context constraints; we do not randomly cut across functions.
>
> 3. The “dynamic evidence graph” refers to how runtime observations relate to each other and to the source locations explicitly referenced in those logs.
>
> 4. For figure 4, we will annotate each JSON field with brief descriptions, label the tools invoked in each step of the example pipeline, and include inline references to the corresponding components introduced in §4.1 so readers can easily follow the workflow without relying on the appendix.
>
> 5. The Trace Analyzer produces raw runtime evidence collected from executing the buggy program: stack traces, sanitizer logs, and error messages. These outputs are normalized into a unified text block.
> In the Detection stage, the LLM receives this combined runtime evidence together with static code context and is asked to identify which source regions are implicated by the observed crash behavior. The model uses coordinates already present in the runtime evidence, such as the call stack’s file: line references, to decide which code slices to examine further. Thus, slice selection is performed directly by the LLM based on the evidence it sees. We will clarify both the Trace Analyzer outputs and the Detection stage’s consumption of them in the revision.
>
> 6. The refinement loop terminates under two conditions: (1) when the model stops generating any new candidate locations, or when all known ground-truth locations for the sample have been discovered; and (2) when the cost budget $1 per case is reached. Thus, the process ends naturally; we will clarify this in the revision.
>
> 7. In the ARVO dataset, the ground-truth for each vulnerability is defined directly from the developer-authored patch: all lines modified in the fixing commit are treated as the localization patch. This choice reflects the practical reality that a real fix may span multiple lines, and that each changed line contributes to repairing the vulnerability.
>
> 8. This is a typo; Thank you for pointing this out. We will modify it in the revision.
>
> 9. SWE-Agent is designed for high-level code editing tasks, typically repairing functional defects or passing test suites, using planning and tool interactions, but it does not use runtime crash traces or sanitizer logs, or output line-level localization aligned to developer patches, which is the core requirement of T2L-ARVO. In contrast, T2L-Agent is built specifically for runtime evidence-guided bug localization on multiple files, reproducible real-world failures.
>
> 10. The highest numbers (58.0% detection, 54.8% localization) come from the full T2L-Agent configuration and therefore appear in Table 2, which presents the complete ablation including the full system. In the revision, we will (1) explicitly label which rows are baselines and which correspond to T2L-Agent, and (2) surface the full-system best results earlier in the main text/tables. This will remove ambiguity and make the performance contributions of T2L-Agent immediately visible.

---

### Official Review · Reviewer_7K5h · 2025-10-25

**Soundness:** 2
**Presentation:** 2
**Contribution:** 2
**Rating:** 2
**Confidence:** 5

**Summary:**

This paper presented T2L-Agent, a multi-agent framework for line-level vulnerability localization. Specifically, T2L-Agent integrates three core components i.e.,  Agentic Trace Analyzer, which bridges runtime crash evidence and static code structure, Detection Refinement, which iteratively improves predictions based on runtime and semantic feedback, and  Divergence Tracing, which explores multiple hypotheses in parallel to capture distributed vulnerabilities. To evaluate its performance, the authors introduce a new benchmark, including 50 expert-verified, reproducible vulnerability cases, enabling evaluation at both coarse (chunk) and fine (line) granularity. Experiments across multiple models (GPT-5, GPT-4, Claude, Qwen, Gemini, etc.) show that T2L-Agent can achieve up to 58% detection and 54.8% localization accuracy.

**Strengths:**

+ focus on a practical topic
+ sound agentic-based approach

**Weaknesses:**

Weakness:

- limited benchmark dataset
- incremental contribution
- no reproduce data
- limited to a subset of CWEs
- performance is low
- no comparison with existing approach

=== comments ===

My first concern is the representativeness of the benchmark dataset, which only contains 50 instances. The scale of the dataset is quite limited, and such a small dataset constrains the generalizability of the findings, i.e., the dataset may not capture the diversity of real-world vulnerability patterns across different languages and domains. I suggest the authors increase the size of the benchmark to address the generalizability issue.

Comparing T2L-Agent to these existing approaches (e.g., D-CIPHER, AgentFL, AutoFL, FuzzGPT, and MirrorFuzz), its contribution appears incremental. Most components, such as trace parsing, AST analysis, and multi-agent refinement, are conceptually similar to existing designs. Also, these tools can be targeted at line-level vulnerability localization with very minor updates.

The benchmark and experiments appear to be restricted to a narrow subset of CWE categories, primarily memory-related or crash-triggering vulnerabilities (e.g., buffer overflow, null pointer dereference, memory leak). While these categories are critical, they only represent a fraction of real-world software vulnerabilities. More CWE types should be evaluated.

Based on the reported results, the absolute performance remains low for practical adoption, i.e., a detection rate of ~58% and localization accuracy of ~54% means the system still fails on nearly half the cases. In real vulnerability triage, missing vulnerabilities can have severe consequences, so higher precision and recall are essential. With such a low performance, the practical value of the proposed approach is limited.

**Questions:**

1. The proposed T2L-ARVO benchmark includes only 50 vulnerability cases. Could you elaborate on the selection criteria?

2. T2L-Agent includes a multi-agent paradigm and integrates runtime trace analysis. Could you clarify what conceptual advances distinguish it from existing agentic frameworks in the related work part?

---

> ### Author Response · Authors · 2025-11-21
>
> 1. Since the original ARVO dataset was designed for human developers to reproduce vulnerabilities, and our goal is to evaluate agents on a newly defined task, the 50 cases in T2L-ARVO were not randomly selected. They were chosen under explicit quality and practicality criteria.
> First, we considered vulnerability type and difficulty. We curated examples to maintain broad coverage across the five major failure families and to ensure that each case is realistic and solvable with the available runtime evidence. We also performed preliminary agent-based checks to confirm that all selected cases function reliably in our evaluation pipeline.
> Second, we accounted for cost and feasibility. The full ARVO corpus contains roughly 6000 cases, and evaluating the entire set is financially impractical. A carefully chosen 50-case subset provides a manageable yet representative benchmark for both our study and the community.
> Third, we emphasized iterability and accessibility. A smaller benchmark allows researchers to run experiments efficiently, explore agent variations without excessive cost, and adopt the dataset with minimal overhead, which encourages broader use.
> We will clarify these selection criteria in the revised version.
>
> 2. Existing agentic frameworks discussed in related work target different tasks, e.g., general tool-use planning or CTF exploit solving, where the goal is high-level reasoning or exploit synthesis rather than line-level, runtime-evidence–guided bug localization. In contrast, T2L-Agent introduces two task-specific conceptual advances:
>  (1) Evidence-centric agent design: T2L-Agent is organized around runtime traces, sanitizer logs, and static alerts as first-class signals. The agent jointly consumes this evidence, generates hypotheses, validates them with diff-ground truth, and performs targeted refinement. Prior agentic systems do not integrate multi-source runtime evidence nor support this localization loop.
>  (2) Localization-oriented multi-agent roles: Our agents are explicitly designed for project-level, line-level localization, not for exploit generation or generic planning. Each role—evidence collection, hypothesis formation, refinement—supports the specific demands of this task (multi-file reasoning, code-slice rereads, diff-based validation).
>  Together, these distinctions enable T2L-Agent to solve a different and more fine-grained task than existing agentic frameworks.
>
> 3. Our benchmark is not designed as a CWE-coverage benchmark; instead, it follows the scope of real-world C/C++ vulnerabilities in ARVO. ARVO itself is a corpus of naturally occurring, developer-fixed bugs from real projects rather than a synthetic dataset organized by CWE taxonomy. Because C/C++ failures that provide reproducible runtime evidence are overwhelmingly dominated by memory-safety issues such as buffer errors, null dereferences, and lifetime mismanagement, the curated T2L-ARVO subset naturally reflects this distribution. In other words, the apparent concentration in certain CWE types is a property of real-world C/C++ crash-inducing vulnerabilities, not a design choice to target specific CWE categories.

---

### Official Review · Reviewer_21Fk · 2025-10-28

**Soundness:** 2
**Presentation:** 3
**Contribution:** 3
**Rating:** 4
**Confidence:** 3

**Summary:**

This paper proposes T2L-Agent and the T2L-ARVO benchmark, aiming to address project-level vulnerability discovery through a multi-agent and iterative runtime feedback framework. On T2L-ARVO, T2L-Agent achieves 58.0% detection accuracy and 54.8% line-level localization accuracy.

**Strengths:**

The motivation is solid: tackling project-level vulnerability discovery goes beyond the more common function-level localization tasks, and it’s closer to real-world scenarios. The method design also appears intuitive and reasonable. However, I find the evaluation section somewhat disappointing, and I have several concerns below.

**Weaknesses:**

### Major: no baseline.

- The authors introduce many agents developed for the NYU CTF Bench, yet none of them are used for comparison. Conceptually, CTF tasks also require exploring the entire project to locate vulnerabilities, so these should serve as intuitive baselines for comparison. In particular, I don’t see any essential difference between D-CIPHER’s planner–executor framework and T2L-Agent’s design. I’m curious: what prevents existing agents such as D-CIPHER from adapting to T2L-ARVO (e.g., by simply adjusting prompts to output chunks or lines)?
- In Section 3, the authors note that methods like BAP and LLMAO already perform line-level localization, though they "still rely on limited runtime evidence" or "are not multi-agent systems". This implies that such methods can tackle the problem, just less effectively. However, there’s no experimental evidence to support this claim.
The paper should include direct comparisons to clarify what challenges T2L-Agent uniquely overcomes, how it differs from existing agents, and what exactly enables its improved performance.

Additionally, even if no specialized baselines are available, general-purpose repository-level agents such as SWE-Agent could be easily adapted (e.g., by prompting them to output chunk- or line-level results). I strongly recommend including such agentic baselines to demonstrate whether the observed improvements come from the proposed T2L-Agent itself or simply from the model’s inherent capabilities.

### Minor: limited evaluation scope

- The concept of a “vulnerability” is inherently tied to how it can be exploited [1]. It’s unclear how a model can learn to fix vulnerabilities without any information about their *exploitation*. If the model can fix them without this context, these issues might represent general *bad code smells* rather than true *security vulnerabilities*—which have normal functionality and typically remain harmless until intentionally exploited by an attacker. From this perspective, the paper doesn’t fully bridge the key research–practice gap. That said, since most current vulnerability-localization work shares this limitation [1], this is a minor issue in comparison.
- The evaluation scale seems small. Conclusions drawn from just 50 vulnerabilities may fluctuate significantly, especially given that accuracy differences among models are modest. I also don’t quite understand how the paper reports metrics like 44.3% based on only 50 samples—shouldn’t 2% be the smallest possible unit? Perhaps I missed something, but the authors should clarify how the metrics were computed.

### Reference
[1] Risse, Niklas, Jing Liu, and Marcel Böhme. “Top score on the wrong exam: On benchmarking in machine learning for vulnerability detection.” Proceedings of the ACM on Software Engineering 2.ISSTA (2025): 388–410.

**Questions:**

- Why are existing project-level or CTF-style agents (e.g., D-CIPHER, LLMAO, BAP) not included as baselines?
- What specific challenges does T2L-Agent address that existing agents cannot? Please show some evidence directly related to such challenges.
- How does the model learn to detect or fix vulnerabilities without access to information about their exploitation?
- Could the authors justify the small evaluation set (50 vulnerabilities) and explain how metrics like 44.3% are derived from such a limited sample size?

---

> ### Author Response · Authors · 2025-11-21
>
> 1. Our T2L-Agent addresses a different task: runtime-evidence–guided bug localization on real-world, multi-file projects (T2L-ARVO), unlike CTF systems focused on binary-level exploit CTF challenges.
> Our idea stems from asking whether agentic systems successful in CTFs can also support fine-grained localization. Both domains share exploit reasoning, code understanding, and environment-driven task execution. This alignment suggests that a CTF-style, containerized format can naturally support fine-grained localization tasks, allowing agents to operate within the same structured workflow used for security challenges. Each sample in our dataset may have multiple vulnerabilities that will increase the task difficulty.
> Both T2L and D-CIPHER use a planner–executor architecture, which is one of the common architectures used in the current long context task planning agents, but the rest diverge: (1) environment—project-level for T2L vs. challenge-level for D-CIPHER; (2) evidence—ATA provides runtime trace tools absent in D-CIPHER; (3) evaluation—we compute multi-location coverage, whereas D-CIPHER uses solved/unsolved. D-CIPHER is optimized for exploit synthesis on binaries, not line-level bug localization. Adapting it to ARVO would require new reward signals, source-aware extraction, and diff-based supervision—far beyond prompt tuning.
>
> 2. BAP and LLMAO are line-level localization methods, but in a very different problem setting. LLMAO analyzes single-file fragments, not multi-file project contexts, using only static code, without runtime traces. BAP likewise assigns attention-based scores to individual lines by probing a frozen LLM and operates purely on static source code without dynamic evidence. In contrast, T2L-Agent is designed for project-level, runtime-evidence–guided localization on T2L-ARVO: it must integrate crash traces, sanitizer logs, and static alerts across multiple files and then iteratively refine candidates to match the ground-truth diff at the line level. Reusing existing methods like BAP/LLMAO on ARVO would require substantial architectural changes (e.g., trace/log ingestion and cross-file context). We will clarify this task difference more explicitly in Sec. 3.
> And both LLMAO and BAP are evaluated primarily on Defects4J, where bugs are typically confined to a single file or even a single function.  In contrast, ARVO contains project-scale, multiple-file vulnerability cases. Thus, the problem setting tackled by LLMAO/BAP is fundamentally different and significantly less complex than the one targeted by T2L-Agent.
>
> 3. Our T2L-ARVO is not an exploitability benchmark; it focuses on a widely studied subproblem shared across recent work, the bug localization from program crashes and sanitizer evidence, without requiring exploit demonstrations. Our dataset contains real bugs from real projects, each paired with its developer-issued security and bug-fix patch, which reflects the community’s operational notion of a vulnerability even when an exploit has not yet been used.
> And industry practice like OSS-Fuzz routinely treats these as security-relevant. T2L-Agent targets this practical setting: given runtime evidence and code context, identify and localize the fault that developers ultimately patched.
> We agree this does not model adversarial exploitation end-to-end, but this limitation is shared by most vulnerability-localization work. Our contribution is therefore to improve evidence-guided localization under realistic, project-level conditions, not to infer exploitability.
>
> 4. We appreciate the concern. Although T2L-ARVO contains 50 vulnerabilities, each sample includes multiple ground-truth bug locations. For a sample with 3 ground-truth locations, if the agent correctly identifies 1 of them, the sample-level accuracy is 1/3 = 0.33%, and overall accuracy is computed by aggregating these per-location outcomes. Thus, metrics like 44.3% reflect hundreds of evaluated locations across the dataset—not 50 points—and are not restricted to 2% increments.
>  Regarding scale: As we mentioned in the appendix, the ARVO dataset is not a well-balanced dataset, as we identify the common vulnerability type composition of the ARVO dataset as shown in Appendices 1 and 2. We purposely curated 50 cases under controlled constraints (vulnerability type, difficulty, realism) to ensure balance and reproducibility. The underlying ARVO corpus contains ~6000 samples, but even at $1 per run, full evaluation would be prohibitively expensive. A 50-case subset also enables rapid research iteration and makes the benchmark accessible for the community to experiment with, which we view as important for adoption and future extension. We will clarify these points in the revision.

---

> > ### Comment · Reviewer_21Fk · 2025-11-25
> >
> > Thank you for your clarification. I understand that D-CIPHER is not suitable for this task, and BAP/LLMAO are weaker and have a much more strong assumption for the task, as the vulnerable single-file segments are given. Therefore, I think a possible experiment design is giving the same location information to T2L-Agent and make a fair comparison with BAP/LLMAO on those single-file cases.
> >
> > In addition, as I suggested in Weakness part, SWE-Agent should naturally be a good baseline, as it is really simple and can be easily adjust for your task by changing prompts. I know it is not specialized for vulnerabilities, but at least it can provide a view of how good T2L-Agent is. I think T2L-Agent should show a clear advantage over it to claim its contribution. To reduce the engineering effort, Mini-SWE-Agent might be also good for comparison. Anyway, without a baseline, we are not sure how good T2L-Agent is.

---

### Official Review · Reviewer_XLJa · 2025-11-01

**Soundness:** 2
**Presentation:** 1
**Contribution:** 2
**Rating:** 2
**Confidence:** 3

**Summary:**

The paper aims to solve the problem of vulnerability localization.
This is an important and challenging problem.
Many vulnerabilities (e.g., an NPD) may have different locations for the root cause and the symptom (the point of crash).
This paper proposes an agent to solve the vulnerability localization problem.

It proposes a new benchmark containing 50 runnable cases with vulnerabilities. The cases are validated by human experts. The proposed agent considers different sources of information (e.g., static information, debugger, crash stack traces, etc.), and iteratively refine generated hypothesis until a final conclusion is made.

**Strengths:**

The paper targets a very important and challenging problem.
It proposes benchmark with human labels.
The proposed technique is comprehensive and systematic.

**Weaknesses:**

However, I find it very difficult to understand the technique details of the proposed technique.
The writing about key techniques is vague and hinders a fair evaluation of the work.
While I believe it is a solid work, a thorough revision is necessary to make the paper better appreciated by the community.


Here are some key details that are missing:

1. In 4.1, the authors mention the proposed agent first collect evidence, then make hypothesis, then iteratively refine those components.
Here are some key details that are missing:
(1) the evidences from different components may support/contradict with each other, how the agent formulate the relationship between different evidences?
(2) how the agent makes hypotheses? Different hypotheses may have dependence with each other. How the agent systematic validate/update the belief across different hypotheses?
(3) How exactly the refinement is achieved? What are the information that are updated? How the updated information is propagated across different hypotheses?

2. What is the relationship between T2L-Agent and ATA? Is ATA a component of the T2L-Agent?

3. At line 214, the authors mention "we score and sort candidates by how well they match across multiple signals". How exactly is this score computed?

4. At line 236, the authors mention "On a second pass, it rereads those slices to find missed patterns by checking syntactic cues, semantic links such as data and control flow, and alignment with runtime evidence." How this is achieved? What are syntactic cues and semantic links? How to justify the re-read stage of the slices? How does it different from the first stage? What is the benefit of having two stages?

5. At line 285, the authors mention "We combine automated screening with expert review to ensure both realism and balance.". How realism is evaluated? What is the labeling process of identifying the ground-truth root cause of a vulnerability?

**Questions:**

Please see the clarification questions in the above discussion.

---

> ### Author Response · Authors · 2025-11-21
>
> 1. (1). We concatenate all available evidence and feed it jointly to the LLM. The agent does not form locations by aggregating per-source predictions; instead, the LLM reasons over the combined evidence and outputs candidate locations directly. Hence, there is no per-evidence contradiction resolution step.
> (2). Hypotheses are generated from the same combined evidence by the LLM. We validate each hypothesis independently against the ground-truth diff, without cross-hypothesis belief sharing.
> (3). Refinement is driven by the code chunks the LLM finds most relevant in its current pass. We focus on those chunks, which yield new candidate locations; the candidate set is updated, and the process iterates until convergence or the budget is reached.
>
> 2. ATA (Agentic Trace Analyzer) is a core component within the overall T2L-Agent framework. T2L-Agent is the end-to-end agent framework that performs the overall “trace-to-line” workflow:
> (1). Collects evidence from runtime and static sources
> (2). Integrates evidence to form hypotheses
> (3). Iteratively refines location predictions until line-level localization is achieved.
> ATA is the key module inside T2L-Agent that handles the evidence integration part, it’s the component that processes runtime traces, sanitizer logs, and static analysis outputs, transforming them into a unified representation that can be understood by the LLM.
>
> 3. At line 214, the “score” refers to the confidence value produced by the LLM for each candidate location.
> During the evidence analysis stage, after integrating all collected signals, the LLM outputs a list of candidate locations each annotated with a confidence tag chosen from {very_unlikely, unlikely, possible, likely, very_likely}.We then sort candidates according to these confidence levels, treating higher-confidence predictions as stronger matches “across multiple signals.”
>
> 4. At line 236, the “second pass” refers to the refinement stage of our two-pass agent process.
> First pass (proposal stage)
> The LLM receives all collected evidence, including runtime traces, logs, and static analysis results, and uses this information to generate initial candidate locations. This stage focuses on broad detection and identifies positions that directly correspond to the available evidence.
> Second pass (refine stage):
>  After the first pass, the agent identifies the code chunks that the LLM showed interest in. These code chunks are then fed back into the LLM together with the previous evidence to let it generate new or refined location candidates.
>  The goal of this reread is to improve coverage and allow the LLM to reason with more contextual details from the actual source code around suspected areas.
> Difference from first stage:
> The first pass reasons only on aggregated evidence. The second pass augments that reasoning with actual code content from LLM-selected chunks, enabling deeper and more precise localization.
> Benefit:
> This two-pass process increases coverage and accuracy because the LLM can revisit relevant code with full context, catching potential faults that may not have been evident from evidence alone.
>
> 5. “Realism” is quantitatively assessed using diff-based structural metrics—files changed, architectural spread, directory depth—and semantic factors such as cross-module coupling, interface changes, and concurrency touchpoints. We then apply dual validation:
> (1) expert review to verify reproducibility and representativeness, and (2) LLM-assisted checks the agent-facing difficulty. Ground-truth root causes are obtained from the developer diff: the patched lines are indexed to establish precise line-level localization baselines. We use a structured expert review to ensure that all selected challenges fall within a moderate and reasonable difficulty level. Following the factors outlined earlier. Although the ARVO dataset was originally designed for human reproducibility, our curation adjusts the difficulty so that agents face challenges that are balanced and suitable for meaningful evaluation.

---

### Meta-Review · Area_Chair_wuQF · 2026-01-09

**Summary:**

This paper addresses an important problem—project-level, line-level vulnerability localization—and proposes a multi-agent framework  along with an expert-verified benchmark. Reviewers appreciated the motivation and the effort to build a runnable dataset, but the committee recommends rejection due to several significant weaknesses. Most notably, the evaluation lacks strong baselines or direct comparisons to existing agentic systems (e.g., SWE-Agent or CTF-style agents) and prior localization methods, making it difficult to assess the true benefit of the proposed framework. In addition, multiple reviewers found the method description and key technical details (evidence integration, hypothesis refinement, scoring, and ground-truth labeling) insufficiently clear in the paper, limiting reproducibility and confidence in the claims. Finally, concerns were raised about the small benchmark size and limited scope, as well as whether the reported performance is strong enough to justify acceptance. Overall, while promising, the work requires clearer exposition and stronger comparative evidence to support its contributions.

**Reviewer Concerns:**

The rebuttal helped clarify several points that were confusing in the initial submission, including the intended relationship between T2L-Agent and ATA, the meaning of candidate “scoring,” the two-pass refinement procedure, and how the benchmark ground truth and metric granularity are computed (e.g., multiple locations per case explaining non-2% increments). It also provided a more concrete justification for the benchmark size and described selection criteria in greater detail.

However, several major concerns remain outstanding. Most importantly, the rebuttal does not resolve the lack of empirical baselines: reviewers explicitly requested comparisons to adapted agentic systems such as SWE-Agent / Mini-SWE-Agent or to simplified settings enabling comparison with BAP/LLMAO, and these experiments are still missing. Additionally, while the rebuttal explains methodological choices, the underlying issue of paper clarity and completeness remains—many details are provided only in response rather than being clearly documented in the submission itself. Finally, concerns about the benchmark’s limited scale/diversity and whether current performance is sufficient for practical impact were acknowledged but not fully mitigated by new evidence.

**Reviewer Scores:**

Speculatively, fuller discussion would likely not change the scores of 7K5h (2→2) and 21Fk (4→4), since both anchor on the still-missing baseline comparisons (and 21Fk explicitly reiterated the SWE-Agent/Mini-SWE request after rebuttal). XLJa (2→2, possibly 2→3) and 3t1k (2→3) might move slightly upward because the rebuttal clarifies several technical ambiguities (agent components, scoring/refinement, ground truth/metrics), but both would still likely remain below acceptance without stronger empirical baselines and clearer exposition in the paper itself.

---

### Decision · Program_Chairs · 2026-01-26

Reject